# Single-component multilayered self-assembling protein nanoparticles presenting glycan-trimmed uncleaved prefusion optimized envelope trimers as HIV-1 vaccine candidates

Yi-Nan Zhang [1,7], Jennifer Paynter[1,7], Aleksandar Antanasijevic[1,2,7], Joel D. Allen [3,7], Mor Eldad [1], Yi-Zong Lee[1], Jeffrey Copps [1,2], Maddy L. Newby[3], Linling He[1], Deborah Chavez[4], Pat Frost[4], Anna Goodroe[4], John Dutton[4], Robert Lanford[4], Christopher Chen [4], Ian A. Wilson [1,2,5], Max Crispin [3], Andrew B. Ward [1,2] & Jiang Zhu [1,6] ✉

Uncleaved prefusion-optimized (UFO) design can stabilize diverse HIV-1 envelope glycoproteins (Envs). Single-component, self-assembling protein nanoparticles (1c-SApNP) can display 8 or 20 native-like Env trimers as vaccine candidates. We characterize the biophysical, structural, and antigenic properties of 1c-SApNPs that present the BG505 UFO trimer with wildtype and modified glycans. For 1c-SApNPs, glycan trimming improves recognition of the CD4 binding site without affecting broadly neutralizing antibodies (bNAbs) to major glycan epitopes. In mice, rabbits, and nonhuman primates, glycan trimming increases the frequency of vaccine responders (FVR) and steers antibody responses away from immunodominant glycan holes and glycan patches. The mechanism of vaccine-induced immunity is examined in mice. Compared with the UFO trimer, the multilayered E2p and I3-01v9 1c-SApNPs show 420 times longer retention in lymph node follicles, 20-32 times greater presentation on follicular dendritic cell dendrites, and up-to-4 times stronger germinal center reactions. These findings can inform future HIV-1 vaccine development.

The HIV-1 envelope glycoprotein (Env) mediates cell entry and is the target of the host humoral immune response. Functional Env is a trimer of noncovalent gp120-gp41 heterodimers, which are generated from the proteolytic cleavage of a gp160 precursor[1]. In addition to substantial genetic diversity caused by an error-prone reverse transcriptase[2], HIV-1 exploits two Env-dependent strategies to evade host immunity: (*i*) Env metastability causes Env misfolding, sheds gp120, and leaves gp41 stumps on the virus[3], and (*ii*) host-produced

[1]Department of Integrative Structural and Computational Biology, The Scripps Research Institute, La Jolla, CA 92037, USA. [2]IAVI Neutralizing Antibody Center, The Scripps Research Institute, La Jolla, CA 92037, USA. [3]School of Biological Sciences, Highfield Campus, University of Southampton, Southampton SO17 1BJ, UK. [4]Southwest National Primate Research Center, Texas Biomedical Research Institute, San Antonio, TX 78227, USA. [5]The Skaggs Institute for Chemical Biology, The Scripps Research Institute, La Jolla, CA 92037, USA. [6]Department of Immunology and Microbiology, The Scripps Research Institute, La Jolla, CA 92037, USA. [7]These authors contributed equally: Yi-Nan Zhang, Jennifer Paynter, Aleksandar Antanasijevic, Joel D. Allen. ✉e-mail: jiang@scripps.edu

glycans form a dense "glycan shield" to mask the Env surface[4]. Nonetheless, up to 30% of infected people eventually develop serum neutralization against the majority of circulating HIV-1 strains. Hundreds of broadly neutralizing antibodies (bNAbs) have been isolated from HIV-1-infected donors[5] and target seven different epitope regions on Env: the CD4 binding site (CD4bs), three peptide-glycan epitopes (the V2 apex, V3 base, and gp120-gp41 interface), silent face (SF), fusion peptide (FP), and membrane-proximal external region (MPER)[6,7]. These bNAbs and the characterization of their epitopes have thus provided a number of templates for "antibody-guided" rational vaccine design[8–11].

The design and characterization of native-like Env trimers marked a milestone in modern HIV-1 research[12,13]. While gp120-derived constructs are promising CD4bs immunogens[14–18], ongoing HIV-1 vaccine efforts have largely shifted to Env trimers, as exemplified by SOSIP[19–22], native flexibly linked (NFL)[23–25], and uncleaved prefusion-optimized (UFO) trimers[26–28]. As the first rational trimer design, SOSIP transformed the HIV-1 vaccine field and generated a wealth of information on Env structure and immunogenicity[29]. Both SOSIP and NFL are empirical designs that may require additional mutations to improve trimer stability (e.g., SOSIP.v9[30]). In contrast, UFO can eliminate the root cause of Env metastability with only two implementations, UFO and UFO-BG, which have been applied to diverse Envs[26–28]. While Env metastability can now be dealt with by these three trimer platforms, the glycan shield remains a challenge due to its complex roles in Env-bNAb interactions[31–36]. Liquid chromatography–mass spectrometry (LC-MS)[37–39] and structural analysis by X-ray crystallography[40–46] and cryo-electron microscopy (cryo-EM)[47–49] have revealed critical details of Env glycosylation and bNAb-glycan interactions, respectively. These studies placed a premium on mimicking the glycan shield of functional viral Env in HIV-1 vaccine design[50]. Although "multistage" vaccine strategies originated from the evolutionary analysis of bNAbs[51], glycans have become a major consideration. Sequential Env immunogens with glycan deletion and reintroduction were designed to guide B cell maturation toward bNAbs[52–54]. Undesirable Env immunodominance is another confounding factor in bNAb elicitation using soluble trimers, as exemplified by glycan holes and a neoepitope at the trimer base[55,56]. The former (small, exposed protein surfaces within the glycan shield) are the primary target of autologous tier-2 NAbs in rabbits[56–65], whereas the latter elicits non-neutralizing antibodies (nNAbs) in multiple species[63,66,67] and causes trimer disassembly during immunization[68]. In rodents, glycan epitopes at C3/V4 and C3/V5 accounted for the autologous NAb response induced by BG505 Env[69–71]. In nonhuman primates (NHPs), the BG505 SOSIP trimer elicited an autologous NAb response to C3/465[60,72]. Therefore, the native-like trimer-induced autologous NAb response appears to be dominated by Env surfaces that either lack or overexpress glycans, leading to contradictory views on the glycan shield in vaccine design, i.e., glycan nativeness[50] vs. glycan removal or addition[52–54,73]. The manipulation of individual glycans may improve targeting of a bNAb epitope or avoidance of a glycan hole but may not provide a general vaccine solution as it could be Env- or strain-specific.

The delivery of Env antigens in multimeric forms to mimic virus-like particles (VLPs) has become a trend in recent vaccine research for HIV-1[74–77], as well as other viruses[78–80]. With virus-like size and shape as well as a dense display of surface antigens, VLPs can induce a more potent and long-lasting immune response than soluble antigens[81–85]. Engineering protein nanoparticles (NPs) to mimic authentic VLPs has been the driving force behind several recent technological advances in the vaccine field[80]. Three technologies, SpyTag/SpyCatcher (termed "SPY")[86], two-component NPs (2c-NPs)[87], and single-component self-assembling protein NPs (1c-SApNPs)[27,88,89], have each been assessed for multiple viral targets. SPY utilizes an isopeptide bond within a bacterial adhesin to covalently link antigens to VLPs or NPs[90]. The 2c-NP and 1c-SApNP platforms rely on in vitro (cell free) and in vivo (in producer cells) assembly, respectively. In terms of vaccine production, SPY and

2c-NP achieve greater versatility at the price of increased complexity and cost, as they both involve multiple plasmids, expression systems, and purification methods. The 1c-SApNPs require a highly optimized antigen and NP carrier, which are encoded within a single plasmid, to achieve manufacturability and quality[91,92]. VLPs and NPs are more immunogenic because of their advantages in antigen trafficking to lymph nodes, high-avidity follicular dendritic cell (FDC) interactions, antigen-presenting cell (APC) activation, and germinal center (GC) reactions, which involve both innate and adaptive systems[77,93]. A recent study suggested that the density of mannose patches is critical for the FDC targeting of HIV-1 Env-NP immunogens[94,95]. Stabilized HIV-1 Env trimers attached to iron oxide NPs or aluminum salt have also been tested as particulate vaccine formulations[96,97].

Here, we displayed the BG505 UFO trimer on multilayered 1c-SApNP platforms[91,92] as our next-generation HIV-1 vaccine candidates. A panel of Env-NP constructs was first designed for biochemical, biophysical, and structural characterization to facilitate immunogen selection. Cryo-EM yielded high-resolution structures for the multilayered E2p and I3-01v9 1c-SApNPs that each present 20 BG505 UFO trimers, with localized reconstructions obtained for the trimer at 7.4 and 10.4 Å, respectively. Extensive glycan and antigenic profiling of the BG505 UFO trimer and 1c-SApNPs bearing wildtype and modified glycans revealed distinct features. Unexpectedly, endoglycosidase H (endo H) trimming of the glycan shield retained NP binding to NAbs/bNAbs that require specific glycans for Env recognition and target the CD4bs. Negative-stain EM revealed differential effects of glycan trimming on the angle of approach for bNAbs PGT128 and VRC01. Next, we immunized mice and rabbits with UFO trimers and 1c-SApNPs bearing the wildtype and trimmed glycans, which were formulated with conventional adjuvants. Glycan trimming diverted NAb responses away from known glycan holes and glycan patches, while increasing the frequency of vaccine responders (FVR). The beneficial effect of glycan trimming on tier 2 NAb elicitation was confirmed in NHP studies. Lastly, we analyzed vaccine delivery and immune responses at the intra-organ, intercellular, and intracellular levels in the mouse model. Compared with the soluble trimer, the multilayered E2p and I3-01v9 1c-SApNPs showed substantially improved retention, presentation, and GC reactions in lymph node tissues. Glycan trimming had little effect on 1c-SApNP trafficking to, and retention in, lymph nodes. Intact 1c-SApNPs and adjuvants in lymph node tissues were directly visualized by transmission electron microscopy (TEM). Our study thus presents a promising strategy to overcome the challenges posed by the Env glycan shield in HIV-1 vaccine development based on the UFO trimer and 1c-SApNP platforms.

## Results

### Rational design and characterization of BG505 UFO trimer-presenting 1c-SApNPs

In the UFO design paradigm, the N terminus of heptad repeat 1 (HR1$_N$, aa 547–569) is the trigger of HIV-1 Env metastability and must be shortened and optimized[26,27]. Crystal structures were determined for clade A, B, and C Envs bearing the HR1$_N$, UFO, and UFO-BG designs[26,27,71,98]. The HR1$_N$-redesigned BG505 trimer was displayed on ferritin (FR), E2p, and I3-01 NPs[27,88]. The reengineered I3-01, when formulated with a Toll-like receptor 3 (TLR3) agonist adjuvant, elicited a robust tier 2 NAb response in mice for the first time[27,71]. We have also applied these protein NP platforms to display rationally designed antigens for other viruses[89,91,92].

Here we revisited the HIV-1 Env-NP vaccine design based on the high-resolution BG505 UFO trimer structure[98] and newly developed "multilayered" 1c-SApNP platforms[91,92]. We first modeled the BG505 UFO trimer (PDB ID: 6UTK) on wildtype FR, E2p, and I3-01, resulting in particles with diameters of 30.2, 41.7, and 46.1 nm, respectively (Fig. 1a). The computationally optimized 8-aa HR1$_N$ bend ("NPDWLPDM") forms a 24-Å triangle in the center of each UFO trimer

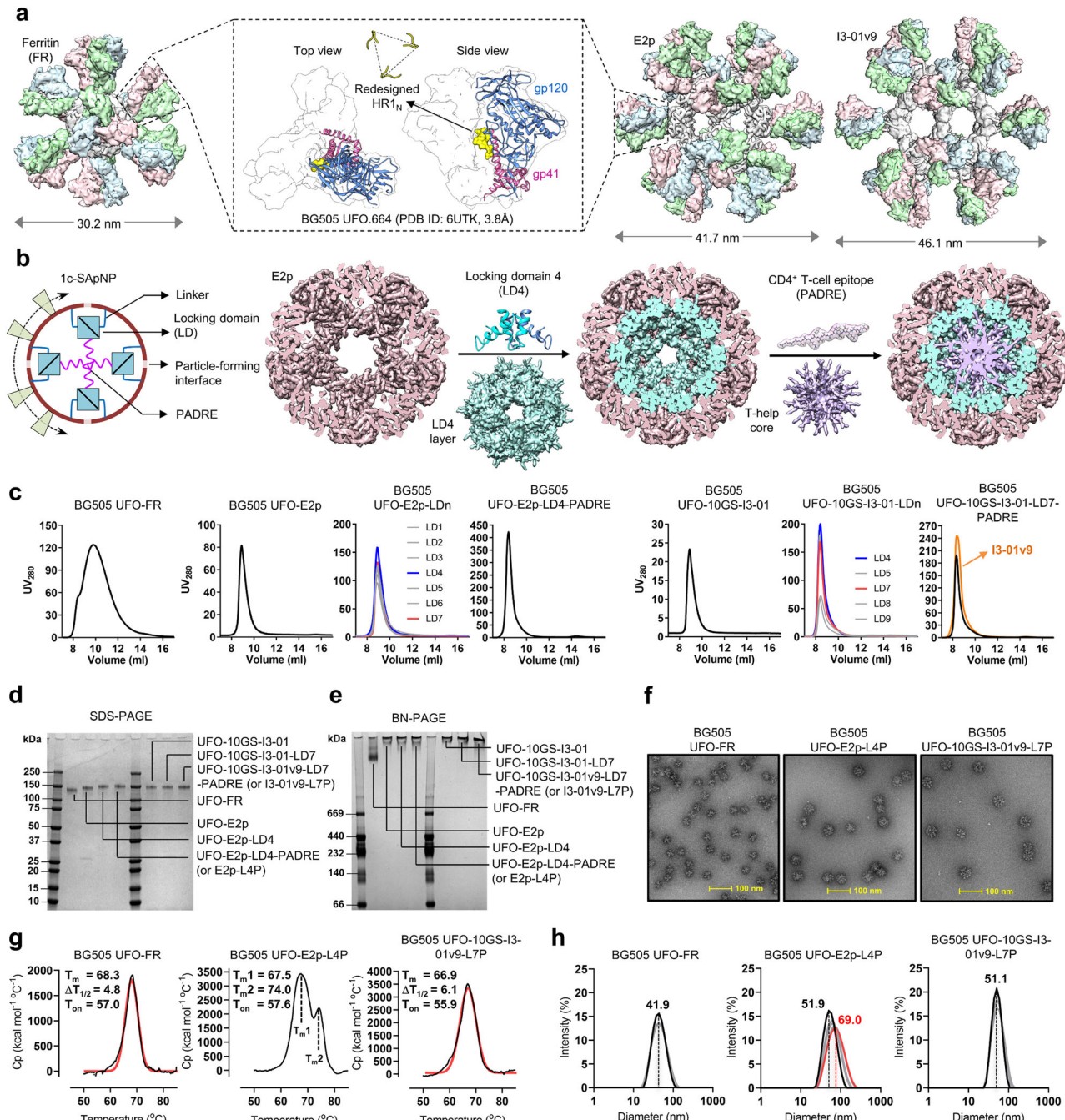

**Fig. 1 | Rational design and characterization of BG505 UFO trimer-presenting nanoparticles. a** BG505 UFO Env (middle) and three UFO trimer-presenting SApNPs (FR: left; E2p and I3-01: right). In the inset, ribbon representation of BG505 UFO gp140 protomer (PDB ID: 6UTK) is shown in transparent molecular surface with gp120 in cornflower blue, gp41 in magenta, and the redesigned HR1$_N$ loop in yellow (as both ribbon and molecular surface models). Surface representations are shown for FR 24-mer and E2p/I3-01 60-mers presenting 8 and 20 BG505 UFO trimers, respectively. The NP carrier is colored in gray and the three protomers of each BG505 UFO trimer are colored differently. The NP size is indicated by diameter (in nm). **b** Schematic representation of a multilayered NP (left), which contains an inner layer of locking domains (LD) and a cluster of T-help epitopes (PADRE), and molecular surface model of E2p (brown, PDB ID: 1B5S) with stepwise incorporation of LD4 (cyan, PDB ID: 2MG4) and PADRE (purple). **c** SEC profiles of BG505 UFO trimer-presenting FR (panel 1), E2p (panel 2), E2p-LD1-7 (panel 3), E2p-LD4-PADRE (or E2p-L4P, panel 4), I3-01 (panel 5), I3-01-LD4-5/7-9 (panel 6), and I3-01/v9-LD7-PADRE (or I3-01/v9-L7P, panel 7) after PGT145 purification. **d** and **e** Reducing SDS-PAGE and BN-PAGE, respectively, of BG505 UFO-FR, E2p, and I3-01/v9 SApNPs after PGT145 purification, with LD and PADRE variants included for E2p and I3-01. **f** Negative-stain EM images of PGT145-purified BG505 UFO-FR, E2p-L4P, and I3-01v9-L7P SApNPs. **g** Thermostability of BG505 UFO-FR, E2p-L4P, and I3-01v9-L7P SApNPs, with $T_m$, $\Delta T_{1/2}$, and $T_{on}$ measured by DSC. **h** Particle size distribution of BG505 UFO-FR, E2p-L4P, and I3-01v9-L7P SApNPs. The hydrodynamic diameter ($D_h$) was measured by DLS. Multiple measurements were obtained for each SApNP, with the average particle size labeled on the distribution plot.

(Fig. 1a, middle). For FR and E2p, the superposition of the BG505 gp140 C termini onto the N termini yielded C$_\alpha$ root-mean-square deviations (RMSDs) of 1.0 and 11.4 Å, respectively. A 3-aa "ASG" linker, with "AS" resulting from the restriction site between the Env and NP genes, enabled the proper display of Env trimers on FR and E2p. For I3-01, a 10-aa (G$_4$S)$_2$ linker was added to overcome the large spacing (50.5 Å) between the N termini of three subunits around each three-fold axis[27,89,91,92]. We then reexamined the multilayered nanoparticle design

concept using a modeling approach (Fig. 1b). Previously, we incorporated a locking domain (LD) and a pan-reactive CD4[+] T-cell epitope (PADRE[99]) into E2p and I3-01 60-mers via gene fusion[91,92]. We hypothesize that antigens will fold properly on the NP surface upon self-assembly, while LD and PADRE may form an inner layer and a hydrophobic core to stabilize the NP shell and deliver a strong T-helper signal, respectively (Fig. 1b, left). E2p-LD4-PADRE, which was used as a carrier for EBOV GP[92], was modeled to visualize the multilayered structure (Fig. 1b, right). Briefly, E2p forms a protein shell of 23.2 nm with a distance of 12.8 nm across the hollow interior measured at the inward-facing tip (G382). LD4, a 58-aa dimeric protein domain (PDB ID: 2MG4), was fused to the C terminus of E2p at the dimeric NP-forming interface with a 5-aa $G_4S$ linker, resulting in a smaller protein shell of 14.4 nm integrated with the outer E2p shell. A 13-aa PADRE with an extended backbone was fused to the C terminus of LD4 with a 2-aa GS linker, resulting in a hydrophobic core of ~5 nm. Similarly to the EBOV vaccine study[92], a set of multilayered BG505 UFO Env-NP constructs was selected for experimental validation.

Nineteen BG505 UFO Env-NP constructs were transiently expressed in 100-ml ExpiCHO cells, followed by PGT145 purification and size-exclusion chromatography (SEC) on a Superose 6 column (Fig. 1c). Overall, the BG505 UFO trimer can be displayed on wildtype FR, E2p, and I3-01 NPs with high purity. Similar to our previous study[92], E2p-LD4 and I3-01-LD7 showed the highest NP yield among various NP-LD combinations. Adding PADRE increased the NP yield by 5.2 to 8.6 times relative to the wildtype NPs, measured by ultraviolet (UV) absorbance at 280 nm. I3-01v9, a variant of I3-01[27], further improved NP yield by 24% based on the UV value. Seven SEC-purified NP samples were analyzed by sodium dodecyl sulfate–polyacrylamide gel electrophoresis (SDS-PAGE) under reducing conditions (Fig. 1d). E2p and I3-01 registered higher bands than FR on the gel (~150 kDa). These seven NP samples were then analyzed by blue native-polyacrylamide gel electrophoresis (BN-PAGE) (Fig. 1e). While FR displayed a distinctive NP band, none of the E2p and I3-01 NPs migrated down from the chambers due to their large size. The SEC-purified FR, E2p-LD4-PADRE (E2p-L4P), and I3-01v9-LD7-PADRE (I3-01v9-L7P) were visualized by negative-stain EM (Fig. 1f). Thermostability was assessed by differential scanning calorimetry (DSC) (Fig. 1g and Fig. S1a). The melting temperature ($T_m$) of the three NPs was comparable to that of the BG505 UFO trimer (66.9–68.3 °C vs. 68.4 °C)[27]. The multilayered E2p displayed a second peak in the thermogram ($T_{m2}$ = 74.0 °C), suggesting a stepwise unfolding process upon heating. The distribution of hydrodynamic diameter ($D_h$) was characterized by dynamic light scattering (DLS) using a Zetasizer (Fig. 1h). UFO trimer-presenting ferritin and I3-01v9 had an average particle size of 41.9 and 51.1 nm, respectively, whereas E2p displayed a wide distribution (51.9–69.0 nm), consistent with the higher polydispersity index (PI) noted for some E2p samples (Fig. S1b). The results suggested that E2p may adopt two distinct states in solution.

The systematic assessment of the two large protein NPs in combination with various LDs and PADRE in the EBOV GP-NP[92] and HIV-1 Env-NP constructs confirmed that E2p-L4P and I3-01v9-L7P are the best combinations and provide a general antigen-display platform. In this study, the BG505 UFO trimer-presenting E2p-L4P and I3-01v9-L7P SApNPs (Fig. S1c) were subjected to detailed in vitro and in vivo characterization to assess their vaccine potential.

## Cryo-EM analysis of multilayered BG505 UFO trimer-presenting E2p and I3-01v9 SApNPs

Cryo-EM played a critical role in determining high-resolution structures of native-like Env trimers[47–49]. Recently, cryo-EM was used to characterize 2c-NPs that display diverse Env trimers[100–103]. In most cases, nanometer resolution was obtained for the NP backbone, with 4–7 Å resolution achieved for the displayed trimers after localized

reconstruction[101,102]. Here, we characterized the two multilayered BG505 UFO trimer-presenting 1c-SApNPs by cryo-EM.

Prior to cryo-EM, the PGT145/SEC-purified BG505 UFO trimer-presenting E2p-L4P and I3-01v9-L7P SApNPs were analyzed by negative-stain EM to confirm their homogeneity (Fig. S2a). In the micrographs, most particles displayed appropriate size and morphology. We then applied cryo-EM to characterize these two SApNPs (Fig. 2), with the data processing protocol described in the Methods section[101,102] and data acquisition parameters listed in Table S1. The BG505 UFO trimers are attached to the E2p and I3-01v9 scaffolds using peptide linkers (Fig. S1c). Owing to conformational flexibility, the Env-corresponding signal (or density) was scattered in the 2D classes and initial 3D maps (Fig. 2a, b, top left). Consequently, we were unable to reconstruct high-resolution maps of full BG505 UFO trimer-presenting E2p-L4P and I3-01v9-L7P SApNPs. To overcome this problem, we applied a previously developed data processing method, in which two flexibly linked entities (i.e., the protein NP scaffold and Env trimer here) are analyzed independently[101]. The resulting maps are represented as a transparent gray mesh (Fig. 2a, b). After localized reconstruction, the resolution of the Env-corresponding portion was improved from 40 Å or lower to 7.4 and 10.4 Å for the E2p and I3-01v9 SApNP immunogens, respectively. The E2p-L4P NP consists of the E2p shell (based on dihydrolipoyl transacetylase, PDB ID: 1B5S[104]), LD4 layer (PDB ID: 2MG4), and PADRE core (Fig. 1b). Focused refinement of the NP core converged on the E2p shell, resulting in a 3.7 Å EM map; and the E2p model was then relaxed into the map (Fig. 2a, right). The resulting structure displayed excellent agreement with the crystal structure of this bacterial enzyme[104] with a $C_\alpha$ RMSD of 0.9 Å, suggesting a negligible effect of the surface-displayed BG505 UFO trimers and encapsulated LD4 layer and PADRE core on particle assembly. Although additional density at the core of the NP and along the five-fold axes was observed in the EM map, attempts to resolve their structures were unsuccessful (Fig. S2b, c). Therefore, although LD4 and PADRE can substantially improve SApNP expression (Fig. 1c), they likely interact with each other in a less well-organized fashion and form a flexible structural network within the E2p shell. The I3-01v9-L7P NP exhibited inherent flexibility due to the 10-aa GS linkers connecting the surface trimers and NP backbone, resulting in relatively low-resolution EM maps. Nonetheless, good fits were obtained by docking the structures of I3-01 (PDB ID: 5KP9[105]) and BG505 UFO trimer (PDB ID: 6UTK[98]) into the corresponding maps (Fig. 2b).

Altogether, cryo-EM confirmed the correct assembly of two multilayered 1c-SApNPs and native-like structural features of surface-displayed BG505 UFO trimers. Given the wide particle size distribution (Fig. 1h), we speculate that both our cryo-EM model and the previously reported crystal structure[104] captured the ground state of E2p, in which the N-terminal loop (A184-S203) is packed against the surface and E2p adopts the most compact particle conformation.

## Site-specific glycan analysis of BG505 UFO immunogens with wildtype and modified glycans

The glycan shield is a defense mechanism used by HIV-1 to escape antibody neutralization[4]. However, the identification of glycan-reactive bNAbs[6] suggests that it can also be exploited as a vaccine target[32,33,36]. The structure of the glycan shield has been extensively characterized to facilitate rational vaccine design[31]. Here, we determined site-specific glycan profiles for BG505 UFO trimers and NPs using LC-MS and a previously established protocol[27,71].

Wildtype BG505 UFO Env glycosylation (Fig. 3a and Fig. S3a) is consistent with previous reports on other BG505 Envs[27,38]. Trimer-associated mannose patches (TAMPs) at N156, N160, N276, and N301 present primarily oligomannose-type glycans, suggesting correct folding of the Env-NP fusion protein. This processing state is conserved across all samples, indicating that particulate display does not disrupt glycan processing near the trimer apex. The intrinsic mannose patch

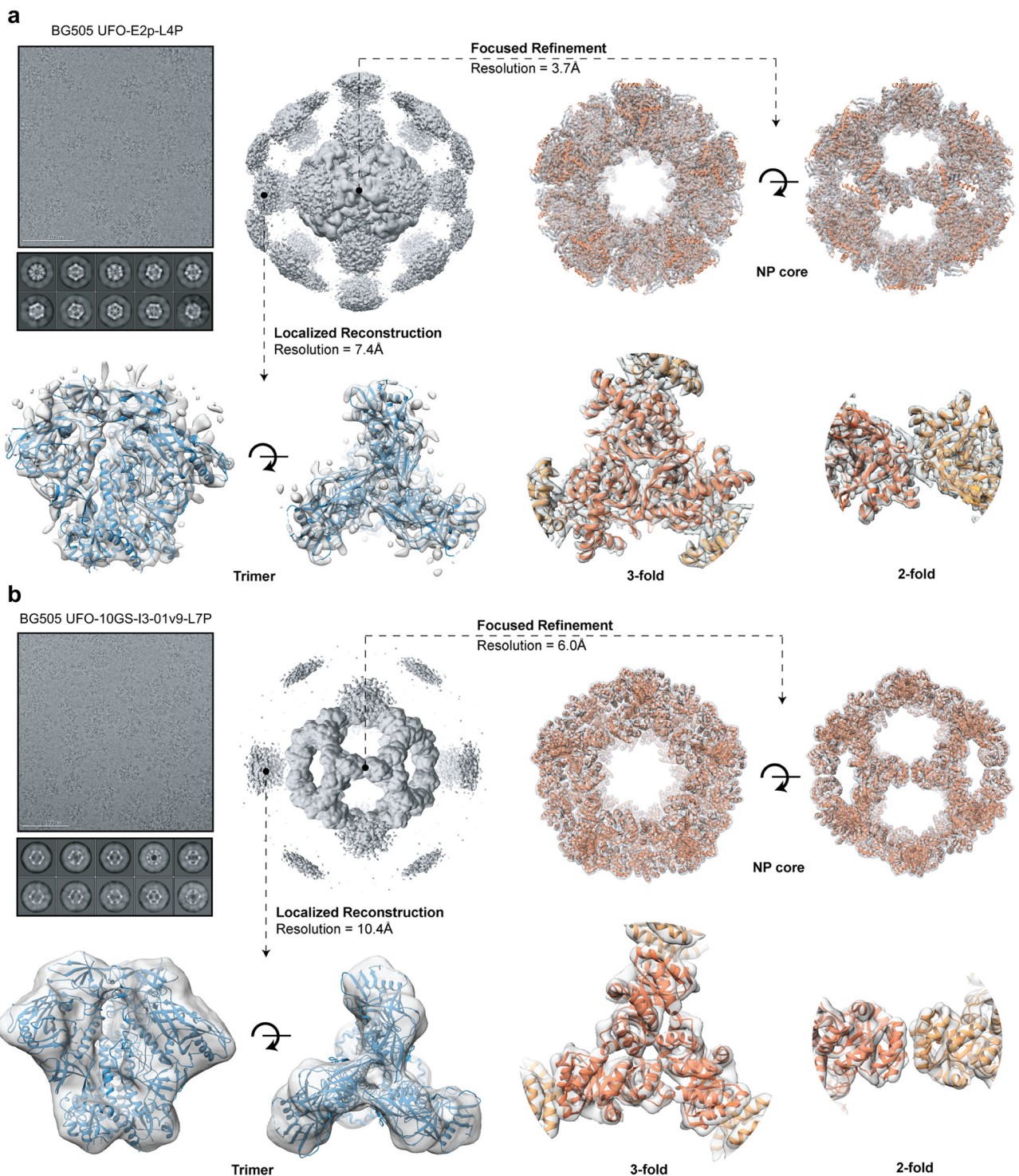

**Fig. 2 | Cryo-EM analysis of the multilayered BG505 UFO trimer-presenting nanoparticles.** Two PGT145/SEC-purified SApNPs were analyzed. **a** BG505 UFO-E2p-L4P. **b** UFO-10GS-I3-01v9-L7P. Representative raw cryo-EM micrographs, 2D classes and 3D maps featuring full SApNPs (with trimer antigens) are displayed in the top left part of the corresponding panels. Focused classification and refinement were applied to reconstruct a 3D map of the SApNP core (top and bottom right), while the localized reconstruction approach was used for extraction and analysis of trimer subparticles (bottom left). The maps are represented as transparent gray mesh and the corresponding models are shown as ribbons (SApNPs: orange; BG505 UFO trimer: blue). The refined model is presented for the E2p NP backbone, while the I3-01v9 and trimer maps are fitted with previously reported models (PDB ID: 5KP9 and 6UTK).

(IMP) around N332 consists of predominantly Man$_9$GlcNAc$_2$ glycans[27,38]. This pattern was observed for all samples tested, demonstrating that the presentation of this glycan supersite is unaffected by multivalent NP display. Complex-type glycans can be found at gp120 N88, N185e, N301/N355 (although high-mannose predominates), N398,

N462, and across the N611, N618, and N637 glycans on gp41 (Fig. S3a)[27,38]. Several potential *N*-linked glycosylation sites (PNGS) lack an attached glycan at a proportion of the sites. For example, N133, N137, N185e, and N197 on gp120 are underoccupied, similar to the BG505 SOSIP trimer[50]. Treatment with swainsonine did not enrich

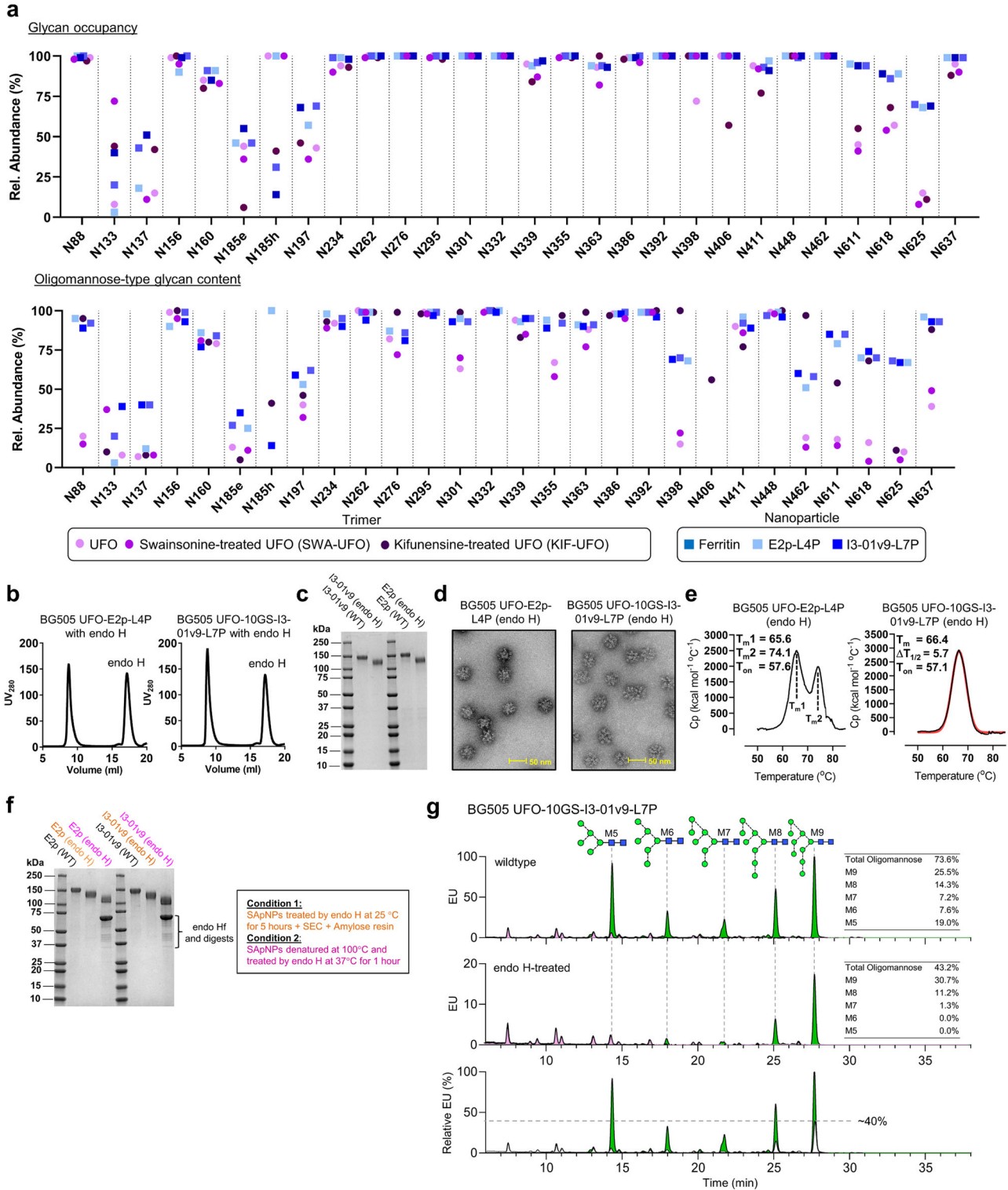

**Fig. 3 | Glycan analysis of BG505 Env immunogens with wildtype and modified glycans. a** Comparison of site-specific occupancy and oligomannose-type glycan content for the BG505 UFO trimer (wildtype, swainsonine, and kifunensine) and three SApNPs (wildtype). The plotted data represent the mean of three analytical repeats, with trimers shown as circles and NPs as squares. The oligomannose content was determined by adding all glycan compositions containing HexNAc(2). **b** SEC profiles of BG505 UFO-E2p-L4P and UFO-10GS-I3-01v9-L7P SApNPs upon endo H treatment. **c** Reducing SDS-PAGE of BG505 UFO-E2p-L4P and UFO-10GS-I3-01v9-L7P SApNPs before and after glycan trimming by endo H. **d** EM images of glycan-trimmed BG505 UFO-E2p-L4P and UFO-10GS-I3-01v9-L7P SApNPs. **e** Thermostability of glycan-trimmed BG505 UFO-E2p-L4P and UFO-10GS-I3-01v9-L7P SApNPs, with $T_m$, $\Delta T_{1/2}$, and $T_{on}$ measured by DSC. **f** Reducing SDS-PAGE of

BG505 UFO-E2p-L4P and UFO-10GS-I3-01v9-L7P SApNPs before and after endo H treatment under two different conditions: 25 °C for 5 h (condition 1) and protein denaturation at 100 °C followed by treatment at 37 °C for 1 h (condition 2). **g** UPLC analysis of released glycans from BG505 UFO-10GS-I3-01v9 SApNP. Top: natively glycosylated; Middle: endo H-treated; Bottom: overlay of two UPLC traces with the treated trace (from the middle panel) recolored in white. The endo H-treated trace has been normalized to the untreated trace using the complex-type glycan peak eluting at -8 min. In top and middle panels: oligomannose-type and complex-type glycans are colored in green and pink, respectively. In the top panel, oligomannose-type glycan peaks are labeled with their representative structures. For both UPLC traces, the percentage of each oligomannose-type is listed on the right.

hybrid-type glycans as expected from glycosidase inhibition, with a nearly identical glycan profile to the wildtype trimer (Fig. 3a and Fig. S3b). Treatment with kifunensine produced predominantly Man$_9$GlcNAc$_2$ glycans at most sites, except for certain sites that were still occupied by Man$_{5-8}$GlcNAc$_2$ due to mannosidase trimming (Fig. 3a and Fig. S3c).

BG505 UFO trimer-presenting SApNPs behaved similarly in terms of glycan processing and occupancy (Fig. 3a and Fig. S3d–f). Moderate differences were noted for N133 and N137, where FR is less occupied than E2p and I3-01v9, whereas the same FR sample is more occupied at N185h compared to all other samples. However, SApNPs showed markedly different glycan profiles from soluble trimers. Variations in glycan occupancy were noted for N133, N137, N185e, N185h, and N197 on gp120. Trimers were less occupied than SApNPs at all four glycan sites on gp41. This pattern was most pronounced for N625, showing a ~50% increase in occupancy for three SApNPs. Since glycan occupancy normally decreases toward the C terminus[106], a possible explanation is that the NP subunit in effect shifts the relative position of gp41 and becomes the new C terminus of the Env-NP fusion protein. Nevertheless, the increased glycan occupancy on gp41 will likely be beneficial and diminish the elicitation or binding of nNAbs to the gp41 base and perhaps the NP backbone. The processing of oligomannose-type glycans also differs between soluble trimers and NPs, especially for sites outside the mannose patches. Two gp120 sites, N88 and N462, which present complex-type glycans on soluble trimers, are less processed on NPs (e.g., N88 shifts from 25% to 90% oligomannose-type glycans). Restricted glycan processing was observed for the gp41 sites located proximal to the base of the NP-displayed Envs; the glycan presented at these gp41 sites is Man$_5$GlcNAc$_2$, not unprocessed Man$_9$GlcNAc$_2$ or a complex-type glycan.

Endoglycosidase H (endo H) cleaves the chitobiose core of oligomannose- and hybrid-type glycans and leaves a single GlcNAc residue attached to the asparagine. Endo H has been used in the glycan analysis of diverse HIV-1 Envs[39]. Here, PGT145/SEC-purified E2p and I3-01v9 SApNPs were treated by endo Hf (a fusion of endo H and maltose-binding protein [MBP]) at 25 °C for 5 h and purified again by SEC (Fig. 3b). Owing to the significant difference in their molecular weights, endo Hf appears as a well-separated peak in the SEC profile. However, a rapid enzyme-linked immunosorbent assay (ELISA) using an MBP-specific mouse antibody detected residual endo Hf in SEC-purified SApNPs (Fig. S3g). Reducing SDS-PAGE showed lower bands for E2p and I3-01v9 SApNPs after endo H treatment (Fig. 3c), with the structural integrity of glycan-trimmed SApNPs confirmed by negative-stain EM (Fig. 3d). Comparable $T_m$ values from DSC suggest that glycan trimming has little effect on the thermostability of SApNPs (Fig. 3e). Next, reducing SDS-PAGE was used to analyze E2p and I3-01v9 SApNPs treated with endo Hf followed by subsequent enzyme removal using the MBP-specific resin (see Methods). Notably, endo H treatment was performed either at 25 °C for 5 h or, alternatively, at 37 °C for 1 h after fully denaturing the SApNPs at 100 °C (Fig. 3f). The denatured material showed the lowest band on the gel, suggesting that some glycans may be protected from endo H cleavage by the Env structure. Lastly, we used ultra-high performance liquid chromatography (UPLC) with in-gel digestion to determine the global glycosylation status for I3-01v9 SApNP before and after endo H treatment with subsequent enzyme removal using the MBP-specific resin (Fig. 3g). While Man$_{5-7}$GlcNAc$_2$ glycans appeared to be more susceptible to endo H, some Man$_{8-9}$GlcNAc$_2$ glycans remained intact after treatment, with a Man$_9$GlcNAc$_2$ signal of ~40% relative to the untreated sample.

## Antigenicity of BG505 UFO immunogens with wildtype and modified glycans

Native-like Env trimers of diverse origins that contain the modified HR1$_N$ bend or full UFO/UFO-BG designs have been assessed against panels of bNAbs and nNAbs[26–28]. Here, BG505 UFO trimers and SApNPs

with wildtype and modified glycans were tested for antibody binding using bio-layer interferometry (BLI). The previously established antibody panel[27] was expanded to include human bNAbs 438-B11, VRC34, and SF12, which target the N332 supersite[98], FP[107], and SF[108], respectively, a tier 2 mouse NAb (M4H2K1) that binds to the C3/V4 region[71], and an NHP-derived nNAb (RM20A3) that recognizes the trimer base[109].

The peak antibody-binding signals were summarized according to immunogen valency and glycan treatment (Fig. 4a), with association and dissociation curves plotted for six concentrations (Fig. S4a–o). Analytical sensors were used for soluble trimers (Fig. 4a, top). Overall, the wildtype BG505 UFO Env preferentially bound to (b)NAbs and not nNAbs, except for V3 tip-directed 19b and 447-52D and base-directed RM20A3 (Fig. S4a). Kifunensine treatment reduced Env binding to (b) NAbs that either interact with or require the accommodation of glycans (Fig. S4b). PGT151, which interacts with complex-type glycans at N611 and N637[47,110–112], showed undetectable binding to the BG505 UFO Env with unprocessed glycans at these two sites. VRC01 binding was made more difficult by the modified glycan barrier around the CD4bs due to the inhibition of ER α mannosidase I in CHO cells. Endo H treatment resulted in an overall reduction in Env binding to glycan-reactive (b)NAbs (Fig. S4c). Compared with the apex bNAbs, glycan trimming had a less severe impact on bNAbs targeting the N332 supersite (e.g., PGT121 and PGT128), which interact with the GDIR motif and high-mannose glycans at N332 and N301[44,48,113]. PGT151 barely recognized the glycan-trimmed BG505 Env. Endo H treatment largely retained accessibility of the CD4bs to bNAb VRC01[114,115] upon trimming of the surrounding glycan barrier. Notably, glycan modification had little effect on an SF-directed bNAb (SF12) that interacts with N262, N295, and N448[108], as well as the base-directed nNAb, RM20A3[109].

Antigenicity was then determined for three SApNPs displaying wildtype and modified Env glycans, with respective trimers included as a control (Fig. 4a, bottom). Quantitation sensors were used in BLI assays to measure the avidity effect of SApNPs[91,92]. Compared with soluble trimer, wildtype SApNPs (Fig. S4d–g) bound at elevated levels to (b)NAbs that target the V2 apex (×2.5–4.7), N332 supersite (×2.0–3.9), and CD4bs (×1.4–1.6) and at reduced levels to (b)NAbs that recognize the FP (×0.4–0.6) and gp120/gp41 interface (×0.2–0.7). When tested against nNAbs, enhanced binding was only observed for the V3 tip (×2.6–4.0) but not the non-neutralizing CD4bs, CD4-induced (CD4i), gp41, and base epitopes. Therefore, particulate display can exert a positive influence, as well as structural constraints, on antibody recognition of an Env epitope depending on the location and accessibility of this epitope on the particle surface[27,88,101,102]. Notably, the wildtype SApNPs only displayed a moderate increase in VRC01 binding with a slow on-rate, suggesting the CD4bs is more occluded compared to the V2 apex and glycan-V3 supersite on the NP surface. SApNPs bearing unprocessed Man$_9$GlcNAc$_2$ glycans exhibited antigenic profiles comparable to wildtype SApNPs (Fig. S4h–k). Similar to the trimer-presenting unprocessed glycans, no PGT151 binding was observed. A similar but less severe effect was noted for VRC34, which interacts with the N terminus of FP and the complex-type N88 glycan[107]. Compared with wildtype SApNPs, endo H-treated SApNPs (Fig. S4l–o) demonstrated unique antigenic profiles with marked improvement in VRC01 binding (×2.0–2.3). Slightly lower binding signals were observed for most glycan-specific NAbs and bNAbs except for two. PG16 binds to a high-mannose glycan at N160 and a hybrid-type glycan at N173[41]. The wildtype SApNPs bound to PG16 with higher signals (×3.4–4.6) than the trimer. In comparison, glycan trimming reduced trimer binding to PG16 but slightly increased SApNP binding, creating a larger fold difference between the trimer and SApNPs. PGT135 requires ManGlcNAc$_2$ at N386, Man$_6$GlcNAc$_2$ at N332, and Man$_8$GlcNAc$_2$ at N392 in Env recognition[116]. The glycan-trimmed SApNPs showed higher PGT135-binding signals than the wildtype SApNPs, 2.3–2.8 and 2.0–2.4-fold,

**a**

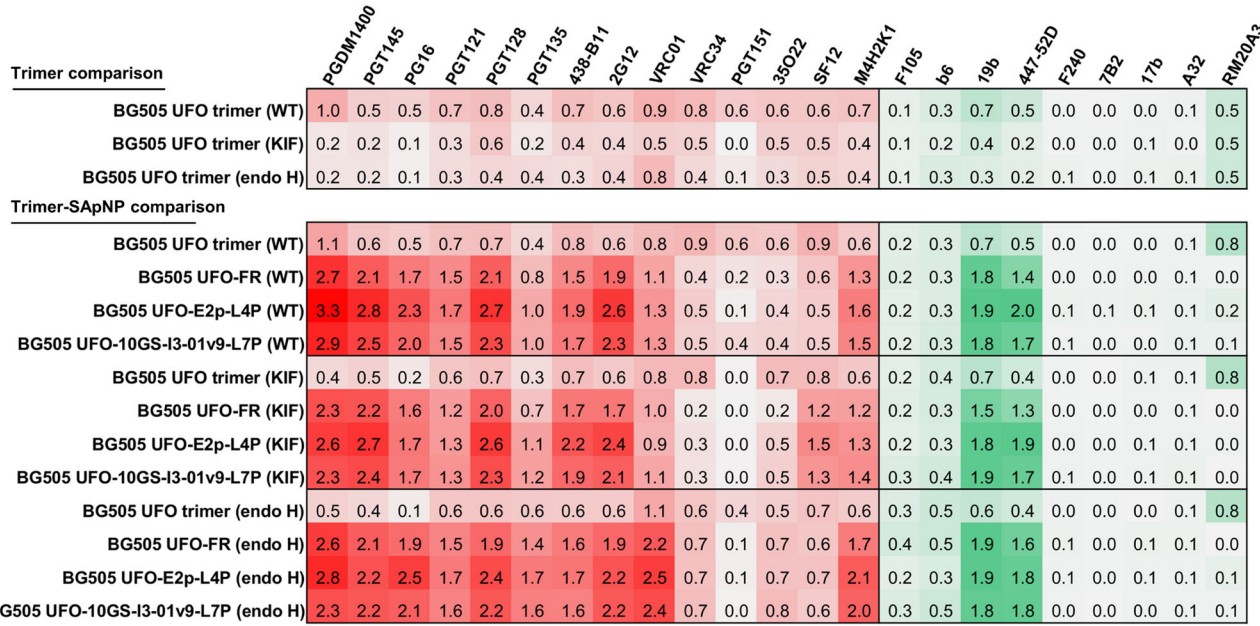

| | PGDM1400 | PGT145 | PG16 | PGT121 | PGT128 | PGT135 | 438-B11 | 2G12 | VRC01 | VRC34 | PGT151 | 35O22 | SF12 | M4H2K1 | F105 | b6 | 19b | 447-52D | F240 | 7B2 | 17b | A32 | RM20A3 |
|---|---|---|---|---|---|---|---|---|---|---|---|---|---|---|---|---|---|---|---|---|---|---|---|
| **Trimer comparison** | | | | | | | | | | | | | | | | | | | | | | | |
| BG505 UFO trimer (WT) | 1.0 | 0.5 | 0.5 | 0.7 | 0.8 | 0.4 | 0.7 | 0.6 | 0.9 | 0.8 | 0.6 | 0.6 | 0.6 | 0.7 | 0.1 | 0.3 | 0.7 | 0.5 | 0.0 | 0.0 | 0.0 | 0.1 | 0.5 |
| BG505 UFO trimer (KIF) | 0.2 | 0.2 | 0.1 | 0.3 | 0.6 | 0.2 | 0.4 | 0.4 | 0.5 | 0.5 | 0.0 | 0.5 | 0.5 | 0.4 | 0.1 | 0.2 | 0.4 | 0.2 | 0.0 | 0.0 | 0.0 | 0.0 | 0.5 |
| BG505 UFO trimer (endo H) | 0.2 | 0.2 | 0.1 | 0.3 | 0.4 | 0.4 | 0.3 | 0.4 | 0.8 | 0.4 | 0.1 | 0.3 | 0.5 | 0.4 | 0.1 | 0.3 | 0.3 | 0.2 | 0.1 | 0.0 | 0.1 | 0.1 | 0.5 |
| **Trimer-SApNP comparison** | | | | | | | | | | | | | | | | | | | | | | | |
| BG505 UFO trimer (WT) | 1.1 | 0.6 | 0.5 | 0.7 | 0.7 | 0.4 | 0.8 | 0.6 | 0.8 | 0.9 | 0.6 | 0.6 | 0.9 | 0.6 | 0.2 | 0.3 | 0.7 | 0.5 | 0.0 | 0.0 | 0.0 | 0.1 | 0.8 |
| BG505 UFO-FR (WT) | 2.7 | 2.1 | 1.7 | 1.5 | 2.1 | 0.8 | 1.5 | 1.9 | 1.1 | 0.4 | 0.2 | 0.3 | 0.6 | 1.3 | 0.2 | 0.3 | 1.8 | 1.4 | 0.0 | 0.0 | 0.0 | 0.1 | 0.0 |
| BG505 UFO-E2p-L4P (WT) | 3.3 | 2.8 | 2.3 | 1.7 | 2.7 | 1.0 | 1.9 | 2.6 | 1.3 | 0.5 | 0.1 | 0.4 | 0.5 | 1.6 | 0.2 | 0.3 | 1.9 | 2.0 | 0.1 | 0.1 | 0.0 | 0.1 | 0.2 |
| BG505 UFO-10GS-I3-01v9-L7P (WT) | 2.9 | 2.5 | 2.0 | 1.5 | 2.3 | 1.0 | 1.7 | 2.3 | 1.3 | 0.5 | 0.4 | 0.4 | 0.5 | 1.5 | 0.2 | 0.3 | 1.8 | 1.7 | 0.1 | 0.0 | 0.0 | 0.1 | 0.1 |
| BG505 UFO trimer (KIF) | 0.4 | 0.5 | 0.2 | 0.6 | 0.7 | 0.3 | 0.7 | 0.6 | 0.8 | 0.8 | 0.0 | 0.7 | 0.8 | 0.6 | 0.2 | 0.4 | 0.7 | 0.4 | 0.0 | 0.0 | 0.1 | 0.1 | 0.8 |
| BG505 UFO-FR (KIF) | 2.3 | 2.2 | 1.6 | 1.2 | 2.0 | 0.7 | 1.7 | 1.7 | 1.0 | 0.2 | 0.0 | 0.2 | 1.2 | 1.2 | 0.2 | 0.3 | 1.5 | 1.3 | 0.0 | 0.0 | 0.1 | 0.0 | 0.0 |
| BG505 UFO-E2p-L4P (KIF) | 2.6 | 2.7 | 1.7 | 1.3 | 2.6 | 1.1 | 2.2 | 2.4 | 0.9 | 0.3 | 0.0 | 0.5 | 1.5 | 1.3 | 0.2 | 0.3 | 1.8 | 1.9 | 0.1 | 0.0 | 0.0 | 0.1 | 0.0 |
| BG505 UFO-10GS-I3-01v9-L7P (KIF) | 2.3 | 2.4 | 1.7 | 1.3 | 2.3 | 1.2 | 1.9 | 2.1 | 1.1 | 0.3 | 0.0 | 0.5 | 1.3 | 1.4 | 0.3 | 0.4 | 1.9 | 1.7 | 0.1 | 0.0 | 0.0 | 0.1 | 0.1 |
| BG505 UFO trimer (endo H) | 0.5 | 0.4 | 0.1 | 0.6 | 0.6 | 0.6 | 0.6 | 0.6 | 1.1 | 0.6 | 0.4 | 0.5 | 0.7 | 0.6 | 0.3 | 0.5 | 0.6 | 0.4 | 0.0 | 0.0 | 0.0 | 0.1 | 0.8 |
| BG505 UFO-FR (endo H) | 2.6 | 2.1 | 1.9 | 1.5 | 1.9 | 1.4 | 1.6 | 1.9 | 2.2 | 0.7 | 0.1 | 0.7 | 0.6 | 1.7 | 0.4 | 0.5 | 1.9 | 1.6 | 0.1 | 0.0 | 0.1 | 0.1 | 0.0 |
| BG505 UFO-E2p-L4P (endo H) | 2.8 | 2.2 | 2.5 | 1.7 | 2.4 | 1.7 | 1.7 | 2.2 | 2.5 | 0.7 | 0.1 | 0.7 | 0.7 | 2.1 | 0.2 | 0.3 | 1.9 | 1.8 | 0.1 | 0.0 | 0.1 | 0.1 | 0.1 |
| BG505 UFO-10GS-I3-01v9-L7P (endo H) | 2.3 | 2.2 | 2.1 | 1.6 | 2.2 | 1.6 | 1.6 | 2.2 | 2.4 | 0.7 | 0.0 | 0.8 | 0.6 | 2.0 | 0.3 | 0.5 | 1.8 | 1.8 | 0.0 | 0.0 | 0.0 | 0.1 | 0.1 |

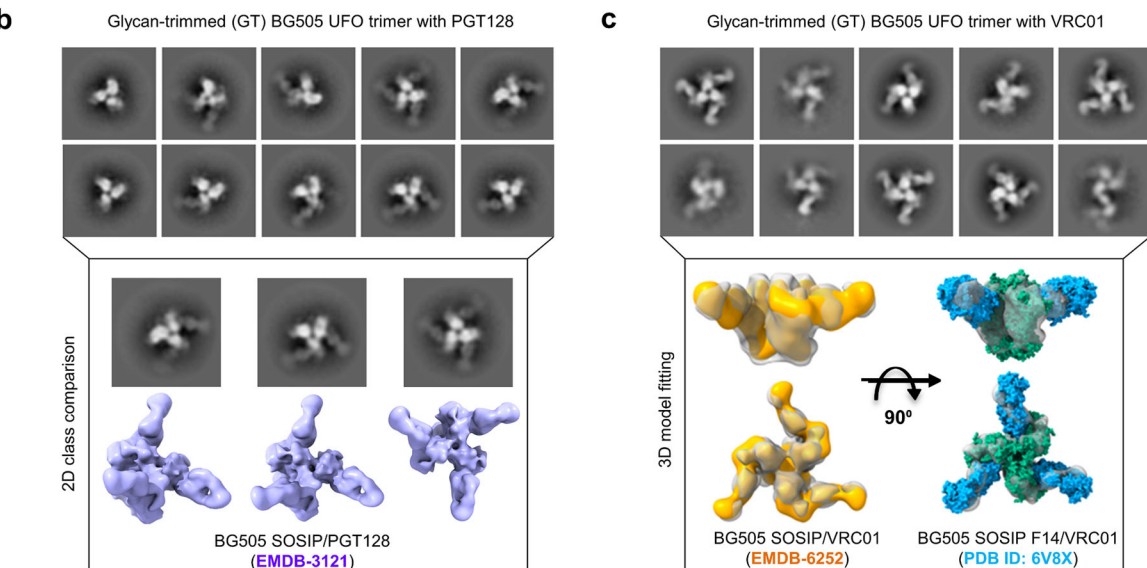

**b** Glycan-trimmed (GT) BG505 UFO trimer with PGT128

BG505 SOSIP/PGT128 (EMDB-3121)

**c** Glycan-trimmed (GT) BG505 UFO trimer with VRC01

BG505 SOSIP/VRC01 (EMDB-6252)  BG505 SOSIP F14/VRC01 (PDB ID: 6V8X)

**Fig. 4 | Antigenicity of BG505 UFO Env immunogens with wildtype and modified glycans. a** Antigenic profiles of BG505 UFO trimer (top) and its three SApNPs based on FR, E2p-L4D, and I3-01v9-L7P (bottom) with different glycan treatments (wildtype, expressed in the presence of kifunensine, and endo H treatment). A total of 14 NAbs/bNAbs and 9 non-NAbs were tested by BLI. Sensorgrams were obtained from an Octet RED96 using an antigen titration series of six concentrations (starting at 266.7 nM for the UFO trimer, 14.9 nM for the FR SApNP, 5.5 nM for the E2p and I3-01v9 SApNPs followed by two-fold dilutions) and are shown in Fig. S4. The peak values at the highest concentration are summarized in the matrix, in which cells are colored in red and green for (b)NAbs and non-NAbs, respectively.

Higher color intensity indicates greater binding signal measured by Octet. **b** Negative-stain EM analysis of the glycan-trimmed BG505 UFO trimer in complex with bNAb PGT128. Examples of 2D class images (top) and comparison with a cryo-EM model of BG505 SOSIP/PGT128 complex (bottom, EMDB-3121, cornflower blue). **c** Negative-stain EM analysis of the glycan-trimmed BG505 UFO trimer in complex with bNAb VRC01. Examples of 2D class images (top) and superposition of the 3D reconstruction onto two previously reported BG505 SOSIP/VRC01 models derived from negative-stain EM (EMDB-6252, orange) and cryo-EM (PDB ID: 6V8X, cyan and green).

respectively, compared to their trimers. PGT151 binding was barely detectable for three SApNPs, but increased binding was observed for another interface bNAb, 35O22[117], and for the FP bNAb, VRC34[107]. Two nNAbs to the V3 tip, 19b and 447-52D, bound to glycan-trimmed SApNPs at comparable levels to wildtype SApNPs. The nNAb RM20A3[109] showed negligible binding to SApNPs independent of their surface glycans, likely because the trimer base would become nearly inaccessible on the NP surface. BG505 UFO trimer and I3-01v9 SApNP,

both wildtype and endo H-treated, were tested against the inferred PGT121 germline (GL) and intermediate (3H109L)[40,118] (Fig. S4p). Glycan trimming showed no effect on Env interactions with these two PGT121 variants. Lastly, wildtype E2p was tested in duplicate to estimate the signal variation, which was within 2.5% for all antibodies with detectable binding except for SF12, which showed a variation of 5.6% (Fig. S4q). Therefore, experimental variation would have had little effect on the patterns observed in the BLI analysis.

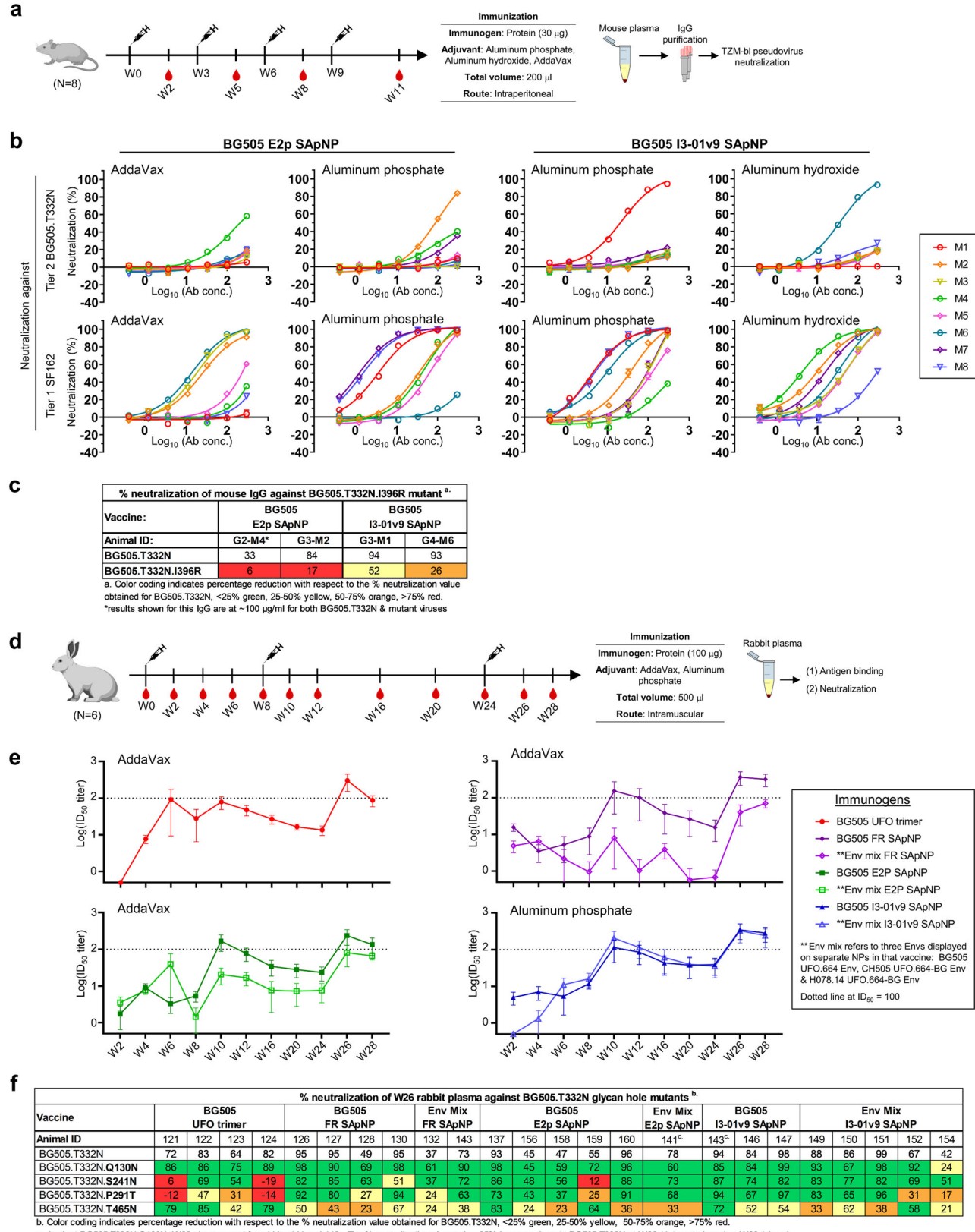

a. Color coding indicates percentage reduction with respect to the % neutralization value obtained for BG505.T332N, <25% green, 25-50% yellow, 50-75% orange, >75% red.
*results shown for this IgG are at ~100 µg/ml for both BG505.T332N & mutant viruses

b. Color coding indicates percentage reduction with respect to the % neutralization value obtained for BG505.T332N, <25% green, 25-50% yellow, 50-75% orange, >75% red.

c. Against BG505.T332N.Q130N: W28 plasma used for rabbits 141 and 143. The % neutralization change is <25% in comparison to BG505.T332N at W28 (data not shown) or W26 (above).

Negative-stain EM was utilized to characterize bNAb interactions with the glycan-trimmed BG505 UFO trimer. PGT128 binds to the V3 base and high-mannose glycans at N295 and N301 or N332[44,48,119]. Previous structures suggest that the mannose moieties of these glycans serve as anchors to orient and stabilize the interaction with PGT128 Fab, which binds the "GDIR" motif with its HCDR2 and HCDR3 tips[44,48].

The removal of mannose groups at these three sites may destabilize PGT128 and affect its angle of approach. Indeed, a single endo H-treated BG505 UFO trimer could bind to one to three PGT128 Fabs, with less resolved density for the bound Fabs in the 2D class averages (Fig. 4b and Fig. S4r). Comparison with the cryo-EM model of the BG505 SOSIP/PGT128 complex (EMDB-3121[48]) confirmed that

**Fig. 5 | Immunogenicity of HIV-1 UFO Env immunogens with wildtype glycans in mice and rabbits. a** Schematic representation of the mouse immunization regimen for two BG505 UFO trimer-presenting E2p and I3-01v9 SApNPs. **b** Neutralization of tier 2 clade A BG505.T332N and tier 1 clade B SF162 by purified mouse IgG from week 11 ($n$ = 8 mice/group). **c** Percent neutralization of week 11 mouse IgG from four vaccine responders against BG505.T332N and its I396R mutant. Vaccine responders are those subjects with $IC_{50}$ < 300 µg/ml in IgG neutralization against BG505.T332N. Color coding indicates the level of reduction in % neutralization relative to BG505.T332N. In both **b** and **c**, the TZM-bl assay was performed in duplicate, starting at an IgG concentration of 300 µg/ml (unless otherwise indicated by an asterisk) followed by three-fold dilutions. Error bars in **b** represent the difference between these duplicate values at each concentration tested for each sample. **d** Schematic representation of the rabbit immunization protocol. Seven groups of rabbits, 6 per group, were immunized to test the BG505 UFO trimer and its FR, E2p, and I3-01v9 SApNPs, with a cocktail group added for each SApNP by mixing equal

amounts of three SApNPs that display BG505 UFO, CH505 UFO-BG, or H078.14 UFO-BG trimers (termed Env mix). **e** Longitudinal analysis of rabbit plasma neutralization between weeks 2 and 28 ($n$ = 6 rabbits/group). All samples were tested in duplicate with a 100-times starting dilution. Average $ID_{50}$ values for each group against BG505.T332N are shown with the SEM. Groups that were immunized with the same vaccine platform are plotted on the same graph: trimer (red), FR (dark/light purple), E2p (dark/light green) and I3-01v9 (dark/light blue). Week 0 data are omitted due to nonspecific background signals. **f** Percentage neutralization of select week 26 rabbit plasma (≥ 35% neutralization against BG505.T332N at the first dilution) against BG505.T332N and four glycan hole mutants. For each mutant virus, rabbit plasma was tested in duplicate, starting at 100-times dilution followed by three-fold dilutions. Results are shown for BG505.T332N from **e** and the mutant viruses at the 100-times dilution. Color coding indicates the level of reduction in % neutralization relative to BG505.T332N. Mouse and rabbit images created with BioRender.com.

PGT128, albeit with greater flexibility, still binds to the same epitope region. VRC01 targets a large proteinaceous area of the CD4bs protected by a ring of glycans with the N276 glycan positioned as a critical barrier to the maturation of VRC01-class bNAbs[114,115,120,121]. Trimming the glycan shield can, in principle, improve the CD4bs recognition by VRC01-class bNAbs. Indeed, we obtained a 12.2 Å-resolution 3D model of the BG505 UFO/VRC01 complex, which superimposes well with the previous EM models of BG505 SOSIP/VRC01 complexes[122,123] (Fig. 4c and Fig. S4r).

BLI demonstrated a cross-panel improvement in bNAb binding to SApNPs compared with soluble trimers, consistent with our previous studies[27,89,91,92]. Glycan trimming retained bNAb binding to major glycan epitopes on the NP-displayed Env and improved recognition of the CD4bs by VRC01-class bNAbs. Negative-stain EM confirmed that bNAbs PGT128 and VRC01 could still recognize their epitopes on the glycan-trimmed Env trimer. Altogether, glycan-trimmed SApNPs may be more effective immunogens due to more balanced epitope accessibility.

## Assessment of HIV-1 UFO Env immunogens with wildtype glycans in mice and rabbits

While BG505 SOSIP and $HR1_N$-redesigned trimers were ineffective at NAb induction in mice[27,55], a reengineered I3-01 NP displaying the $HR1_N$-redesigned trimer elicited tier 2 murine NAbs to the C3/V4 epitope[27,71]. Recently, an autologous tier 2 NAb response was induced in mice by iron oxide NP-attached SOSIP trimers[96] and a DNA trimer vaccine[70] using long-boost intervals. In rabbits, Env-induced tier 2 NAb responses mainly target glycan holes[56].

We first assessed wildtype BG505 UFO immunogens formulated with three commonly used adjuvants in mice. We adopted an immunization protocol used in our previous vaccine studies[27,89,91,92,124] (Fig. 5a). In brief, groups of eight mice were immunized four times at 3-week intervals. SApNPs (30 µg/dose) were formulated with AddaVax (AV), aluminum phosphate (AP), and aluminum hydroxide (AH). A non-adjuvanted group and a heterologous AV × 2/AP × 2 group were included as a negative control and for comparison, respectively. Immunoglobulin G (IgG) was purified from mouse samples to reduce nonspecific antiviral activity in neutralization assays[27,55]. We performed TZM-bl neutralization assays using purified mouse IgG from the last time point, week 11 (Fig. 5b and Fig. S5a). Without adjuvant, both SApNPs failed to elicit tier 2 NAb responses to BG505.T332N, but I3-01v9 was more effective against a tier 1 clade B virus, SF162 (Fig. S5a, middle). E2p mixed with AV or AP and I3-01v9 mixed with AP or AH each showed tier 2 NAb induction in one of the eight mice (Fig. 5b and Fig. S5a, top). However, switching the adjuvant during the immunization proved ineffective. Adjuvanted SApNPs induced a robust tier 1 NAb response (Fig. 5b and Fig. S5a, middle). When tested against a BG505 variant with a C3/V4 epitope knockout mutation, I396R (Fig. 5c and Fig. S5b), reduced neutralization (by 45-82%) was noted for all four

mice that generated tier 2 NAb responses, suggesting the presence of C3/V4-directed NAbs[71] in serum. The remaining neutralizing activity may be attributed to C3/V5[70] and other Env epitopes. As a negative control, purified IgG from three tier 2 positive mice was tested against pseudoparticles (pps) bearing the murine leukemia virus (MLV) Env, MLV-pps, and did not show any neutralization (Fig. S5a, bottom). Borderline neutralization of clade A 398F1 was observed in the TZM-bl assays against a 12-virus global panel (Fig. S5c).

We then tested an extended set of wildtype UFO immunogens in rabbits. Briefly, groups of six rabbits were immunized three times at weeks 0, 8, and 24 with blood draws at 12 timepoints throughout immunization (Fig. 5d). All antigens (100 µg/dose) were mixed with AV except for I3-01v9 SApNPs, which were paired with AP. While the BG505 UFO trimer and its SApNPs (FR, E2p, and I3-01v9) were the focus of this study, a cocktail mixing equal amounts (33 µg/dose) of SApNPs that present clade A BG505 UFO, clade B H078.14 UFO-BG, and clade C CH505 UFO-BG trimers was included for each SApNP platform, resulting in seven groups in total. Longitudinal NAb responses to tier 2 BG505.T332N were examined in rabbit plasma and measured with 50% inhibitory dilution ($ID_{50}$) titers (Fig. 5e). While the overall kinetics were similar across all groups, the cocktail groups exhibited an NP platform-dependent pattern in their NAb responses to BG505.T332N. Reducing the BG505 dose to one-third for each cocktail formulation had negative and negligible effects for FR and I3-01v9, respectively. The tier 2 NAb response became detectable at week 10 (Fig. 5e, above dotted line), with $ID_{50}$ values for almost all vaccine groups peaking at week 26 post immunization (Fig. 5e, Fig. S5d). Of interest, when testing tier 1 clade B SF162, a detectable NAb response was observed as early as week 4 for the two I3-01v9 groups (Fig. 5e). Rabbit plasma at weeks 0, 2, 10, and 26 displayed no MLV-pp reactivity (Fig. S5f). We then investigated the prevalence of glycan hole NAbs in polyclonal plasma using four BG505.T332N mutants, Q130N, S241N, P291T, and T465N (Fig. 5f). Filling glycan holes partially depleted neutralizing activity, consistent with our previous study[71]. The trimer group exhibited a more visible reduction against the 241/289 glycan mutants, whereas all SApNP groups appeared to become more sensitive to the T465N mutation. Finally, all SApNP formulations failed to generate a strong cross-clade NAb response to other tier 2 isolates in a 12-virus global panel, with no advantage in breadth seen for the Env mixture vaccines (Fig. S5g).

## Assessment of BG505 UFO Env immunogens with trimmed glycans in mice and rabbits

We investigated how glycan trimming affects Env-elicited NAb responses in mice and rabbits by testing a homologous regimen ("regimen 1") with glycan-trimmed immunogens in all doses and a heterologous prime-boost regimen ("regimen 2") with glycan-trimmed and wildtype immunogens in doses 1–2 and 3–4, respectively (Fig. 6a). The ExpiCHO-produced BG505 UFO trimer was

**a**

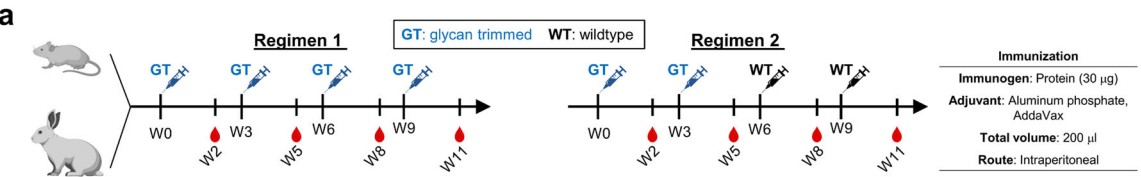

**b** Neutralization against tier 2 BG505.T332N by purified mouse IgG from week 11

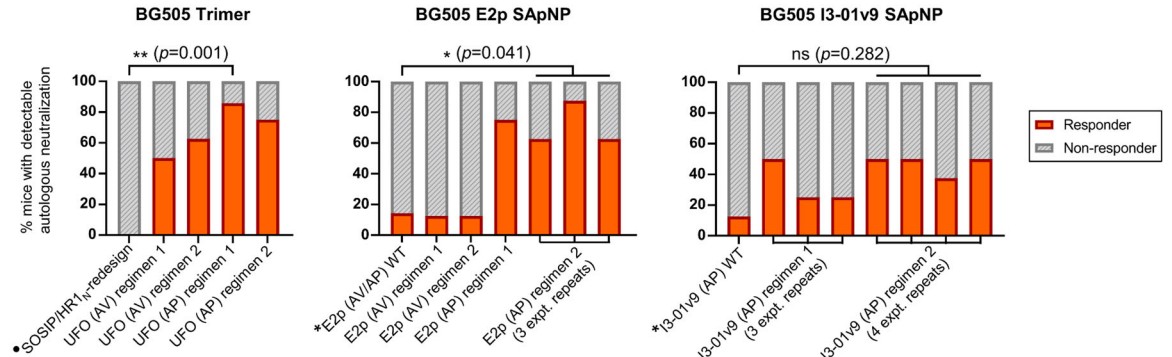

**c**

| | % neutralization of mouse IgG against BG505.T332N.I396R mutant [a.] | | | | | | | | | | | | | | | | | | | |
|---|---|---|---|---|---|---|---|---|---|---|---|---|---|---|---|---|---|---|---|---|
| Vaccine | BG505 UFO trimer (Aluminum phosphate) | | | | | | | | | | | | | | | BG505 E2p SApNP (AddaVax) | | | |
| Glycan trimming regimen | Regimen 1 | | | | | | | Regimen 2 | | | | | | | | Regimen 1 | | Regimen 2 | |
| Animal ID | G3-M1 | G3-M2 | G3-M4 | G3-M5 | G3-M6 | G3-M7 | G3-M8 | G4-M1 | G4-M2 | G4-M3 | G4-M4 | G4-M5 | G4-M6 | G4-M7 | G4-M8 | G1-M2 | G1-M3 | G1-M5 | G2-M4 | G2-M8 |
| BG505.T332N | 95 | 51 | 76 | 98 | 97 | 56 | 37 | 70 | 86 | 97 | 68 | 95 | 33 | 38 | 84 | 51 | 32 | 90 | 32 | 86 |
| BG505.T332N. I396R | 37 | 28 | 63 | 28 | 29 | 46 | 29 | 32 | 24 | 30 | 34 | 33 | 31 | 42 | 42 | 89 | 36 | 27 | 13 | 95 |

| | BG505 E2p SApNP (Aluminum phosphate) | | | | | | | | | | | | | | BG505 I3-01v9 SApNP (Aluminum phosphate) | | | | |
|---|---|---|---|---|---|---|---|---|---|---|---|---|---|---|---|---|---|---|---|---|
| Vaccine | | | | | | | | | | | | | | | | | | | | |
| Glycan trimming regimen | Regimen 1 | | | | | | | Regimen 2 | | | | | | | Regimen 1 | | | Regimen 2 | |
| Animal ID | G1-M1 | G1-M2 | G1-M3 | G1-M4 | G1-M5 | G1-M6 | G1-M7 | G2-M2 | G2-M3 | G2-M4 | G2-M6 | G2-M7 | G2-M8 | | G3-M1 | G3-M3 | G3-M8 | G4-M2 | G4-M3 | G4-M4 | G4-M5 |
| BG505.T332N | 49 | 71 | 73 | 71 | 93 | 39 | 89 | 92 | 98 | 60 | 88 | 91 | 38 | | 70 | 49 | 91 | 93 | 82 | 91 | 87 |
| BG505.T332N. I396R | 40 | 71 | 87 | 80 | 86 | 39 | 30 | 89 | 58 | 62 | 92 | 41 | 37 | | 66 | 43 | 43 | 98 | 94 | 96 | 46 |

a. Color coding indicates percentage reduction with respect to the % neutralization value obtained for BG505.T332N, <25% green, 25-50% yellow, 50-75% orange, >75% red.

**d** Neutralization against tier 2 BG505.T332N by purified rabbit IgG from week 11

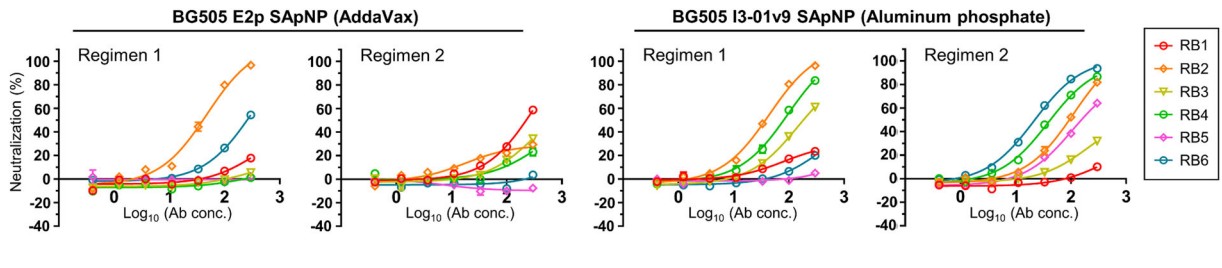

**e**

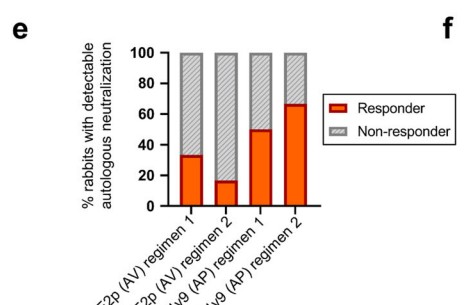

**f**

| | % neutralization of rabbit IgG against BG505.T332N glycan hole mutants [b.] | | | | | | | | | | | |
|---|---|---|---|---|---|---|---|---|---|---|---|---|
| Vaccine | BG505 E2p SApNP | | | | BG505 I3-01v9 SApNP | | | | | | | |
| Glycan trimming regimen | Regimen 1 | | Regimen 2 | | Regimen 1 | | | Regimen 2 | | | | |
| Animal ID | G1-2 | G1-6 | G2-1 | G2-3 | G3-2 | G3-3 | G3-4 | G4-2 | G4-3 | G4-4 | G4-5 | G4-6 |
| BG505.T332N | 97 | 54 | 59 | 34 | 96 | 61 | 84 | 82 | 32 | 87 | 64 | 94 |
| BG505.T332N.**Q130N** | 97 | 53 | 69 | 38 | 97 | 58 | 90 | 87 | 5 | 91 | 77 | 96 |
| BG505.T332N.**S241N** | 99 | 2 | 89 | 26 | 100 | 32 | 96 | 90 | 71 | 91 | 62 | 98 |
| BG505.T332N.**P291T** | 96 | 2 | 74 | 7 | 97 | 11 | 88 | 86 | 53 | 88 | 59 | 95 |
| BG505.T332N.**T465N** | 16 | 59 | 16 | 39 | 36 | 59 | 26 | 65 | 54 | 39 | 49 | 95 |

b. Color coding indicates percentage reduction with respect to the % neutralization value obtained for BG505.T332N, <25% green, 25-50% yellow, 50-75% orange, >75% red.

treated with endo Hf and purified by SEC and an additional resin step (see Methods). Eight and five CHO-K1 cell clones were developed to stably express BG505 UFO trimer-presenting E2p and I3-01v9 SApNPs, respectively. The SApNPs produced by individual cell clones were inspected by negative-stain EM (Fig. S6a) and then pooled for glycan trimming by endo H.

In the mouse study, groups of eight BALB/c mice were immunized four times, at 3-week intervals. BG505 UFO trimer and SApNPs (30 μg/dose) were formulated with AV or AP adjuvants, and the immunization experiment was repeated for some formulation/regimen groups to confirm their immunogenicity. The frequency of vaccine responders (FVR, the ratio of subjects that generate an autologous tier 2 NAb

**Fig. 6 | Immunogenicity of BG505 UFO Env immunogens with trimmed glycans in mice and rabbits. a** Schematic representation of mouse and rabbit immunization regimen 1, where glycan-trimmed immunogens were used throughout, and regimen 2, where glycan-trimmed and wildtype immunogens were used as prime (doses 1 and 2) and boost (doses 3 and 4), respectively. A 3-week interval was used to facilitate the rapid evaluation of various vaccine formulations. **b** Frequency of vaccine responders (FVR) calculated for groups in which mice were immunized with the BG505 UFO trimer (left), E2p SApNP (middle), and I3-01v9 SApNP (right). The first column of each panel represents the all-wildtype regimens as control, with the data either derived from the previous publications (from refs. 27,55, marked with a solid circle) or generated in the current study (Fig. 5b, marked with asterisks). Each experimental repeat (expt. repeat) indicates a different immunization experiment using the same protocol ($n = 8$ mice/group). Purified mouse IgGs from week 11 were tested against tier 2 clade A BG505.T332N. A vaccine responder is defined as a subject with $IC_{50} \leq 300$ μg/ml. No IgG was available for one mouse in the UFO trimer/AP regimen 1 group, two mice in the I3-01v9/AP regimen 1 (1st repeat) group, and one mouse each in the wildtype E2p/AP and E2p/AV comparator groups.

For each vaccine platform (trimer or SApNP), statistical comparison was made between the wildtype group and the glycan-trimmed group(s) with the highest FVR using a two-sided Fisher's exact test. **c** Percent neutralization values of week 11 mouse IgG from select mouse samples (≥ 30% autologous neutralization at the highest IgG concentration) against BG505.T332N and its I396R mutant. **d** Neutralization of tier 2 clade A BG505.T332N by purified rabbit IgG from week 11 ($n = 6$ rabbits/group). **e** FVR of four rabbit groups based on the week 11 autologous neutralization data, with vaccine responders defined as in **b**. **f** Percentage neutralization of select week 11 rabbit IgG samples (≥30% autologous neutralization at the highest IgG concentration) against BG505.T332N and its four glycan hole mutants. All mouse and rabbit TZM-bl assays were done in duplicate at a starting concentration of 300 μg/ml, followed by three-fold dilutions. Error bars in the neutralization curves in **d** represent the difference between the duplicate values for each sample at each concentration tested. Color coding in **c** and **f** indicates the level of reduction in % neutralization relative to BG505.T332N. Mouse and rabbit images created with BioRender.com.

---

response) was calculated for comparison. Overall, glycan trimming (GT) by endo H substantially improved the FVR for the UFO trimer (FVR up to 86%; Fig. 6b, left), E2p SApNP (up to 88%; Fig. 6b, middle), and I3-01v9 SApNP (up to 50%; Fig. 6b, right), in addition to the improved $IC_{50}$ values and less within-group variation compared with the equivalent formulations using wildtype immunogens (Fig. S6b–d). The use of AP adjuvant was found to be associated with a higher FVR independent of the Env immunogens tested, while homologous and heterologous boosting regimens resulted in similar FVRs when the same adjuvant was used. Glycan trimming also increased the tier 1 NAb response against clade B SF162 while showing no nonspecific MLV reactivity in TZM-bl assays (Fig. S6b–d). We then tested the tier 2 positive IgG samples from eight groups against BG505.T332N carrying a C3/V4 epitope knockout (I396R) mutation (Fig. 6c and Fig. S6e). In contrast to those from the trimer groups, most samples from the SApNP groups appeared to lack NAbs to this immunodominant glycan epitope. Vaccine responders from the trimer and SApNP groups, subject to IgG sample availability, were further tested against a 12-virus global panel in TZM-bl assays (Fig. S6f). The $IC_{50}$ titers indicated weak but consistent neutralization of 398F1 (clade A) by an appreciable number of samples from the trimer (regimen 2 only) and SApNP groups (both regimens 1 and 2).

A rabbit study was performed using the mouse protocol (Fig. 6a) to examine the effect of glycan trimming on NAb responses to glycan holes. Four groups of six rabbits were immunized to compare the E2p/AV and I3-01v9/AP formulations, each combined with regimens 1 and 2. While purified IgG at week 0 showed a clean background (Fig. S6g), purified IgG at week 11 neutralized autologous tier 2 BG505.T332N with patterns largely resembling those of the mouse study (Fig. 6d and Fig. S6h). As Env immunization can readily generate an autologous tier 2 NAb response in rabbits, glycan trimming and the boosting strategy had little effect on the FVR (Fig. 6e). Similar to the mouse study, purified IgG samples from weeks 0 and 11 were also tested in TZM-bl assays against tier 1 clade B SF162 and the negative control, MLV (Fig. S6g, h). We then analyzed the prevalence of glycan hole NAbs in selected rabbit samples from each group (Fig. 6f and Fig. S6i). Interestingly, purified IgG from the BG505 I3-01v9/AP (regimen 2) group exhibited notably lower sensitivity to the four glycan hole mutations, supporting our hypothesis that glycan trimming may offer a general solution to overcome glycan holes in Env vaccination. Lastly, select rabbit IgG samples were tested against the 12-virus global panel (Fig. S6j). The $IC_{50}$ titers showed negligible cross-clade NAb responses, except for one animal in the I3-01v9/AP group (regimen 2).

Our results revealed the importance of glycan trimming in improving the immunogenicity of HIV-1 Env vaccines in small animal models. The 50–88% FVR noted for the glycan-trimmed trimer in mice was rather surprising, given the past failures in tier 2 NAb elicitation by

stabilized BG505 trimers[27,55] and the difficulties in improving it[70,96]. Glycan trimming enabled antibody access to a broad range of epitopes that would otherwise be occluded by the dense glycan shield while minimizing glycan-related immunodominance, such as glycan holes in rabbits.

## Comparison of wildtype and glycan-trimmed BG505 UFO Env-SApNPs in NHPs

NHPs have served as a "gatekeeper" in evaluating HIV-1 Env vaccines[125–129]. Various trimer designs, injection routes, and delivery methods have been carefully examined. Native-like Env trimers, only when formulated with a potent adjuvant, elicited autologous tier 2 NAb responses in NHPs. The $SHIV_{BG505}$ challenge study demonstrated that NAb titers, but not T cells or antibody-dependent cell-mediated cytotoxicity (ADCC) activity, correlated with protection[126].

We first tested wildtype SApNPs in NHPs using a regimen modeled after a recent NHP study of BG505 trimers[127] (Fig. 7a). Two groups of six rhesus macaques were immunized subcutaneously three times at weeks 0, 8, and 24, with 13 blood draws throughout the study period. The E2p and I3-01v9 SApNPs (200 μg/dose) were formulated with AV and AP, respectively. TZM-bl neutralization assays were performed to evaluate the NAb response in NHP sera from the last time point at week 28 (Fig. 7b and Fig. S7a, b). Overall, wildtype SApNPs failed to elicit an autologous NAb response to tier 2 clade A BG505.T332N except for one macaque in the E2p/AV group, which was just above the threshold for neutralizing activity, while all macaques in both groups showed robust neutralization of tier 1 clade B SF162 (Fig. 7b and Fig. S7b). In the control assays, the pre-immunization NHP samples exhibited little to no neutralizing activity against HIV-1 (tier 2 BG505.T332N and tier 1 SF162) and MLV (Fig. S7a). The poor immunogenicity observed for wildtype SApNPs is not unexpected. In previous studies, native-like Env immunogens were often formulated with more potent adjuvants to induce autologous NAb responses[125–128]. We next tested glycan-trimmed BG505 SApNPs in NHPs using the two regimens adopted from the mouse study with extended boost intervals (Fig. 7c). Four groups of four rhesus macaques were immunized intramuscularly four times at weeks 0, 4, 12, and 24 with 9 blood draws throughout the study period. In this study, E2p and I3-01v9 SApNPs (100 μg/dose) were formulated with the AP adjuvant based on the mouse immunogenicity data (Fig. 6b). Owing to the limited availability of animals, the experiments were designed to prioritize the evaluation of the glycan trimming strategy in NHPs, with a focus on the comparison of homologous vs. heterologous boost. Overall, glycan trimming appeared to have a positive impact on autologous tier 2 NAb elicitation compared with wildtype glycans (Fig. 7d). Specifically, glycan trimming improved the FVR for both E2p and I3-01v9 with an FVR of 50% for all four groups except for the I3-01v9/AP group using regimen 2 (Fig. 7e, middle and

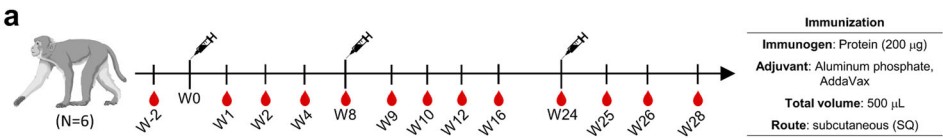

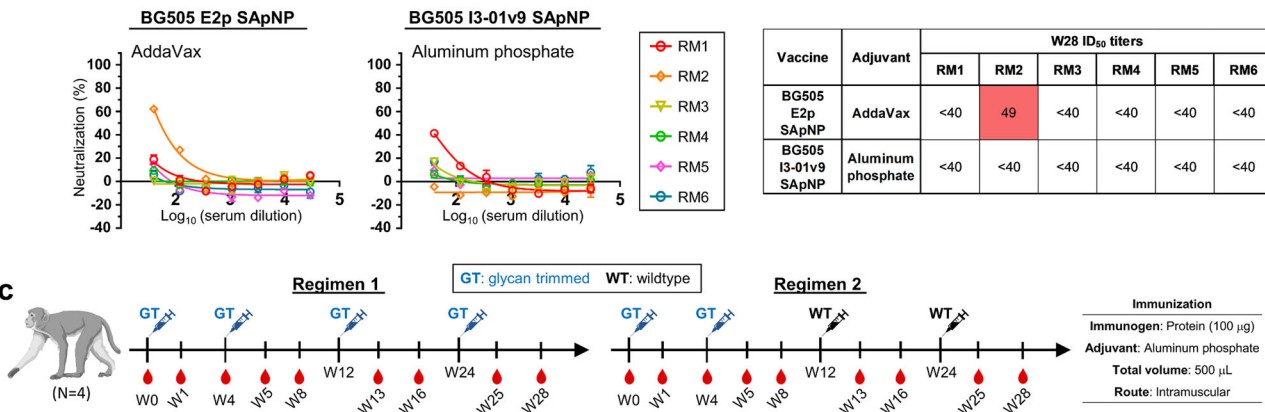

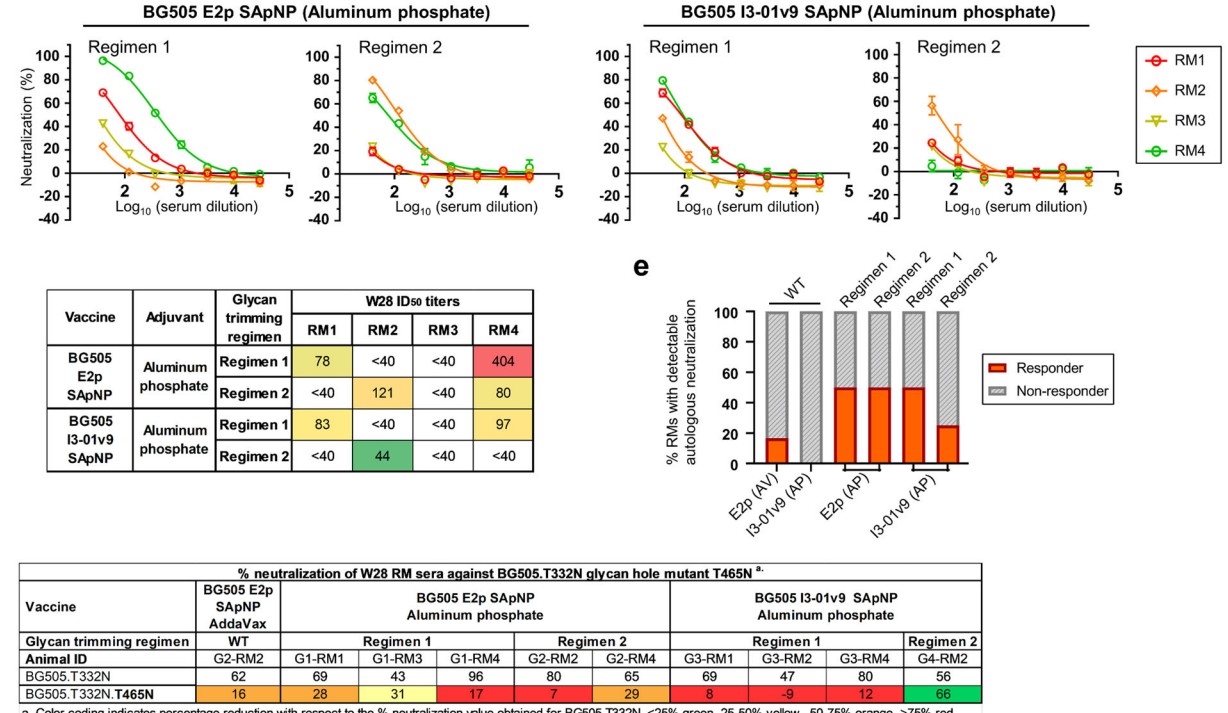

**Fig. 7 | Immunogenicity of BG505 UFO Env immunogens with wildtype and trimmed glycans in NHPs. a** Schematic representation of the immunization regimen for wildtype BG505 UFO trimer-presenting E2p and I3-01v9 SApNPs. E2p was formulated with AddaVax and I3-01v9 with aluminum phosphate. **b** Neutralization of tier 2 clade A BG505.T332N by rhesus macaque (RM) sera from week 28 (*n* = 6 RMs/group). Neutralization curves and ID$_{50}$ titers are shown. **c** Schematic representation of immunization regimen 1, where glycan-trimmed immunogens were used throughout, and regimen 2, where glycan-trimmed and wildtype immunogens were used as prime (doses 1 and 2) and boost (doses 3 and 4), respectively. SApNPs were formulated with aluminum phosphate for all groups in both regimens. **d** Neutralization of tier 2 clade A BG505.T332N by NHP sera from week 28 (*n* = 4

RMs/group). Neutralization curves and ID$_{50}$ titers are shown. **e** FVR comparison for E2p and I3-01v9 SApNPs using the week-28 NHP serum neutralization data against BG505.T332N. Left: wildtype; middle and right: glycan-trimmed (regimens 1 and 2). **f** Select NHP sera from week 28 (≥40% autologous neutralization) against BG505.T332N and its variant with the T465N mutation in the C3/465 epitope. The TZM-bl neutralization assay was performed in duplicate at a starting dilution of 40 and followed by three-fold dilutions. Error bars in **b** and **d** represent the difference between the duplicate values at each dilution for each sample. A vaccine responder is defined as a subject with ID$_{50}$ ≥ 40. Rhesus macaque images created with BioRender.com.

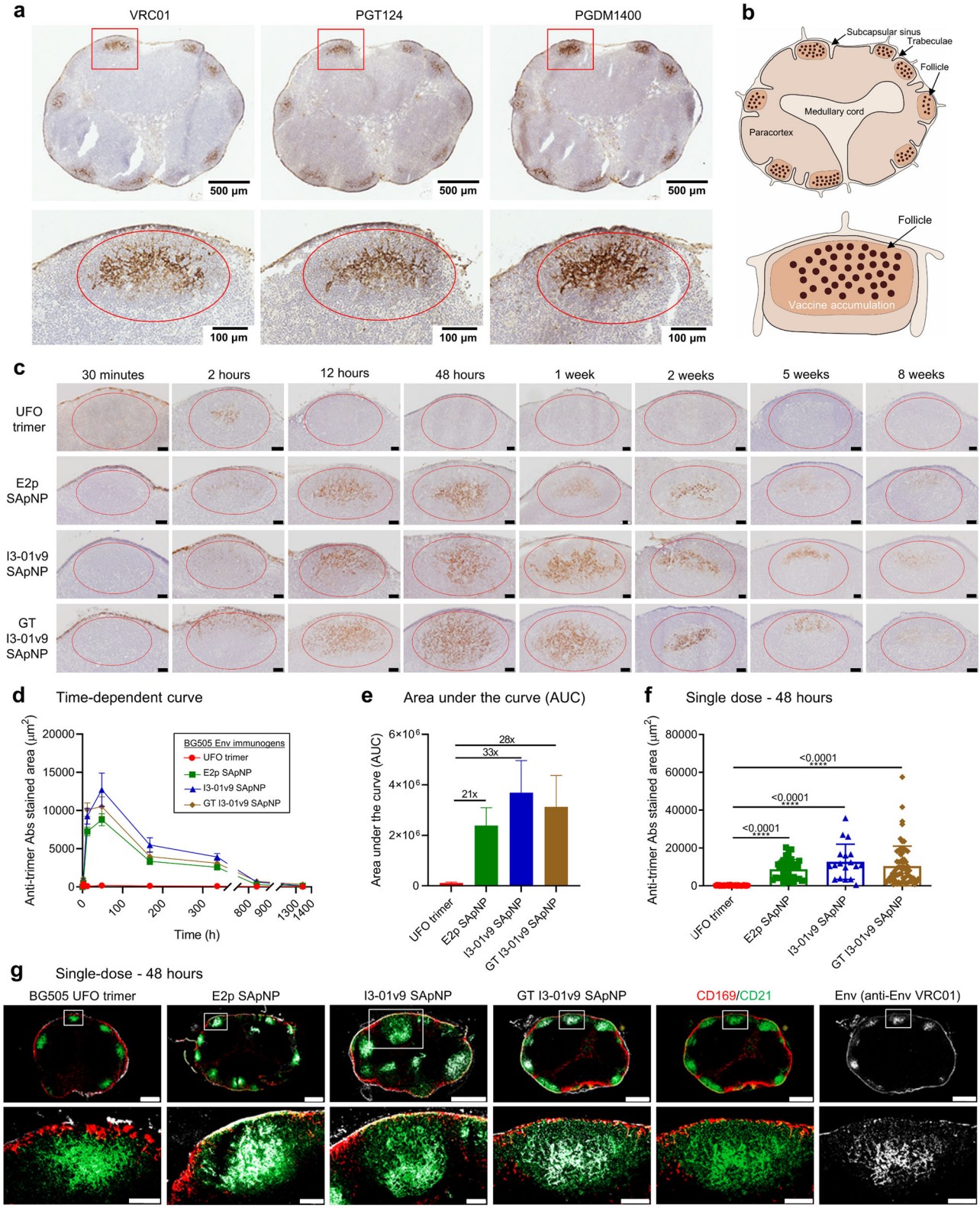

right); in contrast, wildtype E2p and I3-01v9 induced weak autologous tier 2 NAb responses with FVRs of 16.7% and 0%, respectively (Fig. 7e, left). We then analyzed the tier 2 positive NHP sera against the BG505.T332N variant with a T465N mutation in the C3/465 epitope, which is responsible for autologous NAb responses and protection against SHIV$_{BG505}$ challenge in NHPs[60,67,72] (Fig. 7f). Two out of ten samples partially or completely avoided this glycan epitope but still showed tier 2 neutralization. The pre-immunization and week

28 serum samples did not exhibit nonspecific reactivity against MLV in the TZM-bl assays (Fig. S7c, d).

Our NHP data suggest that glycan trimming may be a general strategy to improve vaccine-induced NAb responses. The C3/465 epitope remained a main neutralizing target[60,67,72] after glycan trimming. The finding that glycan-trimmed Env immunogens could be formulated with an aluminum adjuvant to achieve tier 2 NAb elicitation in NHPs is important, because only hints of a tier 2 response were noted

**Fig. 8 | Prolonged retention of BG505 UFO trimer-presenting SApNPs in lymph node follicles. a** Distribution of I3-01v9 SApNPs in a lymph node at 12 h after a single-dose injection (10 μg/injection, 40 μg/mouse). Anti-Env bNAbs VRC01, PGT124, and PGDM1400 were used to stain the lymph node tissue sections. **b** Schematic illustration of I3-01v9 SApNP accumulation in a lymph node. **c** Vaccine trafficking and retention shown by histological images of the distribution of BG505 UFO trimers and E2p and I3-01v9 SApNPs in lymph node follicles at 30 min to 8 weeks after a single-dose injection. Scale bar = 50 μm for each image. All BG505 immunogens had wildtype glycans except that a glycan-trimmed (GT) I3-01v9 SApNP was included for comparison. **d** Time-dependent curve and **e** Area under the curve (AUC) of the VRC01-stained area in histological images of the vaccine retention in lymph node follicles over 8 weeks. Data were collected from more than 10 lymph node follicles (n = 3–5 mice/group). **f** Quantification of vaccine accumulation in lymph node follicles at 48 h after a single-dose injection. Data were collected from 18-66 lymph node follicles. **g** Interaction of BG505 UFO trimers and SApNPs with FDC networks in lymph node follicles at 48 h after a single-dose injection. All SApNP immunogens, regardless of the NP platform and glycan treatment, were colocalized with FDC networks. Immunofluorescent images are pseudo-color-coded (CD21⁺, green; CD169⁺, red; VRC01, white). Scale bars = 500 and 100 μm for complete lymph node and enlarged image of a follicle, respectively. The data points are expressed as mean ± SEM for **d** and SD for **e** and **f**. The data were analyzed using one-way ANOVA followed by Tukey's multiple comparison post hoc test. ****$p < 0.0001$.

for an aluminum-adjuvanted BG505 trimer vaccine in humans[130]. However, the current studies, while promising, were mostly observational due to their differences in the immunization protocols (e.g., group size, vaccine dose, and injection route). More extensive NHP studies with matching regimens would be required to verify these findings.

## Distribution, trafficking, and retention of BG505 UFO trimers and SApNPs in lymph nodes

Following a similar protocol[124], we studied the in vivo behavior of the wildtype BG505 UFO trimer and SApNPs E2p and I3-01v9 to understand how they interact with immune cells and induce immune responses. The endo H-treated I3-01v9 SApNP was included to probe the effect of glycan trimming on the immunological characteristics of SApNPs. To induce a robust humoral response, these vaccines must be transported to lymph nodes, accumulate and be retained in follicles, present native-like Env to B cells, and engage B cell receptors (BCRs)[131–133]. Here, we first examined the transport and distribution of SApNPs, using I3-01v9 as an example, in lymph nodes via footpad injections (4 footpads, 10 μg/footpad). Brachial and popliteal sentinel lymph nodes were isolated from both sides of the mouse body at 12 h after a single-dose injection for histological analysis. Immunostaining by human bNAbs VRC01[115], PGT124[134], and PGDM1400[135] was used to detect the BG505 UFO Env presented on I3-01v9 SApNPs (Fig. 8a). Consistent with findings from previous studies of SARS-CoV-2 spike SApNPs[124] and ovalbumin-conjugated gold NPs[136], BG505 UFO trimer-presenting SApNPs accumulated in the centers of lymph node follicles (Fig. 8a, images on the left; Fig. 8b, schematics on the right). Immunostaining images obtained from all three bNAbs demonstrate similar distributions in lymph nodes (Fig. 8a, b and Fig. S8a, b). The CD4bs-directed bNAb VRC01[115] was selected to study the trafficking of four BG505 Env immunogens in lymph nodes because of its optimal signal-to-noise ratio.

We next determined the trafficking and retention patterns of the BG505 UFO trimer and three SApNPs in lymph node follicles over a period of 8 weeks after a single-dose injection (4 footpads, 10 μg/footpad) (Fig. 8c). Histological images indicated that all Env immunogens were transported into lymph nodes and accumulated in the subcapsular sinus within 30 min (Fig. 8c). The BG505 UFO trimer was trafficked into lymph node follicles within 2 h and cleared by 12 h. In contrast, the three SApNPs first appeared in lymph node follicles at 2 h, further accumulated at 12 h, peaked at 48 h, and remained detectable over a period of 5–8 weeks (Fig. 8c). The VRC01-stained area was quantified in a time-dependent manner, showing a ~420-times longer retention for the three SApNPs vs. the soluble trimer (Fig. 8c, d). The area under the curve (AUC) suggested that the exposure of SApNPs in lymph node follicles is 20–32 times higher than the soluble trimer (Fig. 8e). At 48 h, a significantly greater accumulation, 46-66 times, was found when the BG505 UFO trimer was presented multivalently (Fig. 8f). These findings are consistent with previous studies[124,136], in which small 5–15 nm particles are cleared from the follicles within 48 h, but large 50-100 nm particles persist for weeks.

Interestingly, SApNPs displaying the BG505 UFO trimer were retained notably longer than those displaying the SARS-CoV-2 spike (5–8 weeks vs. ~2 weeks), which correlates well with their antigen thermostability ($T_m$), 68.4 °C for the BG505 UFO Env vs. 47.6 °C for the SARS-CoV-2 S2GΔHR2 spike[27,91]. We also noted that glycan trimming had minimal impact on SApNP distribution, trafficking, and retention in lymph nodes, in contrast to a recent report[95].

Follicular dendritic cells (FDCs) form a network structure and play a critical role in antigen retention and presentation in lymph node follicles[131–133,137]. FDCs retain soluble antigens, immune complexes, viruses, and bacteria, and induce GC initiation and maintenance[132,136,138–140]. Based on our previous studies[124,136], we hypothesized that FDC networks are the major resident site of these BG505 Env immunogens. To test this hypothesis, we collected sentinel lymph nodes at the peak of SApNP accumulation (48 h) and other timepoints (30 min to 8 weeks) after a single-dose injection (Fig. 8g and Fig. S8c–i). Lymph node tissues were stained with bNAb VRC01[141] for BG505 Env (white), anti-CD21 antibodies for FDCs (green), and anti-CD169 antibodies for subcapsular sinus macrophages (red). The signals of SApNPs colocalized with FDC (CD21⁺) networks at 48 h (Fig. 8g), thus supporting our hypothesis.

## Interactions of BG505 UFO Env-SApNPs with FDCs and phagocytic cells in lymph nodes

FDC networks create areas of high antigen density in lymph node follicles to facilitate effective BCR crosslinking, B cell activation, and GC reactions[77,131,132]. Previous analyses by TEM revealed that FDCs could align ovalbumin-conjugated gold NPs[136] and SARS-CoV-2 spike-presenting SApNPs[124] on their surfaces or dendrites through interactions between complement protein 3 (C3) and complement receptor 2 (CR2)[136]. Here, we studied the interface between FDC dendrites and B cells to better understand how FDCs process HIV-1 SApNPs formulated with various adjuvants to engage B cells (Fig. 9 and Fig. S9). To this end, we injected mice with AV- or AP-formulated SApNPs through their hind footpads (2 footpads, 50 μg/footpad). Fresh popliteal sentinel lymph nodes were isolated at 2, 12 and 48 h after a single-dose injection and processed immediately for TEM analysis. TEM images showed that FDC dendrites embrace and interact with B cells in lymph node follicles (Fig. 9a–c and Fig. S9a–m). Intact E2p SApNPs (round-shaped granules, yellow arrows) and AV particles (oil-in-water nano-emulsion of ~150 nm in size, green arrows) were aligned on FDC dendrites and B cell surfaces at 2, 12, and 48 h (Fig. 9a and Fig. S9a–c). Similarly, wildtype and glycan-trimmed I3-01v9 SApNPs (yellow arrows) were aligned on FDC dendrites and B cell surfaces at all three timepoints, but without the colocalization of AP adjuvants (Fig. 9b, c and Fig. S9d–i). In the adjuvant-alone experiment, AV particles were presented on FDC dendrites at 12 h after injection (Fig. S9l).

Phagocytic cells such as macrophages in the subcapsular sinus and medullary sinus of lymph nodes can capture large particles, transport them to migrating B cells, and deposit them on the surface of FDCs via a complement receptor-dependent mechanism[132,133,142–144]. These immune cell populations also serve as APCs that can further promote adaptive immunity. Here, we studied the association between

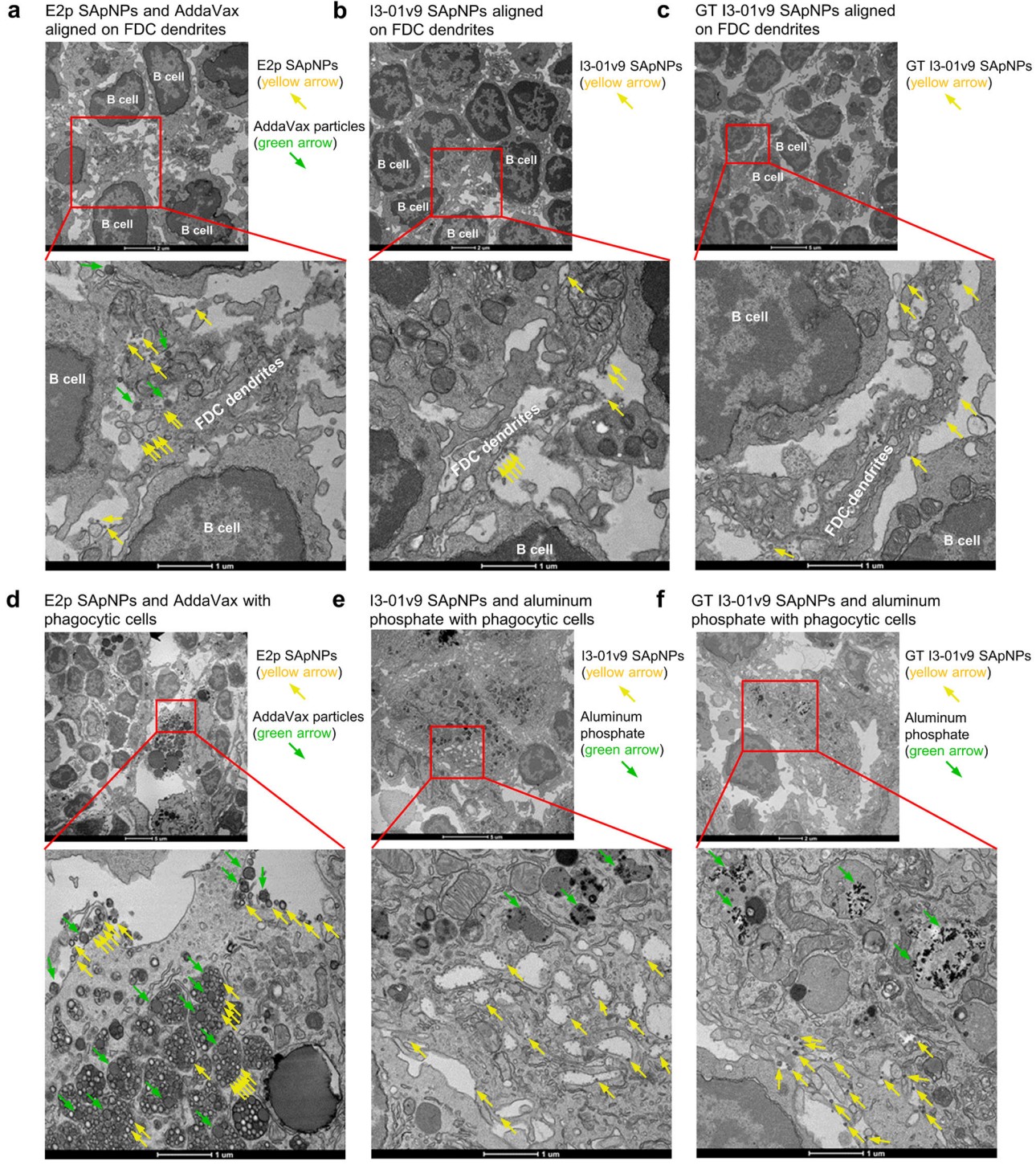

**Fig. 9 | Interaction of BG505 UFO trimer-presenting SApNPs with FDCs and phagocytic cells in lymph nodes. a** TEM images of E2p SApNPs (yellow arrows) and AddaVax adjuvant (green arrows) presented on the FDC dendrites in a lymph node at 12 h after a single-dose injection (2 footpads, 50 μg/footpad). **b** and **c** TEM images of I3-01v9 SApNPs (yellow arrows), wildtype and glycan-trimmed (GT), aligned on FDC dendrites without observable aluminum phosphate (AP) adjuvant at 12 h after a single-dose injection. **d** TEM images of E2p SApNPs (yellow arrows) and AddaVax adjuvant (green arrows) colocalized on the surface or inside endolysosomes of a macrophage at 12 h after injection. **e** and **f** TEM images of I3-01v9 SApNPs (yellow arrows), wildtype and glycan-trimmed (GT), and aluminum phosphate adjuvant (green arrows) internalized inside endolysosomes of a macrophage, with only SApNPs (yellow arrows) on the macrophage surface at 48 h for **e**, or 12 h for **f**, after the injection. TEM images were performed on 2 popliteal lymph nodes for each SApNP construct.

macrophages and adjuvanted SApNPs (Fig. 9d–f and Fig. S9n–v). E2p SApNPs and AV adjuvant particles are colocalized and aligned on the surface or inside the endolysosomes of medullary sinus macrophages at 2 h (Fig. 9d and Fig. S9n). Wildtype and glycan-trimmed I3-01v9 SApNPs behaved similarly, with AP forming visible aggregates inside

the endolysosomes of macrophages and in the extracellular matrix (Fig. 9e, f and Fig. S9o–u). Our results reveal how BG505 UFO trimer-presenting SApNPs associate with adjuvants in the intercellular and intracellular compartments of lymph nodes. Three SApNPs showed similar patterns of

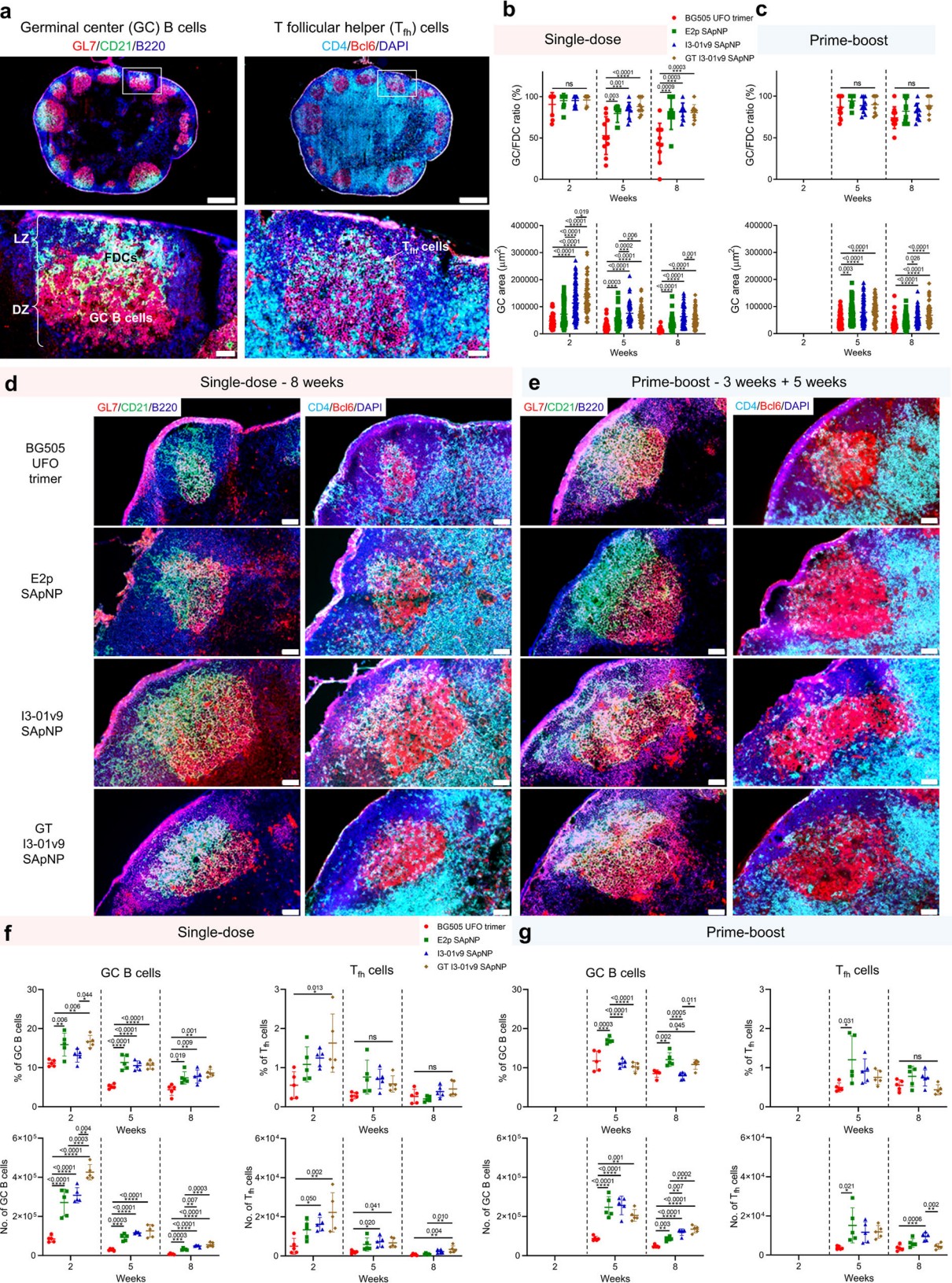

interaction with FDC networks, B cells, and phagocytic cells. Two common adjuvants (AV and AP) with distinct chemical properties interact with cells in lymph nodes through different immune pathways. FDC networks can retain SApNPs on long FDC dendrites and present them to naïve B cells, which is an intrinsic feature independent of adjuvants (Fig. S9j, k).

## Characterization of GC reactions induced by BG505 UFO trimers and SApNPs

B cell somatic hypermutation, selection and affinity maturation occur in long-lived GCs and lead to the formation of immune memory and development of (b)NAbs upon vaccination[139,145–148]. Here, we hypothesized that long-term retention of BG505 UFO trimer-presenting E2p

**Fig. 10 | Induction of robust and long-lived germinal center responses by BG505 UFO trimer-presenting SApNPs. a** Top: Representative immunohistological images of BG505 UFO-10GS-I3-01v9 SApNP vaccine-induced germinal centers (GCs) at 2 weeks after a single-dose injection (10 μg/injection, 40 μg/mouse). Bottom left: GC B cells (GL7+, red) adjacent to FDCs (CD21+, green) in lymph node follicles. Bottom Right: $T_{fh}$ cells in the light zone (LZ) of GCs. Scale bars = 500 and 100 μm for complete lymph node and enlarged image of a follicle, respectively. **b** and **c** Quantification of GCs using immunofluorescent images at 2, 5, and 8 weeks after a single-dose injection or at 2 and 5 weeks after the boost, which occurred at 3 weeks after the first dose ($n$ = 5–10 mice/group). Data were collected from 35–108

lymph node follicles. The GC/FDC ratio and GC size were determined and plotted here. **d** and **e** Representative immunohistological images of GCs induced by the BG505 UFO trimer, E2p, I3-01v9, and GT I3-01v9 SApNP vaccines at 8 weeks using a single-dose or prime-boost regimen. Scale bar = 50 μm for each image. **f** and **g** Quantification of GCs by flow cytometry after a single-dose or prime-boost injections ($n$ = 5 mice/group). Percentage and number of GC B cells and $T_{fh}$ cells were determined and plotted. The data points are shown as mean ± SD. The data were analyzed using one-way ANOVA followed by Tukey's multiple comparison post hoc test for each time point. *$p < 0.05$, **$p < 0.01$, ***$p < 0.001$, ****$p < 0.0001$.

and I3-01v9 SApNPs can induce more robust and long-lived GCs in follicles than the soluble UFO trimer. We first assessed GC reactions induced by wildtype I3-01v9 SApNP at 2 weeks after a single-dose injection (4 footpads, 10 μg/footpad). GC reactions, i.e., GC B cells (GL7+) and T follicular helper ($T_{fh}$) cells (CD4+Bcl6+), in sentinel lymph nodes were characterized by immunohistological analysis. Robust GCs (GL7+, red) were generated that were attached to FDC networks (CD21+, green) with clearly organized dark zone (DZ) and light zone (LZ) compartments in lymph node B cell follicles (B220+, blue) (Fig. 10a, left). $T_{fh}$ cells (CD4+Bcl6+, co-labeled with cyan and red) were mainly located in the LZ to support B cell affinity maturation and GC maintenance (Fig. 10a, right). Next, we applied this analysis to the BG505 UFO trimer and three SApNPs at 2, 5, and 8 weeks after a single-dose injection (Fig. 10b and Fig. S10a–c) and at 2 and 5 weeks after the boost (Fig. 10c and Fig. S10d, e). Two metrics were defined to quantify GC reactions based on immunohistological images: GC/FDC ratio (i.e., whether GC formation is associated with an FDC network, %) and GC size (i.e., occupied area)[124]. All four Env immunogens induced robust GCs, with the glycan-trimmed I3-01v9 SApNP showing the largest GCs, at 2 weeks after a single-dose injection (Fig. 10b and Fig. S10a). Following trimer immunization, the GC/FDC ratio and GC size were small and decreased rapidly over time, whereas a single dose of SApNPs generated long-lived GCs that persisted for 8 weeks (Fig. 10b, d and Fig. S10c). For the soluble trimer, robust GCs can be restored after the boost (Fig. 10c, e and Fig. S10d, e). Overall, the three SApNPs generated larger GCs than the soluble trimer, 2.3–4.2 times larger after one dose (Fig. 10b, d) and 0.4–1.1 times larger after the boost (Fig. 10c, e), both measured at week 8.

We next characterized GC reactions by flow cytometry. We collected sentinel lymph nodes at 2, 5, and 8 weeks after a single-dose footpad injection (Fig. 10f and Fig. S11a, b) and at 2 and 5 weeks after the boost (Fig. 10g and Fig. S11c) (4 footpads, 10 μg/injection). Fresh sentinel lymph nodes were disaggregated into a single-cell suspension and stained with an antibody cocktail. Flow cytometry analysis showed that the percentage and number of GC B cells and $T_{fh}$ cells were in line with the immunohistological data (Fig. 10a–e). The three SApNPs outperformed the soluble trimer at 2 weeks after a single-dose injection (Fig. 10f). Among the three SApNPs, the glycan-trimmed I3-01v9 elicited the largest GC B cell and $T_{fh}$ cell populations. GC reactions peaked at 2 weeks for all immunogens tested and declined over time. The populations of GC B cells and $T_{fh}$ cells induced by the soluble trimer were barely detectable at 8 weeks after a single-dose injection (Fig. 10f). A boost effectively expanded the populations of GC B cells and $T_{fh}$ cells but had little effect on their percentages (Fig. 10g). In addition to size-dependent trafficking and retention in lymph node follicles, GC reactions may also be influenced by adjuvants, considering that AV and AP undergo different mechanisms of cellular interactions (Fig. 9). Overall, the three SApNPs generated 3.4–6.6/0.7–1.8 times more GC B cells and 0.5–4.3/0.2–1.6 times more $T_{fh}$ cells than the soluble trimer at 8 weeks after the single-dose/boost injections, respectively (Fig. 10f–g). Together, our analysis indicates that large SApNPs can generate long-lived GCs in lymph nodes more effectively than the soluble trimer, resulting in a strong Env-specific B cell response.

## Discussion

Over the last decade, HIV-1 vaccine research has been driven by advances in the identification of bNAbs targeting diverse epitopes[6,7] and development of native-like Env trimers[12,13]. For the latter, SOSIP[19,29], NFL[23,25], and UFO[26,27] have been established as the leading trimer design platforms. Multistage vaccine strategies that follow specific B cell maturation events have been proposed and tested in knock-in mice carrying human bNAb (precursor) genes[51]. However, it is unclear whether such strategies will be relevant to human vaccination. Recent HIV-1 vaccine research also received a boost from the engineering of protein particles to mimic VLPs[74–77,80]. Various animal models have been used to assess HIV-1 Env vaccines. Rabbits can readily generate autologous tier 2 NAb responses, which mainly target glycan holes[56]. The tier 2 NAb responses in wildtype mice and NHPs are sporadic and often directed to glycan patches on the HIV-1 Env[67,70,71]. Although bNAb responses have occasionally been reported[52,58,128,149,150], it is unclear whether these are reproducible outcomes of a general vaccine solution or coincidence. Therefore, despite all advances achieved to date, HIV-1 vaccine development still faces critical challenges posed by Env, an atypical viral antigen[151].

The glycan shield is at the heart of HIV-1 Env vaccine design[31–36]. Ideally, an effective vaccine strategy must overcome the conflicting roles of glycans, namely, occluding NAb access to conserved protein epitopes vs. participating in NAb-Env interactions. Here, we aimed to develop such a strategy by combining our trimer stabilization (i.e., the UFO design[26,27]) and particulate display (i.e., the multilayered 1c-SAPNPs[91,92]) platforms. The resulting BG505 UFO-E2p-L4P and UFO-10GS-I3-01v9-L7P SApNPs were thoroughly evaluated both in vitro and in vivo. Notably, a key element in our current vaccine strategy was glycan trimming. We hypothesized that trimming mannose-rich glycans would improve epitope accessibility (e.g., CD4bs) and minimize glycan-related immunodominance (e.g., strain-specific glycan holes or glycan patches), while retaining Env binding to most glycan-reactive bNAbs. The success of this strategy hinges on glycan promiscuity[119], which allows various glycan sites and moieties to substitute for one another in glycan-bNAb interactions. The avidity effect on the NP surface may further compensate for weakened glycan-bNAb interactions on one trimmed Env trimer by enabling interactions with surrounding Env trimers. As a result, we observed rather balanced interactions between glycan-trimmed BG505 Env-SApNPs and bNAbs that target various protein and glycan epitopes.

Mice were used in our large-scale in vivo evaluation because they are inexpensive and known to generate tier 2 NAbs to epitopes that are found in human infection[70,71]. Rabbits were tested primarily to probe NAb responses to glycan holes. The most commonly used, but not the most potent[125,127,152], adjuvants were chosen because of their safety record in humans. Instead of bNAb elicitation, we focused on increasing the frequency of vaccine responders (FVR), which is defined as the ratio of tier 2 positive subjects in a group. This goal is more achievable and the low FVR has been a major challenge for native-like Env trimer vaccines in mice, NHPs, and also humans[130]. Glycan-trimmed Env immunogens mixed with AP yielded a much higher FVR in mice than wildtype ones, 50–88% vs. 12.5%, with reduced glycan-related immunodominance in mice and rabbits (Fig. 6). Our NHP

studies confirmed the beneficial effect of glycan trimming in a primate model, with the FVR reaching 50% for three of the four AP-adjuvanted SApNP groups (Fig. 7). An in-depth mechanistic analysis provided critical insights into the mode of action for these new Env immunogens. The 5- to 8-week retention of E2p and I3-01v9 SApNPs in follicles indicates sustained B cell maturation, suggesting that a longer interval between immunizations should be used for these SApNPs. The identical trafficking, presentation, and retention patterns for wildtype and glycan-trimmed I3-01v9 SApNPs indicate that the density of mannose glycans is not as critical for the follicular localization of HIV-1 NP immunogens as previously thought[95]. Notably, the methods used to track NP immunogens in lymph node tissues may result in different experimental outcomes. In our current study, the trafficking and retention of I3-01v9 SApNPs were quantified through immunostaining of HIV-1 Env trimers displayed on SApNPs using a human bNAb (VRC01) at individual timepoints, whereas in the previous studies, fluorescent dyes conjugated to protein NPs may have affected the readout due to their signal decay over time[94,95]. The stronger GC reactions observed for the glycan-trimmed I3-01v9 SApNP were consistent with early studies where mannose groups were found to be immunosuppressive[153] and their enzymatic removal improved the gp120-induced IgG response when formulated with alum[154].

Future research may focus on several fronts. First, antibodies induced by glycan-trimmed Env need to be mapped onto the wildtype Env surface[63,66] to understand how glycan trimming alters epitope targeting, as previously reported for influenza virus and SARS-CoV-2[155,156]. Second, in addition to the global glycan profiles, site-specific analysis may provide critical information to assist in development of glycan-trimmed Env immunogens for diverse HIV-1 subtypes. Site occupancy combined with structural analysis will help explain how antibodies recognize Env epitopes after glycan trimming. Third, the effect of Env glycans on the immune system warrants a more in-depth investigation, e.g., the involvement of mannose moieties in immunosuppression[153] vs. vaccine trafficking and immunogenicity[94,95]. Fourth, the immunization regimen requires extensive optimization. The 3-week interval used in our screening regimen may not be optimal for SApNPs with a retention time of 5 weeks or longer. A short interval would interrupt the generation of memory B cells and long-lived plasma cells and suppress the recall antibody response. Lastly, a systematic comparison would be needed to identify more effective adjuvants, which could substantially improve vaccine-induced NAb responses, as demonstrated for a SARS-CoV-2 SApNP in our previous study[124]. The resulting vaccine formulation, with an optimized regimen, could then be considered for evaluation in human trials.

## Methods

### Antibodies
We utilized a panel of bNAbs and non-NAbs to characterize the antigenicity of various native-like trimers and gp140 nanoparticles. Antibodies were requested from the NIH AIDS Reagent Program (https://www.aidsreagent.org/) except for 438-B11[98], PGT151[110], and M4H2K1[71], which were produced in-house together with the fragment antigen-binding (Fab) regions of PGT128 and VRC01, using previously described protocols[71].

### Expression and purification of HIV-1 Env trimers and SApNPs
The BG505 UFO trimer and SApNPs used for the in vitro characterization and in vivo assessment of immunogens with wildtype glycans were transiently expressed in ExpiCHO cells (Thermo Fisher, catalog no. A29133) using a previously described protocol[27]. Briefly, ExpiCHO™ cells were thawed and incubated with ExpiCHO™ Expression Medium (Thermo Fisher) in a shaker incubator at 37 °C, 135 rpm, and 8% $CO_2$. When cells reached a density of $10 × 10^6$/ml, ExpiCHO™ Expression Medium was added to reduce cell density to $6 × 10$/ml for transfection. ExpiFectamine™ CHO/plasmid DNA complexes were prepared for 100-ml transfection in ExpiCHO cells following the manufacturer's instructions. Kifunensine (10 mg/l, Tocris Bioscience) and swainsonine (3.4 mg/l, Santa Cruz Biotechnology) were added at the time of ExpiCHO transfection to inhibit α-mannosidase I and II to enrich oligomannose- and complex-type glycans, respectively. For Env constructs tested in this study, 100 µg of antigen plasmid and 320 µL of ExpiFectamine™ CHO reagent were mixed in 7.7 ml of cold Opti-PRO™ medium (Thermo Fisher). After the first feed on day 1, ExpiCHO cells were cultured in a shaker incubator at 32 °C, 120 rpm, and 8% $CO_2$ following the Max Titer protocol with an additional feed on day 5 (Thermo Fisher). Culture supernatants were harvested 13–14 days after transfection, clarified by centrifugation at $1467 × g$ for 20 min, and filtered using a 0.45 µm filter (Thermo Fisher). For the BG505 UFO trimer, the Env protein was extracted from the culture supernatants using a *Galanthus nivalis* lectin (GNL) column (Vector Labs), whereas for all SApNPs, BG505 and non-BG505 (CH505 and H078.14) Env-fusion proteins were purified using PGT145 and 2G12 columns, respectively. Trimer was purified on a Superdex 200 Increase 10/300 GL column or a HiLoad 16/600 Superdex 200 PG column (GE Healthcare), whereas SApNPs were characterized on a Superose 6 10/300 GL column. Protein concentration was determined using $UV_{280}$ absorbance with theoretical extinction coefficients.

### Generation of stable CHO cell lines expressing two BG505 UFO-Env NP vaccine candidates
Stable CHO cell lines expressing BG505 UFO-E2p-L4P and UFO-10GS-I3-01v9-L7P NPs were generated by the cationic lipid mediated transfection of a parental suspension culture adopted CHO-K1 host cell line (ATCC no. CCL-61) using bacterial artificial chromosomes (BACs) as gene transfer vehicles[157]. BAC vector modification and integration of the BG505 Env-NP expression cassettes by recombineering were performed by Gen-H GmbH (Heidelberg, Germany). Parental CHO-K1 host cells were cultured in a 125 ml shake flask (Corning) in 20 ml CD CHO medium (Gibco, Thermo Fisher) supplemented with 8 mM L-glutamine, 6.25 mg/ml phenol red and 1:100 Anti-Clumping Agent (Gibco, Thermo Fisher). Shaker flasks were cultivated in a humidified ISF1-X incubator shaker (Kuhner) at 37 °C, 125 rpm, and 5% $CO_2$, and multi-well plates were maintained in a static incubator cabinet at 37 °C and 5% $CO_2$. The passaging of cells for the purpose of culture maintenance was performed every 3–4 days to starting cell densities of $0.3 × 10^6$ cells/ml. Per transfection, $1 × 10^6$ cells were prepared in 1.8 ml CD DG44 medium (Gibco, Thermo Fisher), 5 µg BAC DNA, and 25 µg Lipofectin® (1 mg/ml; Thermo Fisher) diluted in 0.2 ml of CD DG44 medium, and transfection mixes were incubated for up to 48 h in six-well plates (Greiner). Following incubation, media of transfection pools were replaced with selection medium (CD CHO medium + supplements and 0.5 mg/ml Geneticin sulfate (Sigma, catalog no. G418). Cultures were transferred to 96-well plates (Thermo Fisher) and transfection cell pools were continuously monitored for signs of cell growth and increasing viabilities, as well as BG505 Env-NP expression by ELISA (NP capture: *Galanthus nivalis* lectin (GNL, Sigma)/NP detection: anti-HIV-1 gp120 mAb C2G12 (Polymun Scientific)). High NP expressing cell pools were expanded to 50 ml of TubeSpin® Bioreactor tubes (TPP) and cell pools were evaluated for specific growth (µ) and specific NP expression (qp). The top performing cell pools[2–4] were single-cell-cloned following an in-house established limiting-dilution procedure in 384-well plates (Corning). Clonality was confirmed using a Cell Metric plate imaging device (Solentim). Clonal cell populations were again screened in 96-well plates (Thermo Fisher) by ELISA, and the 20 best performing clonal cell lines were further evaluated in a small-scale screen (5 ml) in 50 ml TubeSpin® Bioreactor tubes (TPP) with respect to cell growth and NP expression by ELISA in addition to non-reducing SDS-PAGE and Western blot. Based on the evaluation results for the top 20 clones, the top eight cell clones were selected for each BG505 Env-NP and challenged in small-scale fed-batch experiments with the aim to identify

clones that may have the potential to serve as prospective production clones under good manufacturing practice (GMP) conditions. Fed-batch experiments were performed in 125 ml shaker flasks with 45 ml culture volume using ActiPro medium (HyClone) and the corresponding nutrient feeds Cell Boost 7a and 7b (both HyClone) in a shaking incubator. Small-scale research cell banks (RCBs) were established and cryo-preserved to support future master cell bank (MCB) establishment. Cell line development was performed at Polymun Scientific GmbH (Klosterneuburg, Austria) through a contract from Uvax Bio.

### SDS-PAGE and BN-PAGE

BG505 Env-NPs were analyzed by sodium dodecyl sulfate–polyacrylamide gel electrophoresis (SDS-PAGE) and blue native-polyacrylamide gel electrophoresis (BN-PAGE). The proteins were mixed with loading dye and added to either a 10% Tris-Glycine Gel (Bio-Rad) or a 4–12% Bis-Tris NativePAGE™ gel (Life Technologies). For SDS-PAGE under reducing conditions, the proteins were first treated with dithiothreitol (DTT, 25 mM) and boiled for 5 min at 100 °C. SDS-PAGE gels were run for 20 min at 250 V using SDS running buffer (Bio-Rad), and BN-PAGE gels were run for 2–2.5 h at 150 V using NativePAGE™ running buffer (Life Technologies) according to the manufacturer's instructions. The gels were stained using Coomassie Brilliant Blue R-250 (Bio-Rad) and de-stained using a solution of 6% ethanol and 3% glacial acetic acid.

### Differential scanning calorimetry (DSC)

Thermal melting curves of BG505 Env UFO-FR, UFO-E2p-L4P, and UFO-10GS-I3-01v9-L7P SApNPs following PGT145 and SEC purification were obtained from a MicroCal PEAQ-DSC Man instrument (Malvern). Samples of the E2p and I3-01v9 SApNPs after glycan trimming by endo Hf and SEC purification were also analyzed. Briefly, the purified SApNP protein was buffer exchanged into 1×PBS buffer and concentrated to 0.8 μM before analysis by the instrument. Melting was probed at a scan rate of 60 °C per hour from 20 °C to 100 °C. Data processing, including buffer correction, normalization, and baseline subtraction, was conducted using MicroCal PEAQ-DSC software. Gaussian fitting was performed using GraphPad Prism 9.3.1 software.

### Dynamic light scattering (DLS)

Particle size distributions of BG505 Env-NPs based on three NP platforms (FR, E2p-L4P, and I3-01v9-L7P) were obtained from a Zetasizer Ultra instrument (Malvern). PGT145/SEC-purified NPs from ExpiCHO cells were diluted to 0.2 mg/ml using 1×PBS buffer, and 30 μl of the prepared NP sample was added to a quartz batch cuvette (Malvern, catalog no. ZEN2112). Particle size was measured at 25 °C in back scattering mode. Data processing was performed on the Zetasizer, and the particle size distribution was plotted using GraphPad Prism 9.3.1 software.

### Endo H treatment and removal

Glycan trimming of Env immunogens was performed using Endo-Hf, a fusion of endo H and MBP (NEB, catalog no. P0703L), by mixing 20 μg of Env protein, 2 μl of 10× GlycoBuffer3, 5 μl of endo H, and H$_2$O (if necessary) to make a 20 μl reaction. The mixture was kept at room temperature (25 °C) for 5 h to facilitate enzymatic processing of the Env glycans. The reaction volume can be scaled up proportionally to generate more glycan-trimmed material for immunization. After 5 h of incubation, the mixture was passed through a Superose 6 column to remove the MBP-tagged endo H. Both NP and endo Hf fractions were collected for quantitation of the residual endo Hf using an anti-MBP mouse antibody (Sigma, catalog no. M6295-.2 ML) in ELISA. While most endo Hf can be readily removed from NP samples by SEC, a second purification step must be included for the trimer because trimer and endo Hf have separate but close SEC peaks. Amylose resin (NEB,

catalog no. E8021S) was used to remove residual endo Hf from the SEC-purified trimer fractions. The flow-through was collected to quantify the residual endo Hf and then adjusted to appropriate concentrations for in vitro and in vivo studies. To achieve the most complete glycan trimming, 1–20 μg of the Env protein was mixed with 1 μl of Glyco-protein Denaturing Buffer (NEB, 10×) and H$_2$O (if necessary) in a reaction volume of 10 μl. The reaction was then heated to 100 °C for 10 min to denature the protein, prior to endo H treatment at 37 °C for 1 h. This protocol was only used to test the maximum trimming of Env glycans by endo H in Fig. 3g. The denatured, glycan-trimmed materials were not used in animal studies.

### Enzyme-linked immunosorbent assay (ELISA)

In this study, ELISA was performed to detect residual endo Hf in trimer and SApNP samples after glycan trimming. Each well of a Costar™ 96-well assay plate (Corning) was first coated with 50 μl of PBS containing 0.2 μg of appropriate antigens. The plates were incubated overnight at 4 °C, and then washed five times with wash buffer containing PBS and 0.05% (v/v) Tween 20. Each well was then coated with 150 μl of a blocking buffer consisting of PBS and 40 mg/ml$^{-1}$ blotting-grade blocker (Bio-Rad). The plates were incubated with the blocking buffer for 1 h at room temperature, and then washed five times with wash buffer. The human bNAb VRC01 and MBP-specific mouse antibody (Sigma, catalog no. M6295-.2 ML) were diluted in blocking buffer to a maximum concentration of 1 μg/ml and to reach × 466 dilution (as recommended by the manufacturer), respectively, followed by a 10-fold dilution series. For each antibody dilution, a total 50 μl volume was added to the appropriate wells. Each plate was incubated for 1 h at room temperature, and then washed five times with PBS containing 0.05% Tween 20. A 1:5000 dilution of goat anti-human IgG antibody (Jackson ImmunoResearch Laboratories) or a 1:3000 dilution of goat anti-mouse IgG antibody (Jackson ImmunoResearch Laboratories) was then made in wash buffer (PBS containing 0.05% Tween 20), with 50 μl of this diluted secondary antibody added to each well. The plates were incubated with the secondary antibody for 1 h at room temperature, and then washed five times with PBS containing 0.05% Tween 20. Finally, the wells were developed with 50 μl of TMB (Life Sciences) for 3–5 min before stopping the reaction with 50 μl of 2 N sulfuric acid. The resulting plate readouts were measured at a wavelength of 450 nm. As a rapid screening ELISA assay, only four antibody concentrations/dilutions were tested.

### Negative-stain EM analysis

The initial evaluation of various Env-NP samples was performed by the Core Microscopy Facility at The Scripps Research Institute. All NP samples were prepared at a concentration of 0.01–0.05 mg/ml. Carbon-coated copper grids (400 mesh) were glow-discharged and 8 μl of each sample was adsorbed for 2 min. Excess sample was wicked away and grids were negatively stained with 2% uranyl-formate for 2 min. Excess stain was wicked away and the grids were allowed to dry. Samples were analyzed at 80 kV with a Talos L120C transmission electron microscope (Thermo Fisher) and images were acquired with a CETA 16M CMOS camera. The structural analysis of glycan-trimmed BG505 UFO trimer bound to bNAbs PGT128 and VRC01 was performed by the Core Microscopy Facility following the protocol described above. For each complex, a total of 55 images were manually collected at a magnification of 73,000×. CryoSPARC 2 software[158] on the Scripps Garibaldi cluster was used to analyze the EM images, including imaging processing, particle picking, 2D classification, and 3D reconstruction (Fig. S4r). Validation of BG505 UFO-E2p-L4P and UFO-10GS-I3-01v9-L7P NP samples prior to high-resolution cryo-EM was performed in the Hazen EM facility at The Scripps Research Institute. The experiments were performed as previously described[101,159]. The concentrated BG505 Env-NP samples were diluted to 50 μg/ml in TBS buffer (Alfa Aesar) and loaded onto the carbon-coated 400-mesh Cu grid. Prior to sample application

the grids were glow-discharged at 15 mA for 30 s. The samples were blotted off the grids after 10 s and negatively stained with 2 % (w/v) uranyl-formate for 60 s. A Tecnai Spirit electron microscope (120 keV) featuring a Tietz 4k × 4k TemCam-F416 CMOS camera was used for data acquisition. The nominal magnification was 52,000×. The resulting pixel size at the specimen plane was 2.05 Å, and the defocus was set to −1.50 μm. Total electron dose per image was adjusted to 25 e⁻/Å². Images were recorded using the Leginon software suite[160]. Data were visualized in the Appion data processing suite[161].

## Cryo-EM analysis of two BG505 UFO trimer-presenting SApNPs

The cryo-EM grid preparation was performed as the following. BG505 UFO-E2p-L4P and UFO-10GS-I3-01v9-L7P samples were concentrated to 1.5 and 2.5 mg/ml, respectively. Immediately prior to grid application, lauryl maltose neopentyl glycol (LMNG) was added to each sample at a final concentration of 0.005 mM. Quantifoil R 2/1 holey carbon copper grids (Cu 400 mesh) were the main type of grids used. The grids were pretreated with Ar/O2 plasma (Gatan Solarus 950 plasma system) for 10 s before sample loading. Vitrobot Mark IV (Thermo Fisher Scientific) was used for the sample application, blotting, and vitrification steps. The temperature was set to 10 °C, humidity was 100%, blotting force was set to 0, wait time was 10 s, and blotting time varied in the range of 3.5–5.5 s. For grid preparation, 3 μl of the NP sample (with LMNG) was loaded onto plasma-activated grids. Following the blot step the grids were plunge-frozen into liquid ethane cooled with liquid nitrogen. The cryo-EM data were collected from two microscopes. Specifically, the I3-01v9-L7P NP data were collected on a Talos Arctica TEM (Thermo Fisher Scientific) operating at 200 kV. The E2p-L4P NP data were acquired on an FEI Titan Krios TEM at 300 kV (Thermo Fisher Scientific). Both microscopes were equipped with a Gatan K2 Summit direct electron detector camera and sample auto-loader. The Leginon software suite[160] was used for automated data acquisition. Data collection parameters are shown in Table S1. Cryo-EM data were processed as described previously[101]. MotionCor2[162] was used for frame alignment and dose-weighting and GCTF was applied for the estimation of CTF parameters. The early processing steps were performed in CryoSPARC 2[158]. Particles were picked using template picker, extracted, and subjected to two rounds of 2D classification to eliminate bad picks and heterogeneously looking particles. After the 2D cleaning step, 17097 and 4806 particles were retained for BG505 UFO-E2p-L4P and UFO-I3-01v9-L7P NPs, respectively, for further processing steps. These particle subsets were transferred to Relion/3.0[163]. Ab-initio reconstruction in cryoSPARC was used to generate the starting reference models for the 3D steps in Relion. Two iterative rounds of 3D classification and refinement with imposed icosahedral symmetry restraints were applied to produce the final 3D maps of the E2p-L4P and I3-01v9-L7P NP scaffolds. A soft solvent mask around the corresponding NP scaffold was introduced for 3D classification, 3D refinement, and postprocessing steps to exclude signal contributions from flexibly linked BG505 trimers and additional stabilizing domains (e.g., LD4 in E2p-L4P). For the E2p-L4P NP, the final subset had 7,672 particles and yielded a map resolution of 3.7 Å. For the I3-01v9-L7P NP, the final subset consisted of 4,806 particle projection images and was reconstructed to 6.0 Å resolution.

For the analysis of NP-attached BG505 UFO trimers, we used the localized reconstruction method[164]. Localized reconstruction v1.2.0 was applied to extract subparticles corresponding to trimer antigens from pre-aligned E2p-LD4 and I3-01v9 particle datasets. Each NP presents 20 trimers on the surface. The starting trimer datasets consisted of 153,400 (20 × 7672) and 96,120 (20 × 4806) subparticles for BG505 UFO-E2p-L4P and UFO-10GS-I3-01v9-L7P, respectively. Subparticle datasets were subjected to two rounds of 2D classification and two rounds of 3D classification in Relion/3.0[163] to eliminate cropped, overlapping, and heterogeneously looking particles. For the 3D steps, we applied an HIV-1 Env trimer model from negative-stain EM, low-pass

filtered to 40 Å resolution. The final subset consisted of 6726 and 3723 particles for E2p-L4P and I3-01v9-L7P-displayed BG505 UFO trimers. These subsets were subjected to 3D refinement. In the case of E2p-L4P-displayed trimers, a soft solvent mask around the trimer was used for the 3D refinement and postprocessing steps. C3 symmetry restraints were imposed during refinement. Maps at 7.4 and 10.4 Å resolution were obtained for E2p-L4P and I3-01v9-L7P-bound BG505 UFO trimers, respectively. The resulting maps after 3D refinement and postprocessing, and corresponding half-maps and solvent masks were submitted to the Electron Microscopy Data Bank (EMDB). Model refinement was performed only for the E2p-L4P NP scaffold, using the B-factor-sharpened map after postprocessing. The structure of dihydrolipoyl transacetylase from PDB entry 1B5S[104] was used as a starting model. Iterative rounds of manual model building in Coot[165] and Rosetta relaxed refinement[166] were applied to generate the final structures. Model evaluation was performed using the EMRinger[167] and MolProbity[168] packages, with statistics reported in Table S2. The model was submitted to the Protein Data Bank (PDB).

## Site-specific glycan analysis of BG505 trimer and SApNP immunogens

Three 50 μg aliquots of each sample were denatured for 1 h in 50 mM Tris/HCl, pH 8.0, containing 6 M urea and 5 mM DTT. Next, Env proteins were reduced and alkylated by adding 20 mM iodoacetamide (IAA) and incubated for 1 h in the dark, followed by 1 h of incubation with 20 mM DTT to eliminate residual IAA. The alkylated Env proteins were buffer exchanged into 50 mM Tris/HCl, pH 8.0 using Vivaspin columns (3 kDa) and digested separately overnight using trypsin, chymotrypsin, or elastase (Mass Spectrometry Grade, Promega) at a ratio of 1:30 (w/w). The next day, the peptides were dried and extracted using C18 Zip-tip (MerckMilipore). The peptides were dried again, resuspended in 0.1% formic acid, and analyzed by nanoLC-ESI MS with an Easy-nLC 1200 (Thermo Fisher Scientific) system coupled to a Fusion mass spectrometer (Thermo Fisher Scientific) using higher energy collision-induced dissociation (HCD) fragmentation. Peptides were separated using an EasySpray PepMap RSLC C18 column (75 μm × 75 cm). A trapping column (PepMap 100 C18 3 μM 75 μM × 2 cm) was used in line with the LC prior to separation with the analytical column. The LC conditions were the following: 275 min linear gradient consisting of 0-32% acetonitrile in 0.1% formic acid over 240 min followed by 35 min of 80% acetonitrile in 0.1% formic acid. The flow rate was set to 200 nl/min. The spray voltage was set to 2.7 kV and the temperature of the heated capillary was set to 40 °C. The ion transfer tube temperature was set to 275 °C. The scan range was 400−1600 $m/z$. The HCD collision energy was set to 50%, appropriate for the fragmentation of glycopeptide ions. Precursor and fragment detection were performed using an Orbitrap at a resolution MS1 = 100,000 MS2 = 30,000. The AGC target for MS1 = 4e5 and MS2 = 5e4 and injection time: MS1 = 50 ms MS2 = 54 ms.

Glycopeptide fragmentation data were extracted from the raw file using Byonic™ (Version 3.5) and Byologic™ software (Version 3.5; Protein Metrics). The glycopeptide fragmentation data were evaluated manually for each glycopeptide; the peptide was scored as true-positive when the correct b and y fragment ions were observed along with oxonium ions corresponding to the glycan identified. The MS data were searched using the Protein Metrics 305 N-glycan library. The relative amounts of each glycan at each site, as well as the unoccupied proportion, were determined by comparing the extracted chromatographic areas for different glycotypes with an identical peptide sequence. All charge states for a single glycopeptide were summed. The precursor mass tolerance was set to 4 parts per million (ppm) and 10 ppm for fragments. A 1% false discovery rate (FDR) was applied. Glycans were categorized according to the composition detected. HexNAc(2)Hex(9 − 5) was classified as M9 to M5, HexNAc(3)Hex(5 − 6) X as Hybrid with HexNAc(3)Fuc(1)X classified as Fhybrid. Complex-

type glycans were classified according to the number of HexNAc residues, which are attributed to the number of processed antenna/bisecting GlcNAc (B), and fucosylation (F). For example, HexNAc(3) Hex(3−4)X was assigned to A1, HexNAc(4)X to A2/A1B, HexNAc(5)X to A3/A2B, and HexNAc(6)X to A4/A3B. If all of these compositions had a fucose, then they were assigned to the corresponding FA category. Note that this analytical approach does not distinguish between isomers, which could influence formal assignment of the number of antennae in some cases.

## Global glycan analysis of SApNP immunogens before and after endo H treatment

SDS-PAGE gel bands corresponding to BG505 UFO-10GS-I3-01v9-L7P SApNP were excised and washed three times with alternating 1 ml acetonitrile and water, incubating and shaking for 5 min following addition of each wash solution. All liquid was removed following the final wash stages and N-linked glycans were released in-gel using PNGaseF, (2 µg enzyme in 100 µl $H_2O$) (New England Biolabs) at 37 °C overnight. Following digestion, the liquid was removed from the gel bands and placed into a separate Eppendorf. The gel bands were then washed twice with 100 µl MilliQ $H_2O$ and this was pooled with the original solution. The extracted glycans were then dried completely in a speed vac at 30 °C. The released glycans were subsequently fluorescently labeled with procainamide using 110 mg/ml procainamide and 60 mg/ml sodium cyanoborohydride in a buffer consisting of 70% DMSO, 30% acetic acid. For each sample, 100 µl of labeling mixture was added. Labeling was performed at 60 °C for 2 h. Excess label and PNGaseF were removed using Speed Amide-2 cartridges (Applied Separations). First, the cartridges were equilibrated sequentially with 1 ml acetonitrile, water, and acetonitrile again. Then 1 ml of 95% acetonitrile was added to the procainamide-released glycan mixture and applied to the cartridge, allowing the cartridge to drain by gravity flow. After the mixture has emptied the cartridge, two washes using 97% acetonitrile were performed. To elute the labeled glycans 1 ml HPLC-grade water was added to the cartridges and the elution collected. The elution was then dried completely using a speed vac, before resuspending in 24 µl of 50 mM ammonium formate. A 6 µl aliquot of the resuspended glycans were mixed with 24 µl of acetonitrile and analyzed on a Waters Acquity H-Class UPLC instrument with a Glycan BEH Amide column (2.1 mm × 150 mm, 1.7 µM, Waters), with an injection volume of 10 µl. A gradient of two buffers; 50 mM ammonium formate (buffer A) and acetonitrile (buffer B) was used for optimal separation. Gradient conditions were as follows: initial conditions, 0.5 ml/min 22% buffer A, increasing buffer A concentration to 44.1% over 57.75 min. Following this the concentration of buffer A was increased to 100% at 59.25 min and held there until 66.75 min and the flow rate was dropped to 0.25 ml/min, to fully elute from the column. Finally, the %A was reduced to 20% to prepare for subsequent runs. Wavelengths used for detection of the procainamide label were 310 nm (excitation) and 370 nm (emission). Data were processed using Empower 3 software (Waters, Manchester, UK). The relative abundance of oligomannose-type glycans was measured by digestion with Endoglycosidase H (per sample in 20 µl volume) (Endo H; New England Biolabs). A 6 µl aliquot of labeled glycans was combined with 1 µg endo H to a final volume of 20 µl. Digestion was performed for 1 h at 37 °C. Digested glycans were cleaned using a 96-well PVDF protein-binding membrane (Millipore) attached to a vacuum manifold. Prior to application to the membrane, 100 µl HPLC-grade $H_2O$ was added to each sample. Following equilibration with 150 µl ethanol, and 2× 150 µl HPLC-grade $H_2O$, the sample was added to the 96-well plate and the flow-through was collected in a 96-well collection plate. Each well was then washed twice with HPLC-grade $H_2O$ to a final elution volume of 300 µl. The elution was then dried completely at 30 °C. Prior to analysis the sample was resuspended in 6 µl ammonium formate and 24 µl acetonitrile.

To generate a global glycan profile, a wildtype or glycan-trimmed I3-01v9 SApNP sample was subjected to SDS-PAGE. The corresponding NP bands were processed by PNGaseF to release glycans, which were labeled with procainamide. One aliquot of released glycans was analyzed by ultra-performance liquid chromatography (UPLC), with the chromatogram displayed in green. A second aliquot of released glycans was treated with endo H, which cleaved the first GlcNAc and the procainamide label while depleting oligomannose-type glycans from the sample. This sample was then analyzed by UPLC. The resulting chromatogram was normalized to the untreated aliquot and displayed in pink when merged with the first chromatogram to generate the profile.

## Bio-layer interferometry (BLI)

The antigenic profiles of BG505 Env immunogens, the UFO trimers and SApNPs with wildtype and modified glycan shields, were measured using an Octet RED96 instrument (FortéBio, Pall Life Sciences) against a large panel of HIV-1 NAbs, bNAbs, and nNAbs in the IgG form. All assays were performed with agitation set to 1000 rpm in FortéBio 1× kinetic buffer. The final volume for all solutions was 200 µl per well. Assays were performed at 30 °C in solid black 96-well plates (Geiger Bio-One). For UFO trimers, 5 µg/ml antibody in 1× kinetic buffer was loaded onto the surface of anti-human Fc Capture Biosensors (AHC) for 300 s. For UFO trimer-presenting SApNPs, anti-human Fc Quantitation Biosensors (AHQ) were used because they have been shown to be more suitable for measuring the avidity effect for particulate immunogens[27,89,91,92]. Notably, the respective trimer was also measured using AHQ biosensors to facilitate comparisons of antibody binding with trimer-presenting SApNPs. A 60 s biosensor baseline step was applied prior to the analysis of the association of the antibody on the biosensor to the antigen in solution for 200 s. A two-fold concentration gradient of antigen, starting at 266.7 nM for the UFO trimer, 14.9 nM for the FR SApNP, and 5.5 nM for the E2p and I3-01v9 SApNPs, was used in a titration series of six. The dissociation of the interaction was followed for 300 s. The correction of baseline drift was performed by subtracting the mean value of shifts recorded for a sensor loaded with antibody but not incubated with antigen, and for a sensor without antibody but incubated with antigen. Octet data were processed by FortéBio's data acquisition software v.8.1. Peak signals at the highest antigen concentration were summarized in a matrix and color-coded accordingly to facilitate comparisons between different vaccine platforms and glycan treatments. A separate Octet experiment was performed for the wildtype E2p SApNP in duplicate at the highest concentration (5.5 nM) to determine the intra-experiment signal variation, which is shown in Fig. S4q.

## Mouse immunization and sample collection

Similar immunization protocols have been used in our previous vaccine studies[27,89,91,92,124]. Briefly, Institutional Animal Care and Use Committee (IACUC) guidelines were followed with animal subjects tested in the immunization study. Six-to-eight-week-old female BALB/c mice were purchased from The Jackson Laboratory and housed in ventilated cages in environmentally controlled rooms at The Scripps Research Institute, in compliance with an approved IACUC protocol and Association for Assessment and Accreditation of Laboratory Animal Care (AAALAC) international guidelines. Mice were housed in a controlled environment at 20 °C, 50% humidity, and 12-h light-dark cycles. Mice were immunized at weeks 0, 3, 6, and 9 with 200 µl of antigen/adjuvant mix containing 30 µg of vaccine antigen and 100 µl of adjuvant, AddaVax, Adju-Phos, or Alhydrogel (InvivoGen), via the intraperitoneal (i.p.) route. Blood was collected 2 weeks after each injection. All bleeds were performed through the retro-orbital sinus using heparinized capillary tubes into EDTA-coated tubes. Samples were spun at 1200 rpm for 10 min to separate plasma (top layer) and the rest of the whole blood layer. Upon heat inactivation at 56 °C for 30 min, plasma

was spun at 2000 rpm for 10 min to remove precipitates. The rest of the whole blood layer was diluted with an equal volume of PBS and then overlaid on 4.5 ml of Ficoll in a 15 ml SepMate™ tube (STEMCELL Technologies) and spun at 1200 rpm for 10 min at 20 °C to separate peripheral blood mononuclear cells (PBMCs). Cells were washed once in PBS and then resuspended in 1 ml of ACK Red Blood Cell lysis buffer (Lonza). After washing with PBS, PBMCs were resuspended in 2 ml of Bambanker Freezing Media (Lymphotec). Spleens were harvested at week 11 and ground against a 70-µm cell strainer (BD Falcon) to release splenocytes into a cell suspension. Splenocytes were centrifuged, washed in PBS, treated with 5 ml of ACK lysing buffer (Lonza), and frozen with 3 ml of Bambanker freezing media. Total plasma from week 11 was purified using CaptureSelect™ IgG-Fc (Multispecies) Affinity Matrix (Thermo Scientific) following the manufacturer's instructions. Purified IgG samples from individual mice were analyzed in TZM-bl assays to determine the vaccine-induced neutralizing response.

### Rabbit immunization and sample collection

The Institutional Animal Care and Use Committee (IACUC) guidelines were followed for the animal subjects tested in the immunization studies. For the evaluation of wildtype Env and Env-NP immunogens using a long regimen, rabbit immunization and blood sampling were performed under a subcontract at Covance (Denver, PA). Seven groups of 3-to-4-month-old female New Zealand White rabbits, six rabbits per group, were immunized intramuscularly (i.m.) with 100 µg of HIV-1 Env antigen formulated in 250 µl of adjuvant, AddaVax or Adju-Phos (InvivoGen), with a total volume of 500 µl, at weeks 0, 8, and 24. Blood samples (15 ml each) were collected at day 0 and weeks 1, 2, 4, 6, 8, 10, 12, 16, 20, 24, 26, and 28. Plasma was separated from blood and heat inactivated (56 °C for 30 min) for the TZM-bl neutralization assays. For the assessment of glycan-trimmed Env-NP immunogens using a short regimen, rabbit immunization and blood sampling were performed under a subcontract at ProSci (San Diego, CA). Four groups of female New Zealand White rabbits, six rabbits per group, were intramuscularly (i.m.) immunized with 30 µg of vaccine antigen formulated in 250 µl of adjuvant, AddaVax or Adju-Phos (InvivoGen), with a total volume of 500 µl, at weeks 0, 3, 6, 9, and 12. Blood samples, 20 ml each time, were collected from the auricular artery at day 0 (Pre) and weeks 2, 5, 8, 11, 14, and 16. More than 100 ml of blood was taken at week 16, via cardiac puncture, for PBMC isolation. Plasma samples were heat inactivated for further characterization, and purified rabbit IgGs were assessed in TZM-bl neutralization assays.

### Nonhuman primate immunization and sample collection

Research-naive adult rhesus macaques of Indian origin were genotyped and selected as negative for the protective major histocompatibility complex (MHC) class I alleles Mamu-A*01, Mamu-B*08 and Mamu-B*17. Rhesus macaques were sourced from the Southwest National Primate Research Center (SNPRC) and Tulane National Primate Research Center (TNPRC). Animals from TNPRC were quarantined at SNPRC for 42 days during which a physical exam, ova and parasite evaluation, and three TB skin tests were performed. All experimental procedures were performed at SNPRC in San Antonio, TX, USA according to the guidelines of the AAALAC standards. These female and male macaque experiments were carried out in compliance with all pertinent US National Institutes of Health (NIH) regulations and were approved by the IACUC of SNPRC. Rhesus macaques were moved into research housing and given a physical exam. The animals were randomly assigned to the study groups, which were balanced for age, weight, and gender. Animals were allowed to acclimate to research housing for 14 days before initiation of the study protocol. On the immunization timepoints, animals received a 250 µl subcutaneous or intramuscular injection of vaccine into each quadricep. Injection sites were visually observed at 24 and 48 h post injection. Blood collections were made from the femoral vein under anesthesia. In the study of

wildtype SApNPs, rhesus macaques (age 1.5–16.3 years old, female) were immunized subcutaneously at weeks 0, 8, and 24 with 200 µg antigen formulated with AddaVax (AV, for E2p) or aluminum phosphate (AP, for I3-01v9) and blood was collected at weeks −2, 1, 2, 4, 8, 9, 10, 12, 16, 24, 25, 26, and 28. In the study of glycan-trimmed SApNPs, macaques (age 4.0–11.5 years old, both female and male) were immunized intramuscularly at weeks 0, 4, 12, and 24 with 100 µg antigen formulated with AP (for both E2p and I3-01v9) and blood was collected at weeks 0, 1, 4, 5, 8, 13, 16, 25, and 28. Animals were returned to the colony 28 weeks after the initial vaccination.

### Pseudovirus production and neutralization assays

Pseudoviruses were generated by the transfection of HEK293T cells (ATCC, no. CRL-3216) with an HIV-1 Env expressing plasmid and an Env-deficient genomic backbone plasmid (pSG3ΔEnv), as previously described[169]. HIV-1 Env expressing vectors for BG505 (catalog no. 11518), SF162 (catalog no. 10463), and the 12-virus global panel[170] (catalog no. 12670) and TZM-bl cells were obtained through the NIH AIDS Reagent Program (https://www.aidsreagent.org/). A T332N mutation was introduced into BG505 Env to produce the BG505.T332N clone. Other BG505.T332N mutants were created by introducing mutations as previously described[56,60,69]. Pseudoviruses were harvested 72 h post-transfection for use in neutralization assays. The neutralizing activity of each mouse, rabbit or NHP sample was assessed using a single round of replication pseudovirus assay and TZM-bl target cells, similar to that described previously[169]. Briefly, pseudovirus was incubated with serial dilutions of antibodies or heat inactivated rabbit plasma/NHP sera in a 96-well flat bottom plate for 1 h at 37 °C before TZM-bl cells were seeded in the plate. For purified mouse and rabbit IgGs, a starting concentration of 300 µg/ml was used, or an initial dilution of either 40-times or 100-times for rabbit plasma or NHP sera, and then subjected to 3-times serial dilutions in the TZM-bl assays. Samples were tested in duplicate with a full series of dilutions unless otherwise specified in the Results, with such exceptions including tests against the 12-virus global panel performed in singlet to conserve purified IgG samples. Luciferase reporter gene expression was quantified 60-72 h after infection upon lysis and the addition of Bright-Glo™ Luciferase substrate (Promega). Data were retrieved from a BioTek microplate reader with Gen 5 software. Values from experimental wells were compared against a well containing virus only, with background luminescence from a series of uninfected wells subtracted from both. Dose–response neutralization curves were then fit by nonlinear regression in GraphPad Prism 9.3.1, with $IC_{50}$ and $ID_{50}$ values calculated with constraints set between 0 to 100. As a negative control, pseudoparticles displaying the envelope glycoproteins of MLV (MLV-pps) were tested following the same protocol.

### Histology, immunostaining, and imaging

To study vaccine distribution, trafficking, cellular interaction, and GCs in lymph nodes, BG505 Env immunogens were intradermally injected into mouse footpads using a 29-gauge insulin needle under 3% isoflurane anesthesia with oxygen. Similar immunization and tissue analysis protocols were used in our previous study[124]. The administered dose was 80 µl of antigen/adjuvant mix containing 40 µg of vaccine antigen per mouse or 10 µg per footpad. Mice were euthanized at 30 min to 8 weeks after a single-dose injection. Brachial and popliteal sentinel lymph nodes were collected from both sides of the mouse body for histological analysis. Fresh lymph nodes were merged into frozen section compound (VWR International, catalog no. 95057-838) in a plastic cryomold (Tissue-Tek at VWR, catalog no. 4565). Sample molds were merged into liquid nitrogen and then stored at −80 °C before shipping to The Centre for Phenogenomics in Canada for sample processing, immunostaining, and imaging. Tissue sections were sliced 8 µm thick on a cryostat (Cryostar NX70) and collected on charged slides. Samples were then fixed in 10% neutral buffered

formalin and further permeabilized in PBS that contained 0.5% Triton X-100 before staining. The slides were blocked with protein Block (Agilent) to prevent nonspecific antibody binding. Primary antibody was then applied on the sections and incubated overnight at 4 °C. After washing with TBST, secondary antibodies that were conjugated with either biotin or a fluorophore were used, and the samples were incubated for 1 h at 25 °C. Lymph node sections were stained with bNAbs VRC01[115], PGT124[134], and PGDM1400[135] (1:50), and biotiny-lated goat anti-human secondary antibody (Abcam, catalog no. ab7152, 1:300), followed by streptavidin-horseradish peroxidase (HRP) reagent (Vectastain Elite ABC-HRP Kit, Vector, catalog no. PK-6100) and diaminobenzidine (DAB) (ImmPACT DAB, Vector, catalog no. SK-4105).

To visualize interactions between HIV-1 Env vaccines and cells in mouse lymph nodes, FDCs were labeled using anti-CD21 primary antibody (Abcam, catalog no. ab75985, 1:1800), followed by anti-rabbit secondary antibody conjugated with Alexa Fluor 555 (Thermo Fisher, catalog no. A21428, 1:200). B cells were labeled using anti-B220 antibody (eBioscience, catalog no. 14-0452-82, 1:100) followed by anti-rat secondary antibody conjugated with Alexa Fluor 647 (Thermo Fisher, catalog no. A21247, 1:200). Subcapsular sinus macrophages were labeled using anti-sialoadhesin (CD169) antibody (Abcam, catalog no. ab53443, 1:600) followed by anti-rat secondary antibody conjugated with Alexa Fluor 488 (Abcam, catalog no. ab150165, 1:200). HIV-1 vaccine-induced GCs were studied using immunostaining, GC B cells stained using rat anti-GL7 antibody (FITC; BioLegend, catalog no. 144604, 1:250), T$_{fh}$ cells stained using anti-CD4 antibody (BioLegend, catalog no. 100402, 1:100) followed by anti-rat secondary antibody conjugated with Alexa Fluor 488 (Abcam, catalog no. ab150165, 1:1000), GC cells stained using Bcl6 antibody (Abcam, catalog no. ab220092, 1:300) followed by anti-rabbit secondary antibody conjugated with Alexa Fluor 555 (Thermo Fisher, catalog no. A21428, 1:1000). Nuclei were labeled using 4′,6-diamidino-2-phenylindole (DAPI) (Sigma-Aldrich, catalog no. D9542, 100 ng/ml).

The stained lymph node sections were scanned using an Olympus VS-120 slide scanner with a Hamamatsu ORCA-R2 C10600 digital camera for bright-field and fluorescent images. The transport of HIV-1 BG505 Env immunogens and their induced GCs in mouse lymph nodes were quantified through bright-field and fluorescent images using ImageJ software[171].

### Electron microscopy analysis of protein nanoparticles and lymph node tissues

TEM analysis was conducted by the Core Microscopy Facility at The Scripps Research Institute. To visualize interactions of BG505 UFO Env-SApNPs with FDCs and phagocytic cells in lymph nodes using TEM, mice were injected with 140 µl of antigen/adjuvant mix containing 100 µg of vaccine antigen (40 µl of adjuvant) into the two hind footpads or 50 µg per footpad. Popliteal sentinel lymph nodes were isolated at 2, 12, and 48 h after administration, bisected and immersed in oxygenated 2.5% glutaraldehyde and 4% paraformaldehyde in 0.1 M sodium cacodylate buffer (pH 7.4) fixative overnight at 4 °C. The lymph node tissues were washed using 0.1 M sodium cacodylate buffer and post-fixed in buffered 1% osmium tetroxide and 1.5% potassium ferrocyanide for 1–1.25 h at 4 °C. The samples were rinsed in Corning Cell Culture Grade Water and stained *en bloc* with 0.5% uranyl acetate overnight at 4 °C. The tissues were rinsed with double-distilled H$_2$O and dehydrated through a graded series of ethanol followed by acetone, infiltrated with LX-112 (Ladd) epoxy resin, and polymerized at 60 °C. Ultrathin tissue sections (70 nm) were prepared on copper grids for TEM imaging. Tissue samples were imaged at 80 kV with a Talos L120C transmission electron microscope (Thermo Fisher), and images were acquired with a CETA 16 M CMOS camera.

### Lymph node disaggregation, cell staining, and flow cytometry

GC B cells (GL7$^+$B220$^+$) and T$_{fh}$ cells (CD3$^+$CD4$^+$CXCR5$^+$PD-1$^+$) were characterized using flow cytometry (Fig. S11a). Mice were euthanized at 2, 5, and 8 weeks after a single-dose injection and at 2 and 5 weeks after the boost, which occurred at 3 weeks after the first dose (4 footpads, 10 µg/footpad). Fresh axillary, brachial, and popliteal sentinel lymph nodes were isolated. After mechanically disaggregating the lymph node tissues, samples were merged in enzyme digestion solution in an Eppendorf tube containing 958 µl of Hanks' balanced salt solution (HBSS) buffer (Thermo Fisher Scientific, catalog no. 14185052), 40 µl of 10 mg/ml collagenase IV (Sigma-Aldrich, catalog no. C5138), and 2 µl of 10 mg/ml DNase (Roche, catalog no. 10104159001). Lymph node tissues were incubated at 37 °C for 30 min and filtered through a 70 µm cell strainer. Samples were spun down at 400 × $g$ for 10 min to isolate cell pellets, which were resuspended in HBSS blocking buffer with 0.5% (w/v) bovine serum albumin and 2 mM EDTA. Anti-CD16/32 antibody (BioLegend, catalog no. 101302, 1:50) was added to block the nonspecific binding of Fc receptors, while the sample solution was kept on ice for 30 min. Samples were then transferred to 96-well microplates with pre-prepared cocktail antibodies, which included the Zombie NIR live/dead stain (BioLegend, catalog no. 423106, 1:100), Brilliant Violet 510 anti-mouse/human CD45R/B220 antibody (BioLegend, catalog no. 103247, 1:300), FITC anti-mouse CD3 antibody (BioLegend, catalog no. 100204, 1:300), Alexa Fluor 700 anti-mouse CD4 antibody (BioLegend, catalog no. 100536, 1:300), PE anti-mouse/human GL7 antibody (BioLegend, catalog no. 144608, 1:500), Brilliant Violet 605 anti-mouse CD95 (Fas) antibody (BioLegend, catalog no. 152612, 1:500), Brilliant Violet 421 anti-mouse CD185 (CXCR5) antibody (BioLegend, catalog no. 145511, 1:500), and PE/Cyanine7 anti-mouse CD279 (PD-1) antibody (BioLegend, catalog no. 135216, 1:500). The cell samples mixed with antibody cocktail were placed on ice for 30 min and centrifuged to remove excess antibody. After washing with the HBSS blocking solution, cells were fixed with 1.6% paraformaldehyde (Thermo Fisher Scientific, catalog no. 28906) in HBSS on ice for 30 min. The samples were then placed in HBSS blocking solution at 4 °C. Sample events were acquired by a 5-laser AZE5 flow cytometer (Yeti, Bio-Rad) with Everest software at the Core Facility of The Scripps Research Institute. The data were analyzed using FlowJo 10 software.

### Statistics and reproducibility

SEC was performed for all HIV-1 Env trimers and SApNPs at least once for in vitro characterization and multiple times during protein production for animal studies. Representative SEC profiles were selected for comparison. SDS-PAGE and BN-PAGE were performed for all BG505 UFO trimer and SApNP constructs at least once during screening, with selected constructs run on the same gel to facilitate visual comparison. Negative-stain EM was performed routinely for all NP constructs during in vitro characterization and protein production for animal studies. DSC was performed up to three times to validate key thermal parameters and thermographs. DLS was performed multiple times to measure the hydrodynamic diameter of each SApNP construct. BLI was performed for all trimers and SApNP constructs in singlet, with a separate experiment performed in duplicate for the E2p SApNP to assess the variability between technical replicates. All ELISA and neutralization assays were performed with duplicates, except when testing some mouse samples in the MLV and SF162 assays and all mouse and rabbit samples in the 12-virus assays that were performed as singlets due to the limited sample availability. Data collected from 8 mice per group in the immunization study and 3–10 mice per group in the mechanistic study were used for statistical analyses. To assess the improvement in FVR, comparison was made for each vaccine platform (trimer or SApNP) between the wildtype group and the glycan-trimmed group(s) that had the highest FVR. This statistical comparison was performed using a two-sided Fisher's exact test. For the vaccine

transport and GC study in lymph nodes, different vaccine groups were compared using one-way ANOVA, followed by Tukey's multiple comparison post hoc test. Statistical significance is indicated as the following in the figures: ns (not significant), $*p < 0.05$, $**p < 0.01$, $***p < 0.001$, $****p < 0.0001$. Both data processing and graphing were performed using GraphPad Prism 9.3.1 software.

### Reporting summary
Further information on research design is available in the Nature Portfolio Reporting Summary linked to this article.

## Data availability
The EM data generated in this study have been deposited in the Electron Microscopy Data Bank (EMDB, https://www.ebi.ac.uk/emdb/), under accession codes EMD-28540, EMD-28541, EMD-28542, EMD-28543, and EMD-28555. The structural data have been deposited in the Protein Data Bank (PDB, https://www.rcsb.org/), under accession code 8EQN [www.rcsb.org/structure/8EQN]. RAW files associated with the site-specific glycan analysis can be accessed at massive.ucsd.edu under doi:10.25345/C55D8NQ7W. The authors declare that the data supporting the findings of this study are available within the article and its Supplementary Information files. Source data are provided with this paper.

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

## Acknowledgements

Y.-N.Z. acknowledges support from the Natural Sciences and Engineering Research Council of Canada (NSERC) for the postdoctoral fellowship. We thank M. Ganguly, K. Duffin, and G. Ossetchkine at The Centre for Phenogenomics for their technical support in immunohistology. We thank K. Vanderpool, T. Fassel, and S. Henderson of the Core Microscopy Facility at The Scripps Research Institute for their expert assistance in the TEM analysis. We thank A. Saluk, B. Seegers, and B. Monteverde of the Flow Cytometry Core Facility at The Scripps Research Institute for their technical support in flow cytometry. We thank P. Berman and D. Burton for helpful discussions and M. Arends, S. Auclair, and V. Tong for proofreading the manuscript. This work was supported by the International AIDS Vaccine Initiative (IAVI) through grant INV-008352/OPP1153692 (M.C.) and the IAVI Neutralizing Antibody Center through the Collaboration for AIDS Vaccine Discovery grant (INV-034657) (I.A.W., A.B.W.), and INV-008352/OPP1153692 (M.C.) funded by the Bill and Melinda Gates Foundation; Scripps Consortium for HIV/AIDS Vaccine Development (CHAVD 1UM1 AI144462) (M.C., A.B.W. and I.A.W.); HIV Vaccine Research and Design (HIVRAD) program (P01 AI124337) (J.Z.); NIH grants R01 AI129698 (J.Z.) and R01 AI140844 (J.Z., I.A.W.); Southwest National Primate Research Center through grants P51 OD011133 and U42OD010442 and Tulane National Primate Research Center through grant P51 OD011104 from the Office of Research Infrastructure Programs, NIH (R.L., C.C.); Ufovax/SFP-2018-0416 (J.Z.) and Ufovax/SFP-2018-1013 (J.Z.).

## Author contributions

Project design by Y.-N.Z., J.P., L.H., I.A.W., and J.Z.; immunogen expression and purification by M.E. and L.H.; negative-stain EM and cryo-EM by A.A., Y.-Z.L., J.C., and A.B.W.; glycan analysis by J.D.A., M.N., and M.C.; mouse and rabbit sample preparation, and neutralization assays by J.P. and L.H.; NHP immunization and sample collection by D.C., P.F., A.G., J.D., R.L., and C.C.; NHP serum neutralization and data analysis by M.E. and L.H.; mouse immunization, lymph node isolation, immunohistology, TEM, and flow cytometry by Y.-N.Z.; manuscript written by Y.-N.Z., A.A., J.P., J.D.A., I.A.W., M.C., A.B.W., and J.Z. All authors were asked to comment on the manuscript. The Scripps Research Institute manuscript number is 30193.

## Competing interests

The authors declare no competing interests.
