## [Peer Review File · Nature Communications]

Single-component multilayered self-assembling protein nanoparticles presenting glycan-trimmed uncleaved prefusion optimized envelope trimers as HIV-1 vaccine candidatesReviewers' Comments:

Reviewer #1:

Remarks to the Author:

To the authors,

Zhang et al. describe the design of HIV-1 Env-based nanoparticles and the effect of glycan trimming on their folding and immunogenicity. The authors used their previously reported UFO design for Env and used three different nanoparticle platforms. Subsequently, the authors trimmed glycans on these immunogens and used them to immunize mice, rabbits and macaques. The number of (advanced) techniques that were used to study these immunogens is truly impressive (cryoEM to study NPs, site-specific analysis, immunohistochemistry, TEM) as is the number of experiments (Octet runs, neutralization assays) that have been performed. Unfortunately, however, I failed to see what could actually be learned from this enormous study. The lack of proper controls for the immunogenicity studies also did not help (see below).

In summary, in my opinion, while most of it seemed technologically sound, this manuscript lacks focus and probably needs thorough reevaluation by its authors. My concerns are listed below.

Main concerns:

1. In many cases, claims could not be substantiated by the data, mostly because the comparison could not be made. For example, the authors claim that glycan trimming (GT) improved NAb responses in macaques (lines 490-492, 501). But the macaques receiving GT versions received 4 shots (instead of 3) at different intervals, using IM immunization instead of SQ. This is not a fair comparison. Similarly, rabbits were immunized with trimers nanoparticles with Addavax, except I3-01v9 (receiving AP). The authors claim I3-01v9 outperformed the other groups: but is this not an adjuvant effect?
2. Figure 6A: why did the authors not include a regimen 3: 4x WT? This would be a direct comparison to show if glycan trimming is indeed affecting immunogenicity. The lack of such a comparator arm makes it difficult to make comparisons in terms of immunogenicity.
3. The authors often do not present statistical evidence for their claims: e.g. line 405-407: I3-01v9 is claimed to outperform the rest (see comment 1), but eyeballing suggests that UFO trimer elicited similar titers (at least at wk26). Additionally, it is unclear how (immunogenicity) groups are compared: Figure 6B: where is the SOSIP/HR1N-redesign group? Was it tested (not clear from Fig 6A)? Or is this compared to what is known from literature?
4. The whole paper and study lack focus. E.g. in the design phase locking domains are used, but it is unclear if these ended up in animal testing. Why was suddenly a cocktail immunization group included? The authors did a lot of work, but I often failed to see why this was done and what the value for it was.
5. The use of all kinds of different ways to measure serum neutralization: ID50 or IC50 titers are commonly used. However, reporting IC30 is highly unusual and only seemed to have been done to "boost" the data. E.g. Fig. S5G suggests that the authors achieved relatively broad neutralization in 1/3 of the BG505 UFO trimer immunized rabbits! This seems highly improbable and this has not been achieved by any other group using similar immunogens and animal models.
6. Usually, serum neutralization assay results differ between experiments, this is normal. However, the differences between the repeats of the individual mouse sera neutralization (Table on top in Fig. S6D) is extreme: from non-neutralizing to relatively potent neutralization and back. To be blunt: how can these data be trusted? Or did I misinterpret these data?
7. The authors tested lots of different bNAbs in Octet. I would be curious if the authors also checked binding of germline versions of the PGT12x family of bNAbs (or other germline bNAbs). I can imagine that glycan trimming might have a beneficial effect on this.
8. Site-specific analysis was performed on non-processed, swainsonine and kifunensis treated Env-NP proteins. But many other assays and the immunizations were performed EndoH-treated Env(-NP). Why did the authors not check EndoH-treated Env-NP using site-specific analysis?? This should

definitely be done and it would be highly informative for interpreting the antigenicity and immunogenicity data.

Minor comments

1. Line : 85-87. This seems a strange statement, since manipulating individual glycans is important to engage germline bNAbs (i.e. VRC01-class, PGT12x-class). So, in contrast, manipulating individual glycans might actually be part of a general solution for a broadly protective vaccine. Numerous studies have been published on this: Jardine Science 2013, Escolano/Gristick Nature 2019, etc. The 64 reference showed that strain-specific glycan holes (i.e. 241/289 glycan hole in BG505) are not easily broadened, which is something else entirely. Other examples (e.g. Ringe et al. J Virol. 2019 and del Moral-Sanchez et al. npj Vaccines 2022 come to mind) showed that you can redirect responses when closing (or opening) glycan holes. This statement should be removed or clarified.
2. Some language should be used more carefully: line 137: I would not call those responses "potent"
3. Unclear: line 142: redesigned? Newly designed bent? Or from the earlier design?
4. % polydispersity in DLS is missing?
5. Line 278-281: slightly unclear what was done. Reader might think both samples were put at 100C and then processed at 25C for 5h or 37C for 1h?
6. Line 441-443: it is not escape, but simply polyclonal sera not targeting this epitope.
7. What was the resolution of the attached Envs in the reconstructions? The text seems to only refer to the resolution of the core of the NPs?
8. The different names of the nanoparticles in the manuscript seem overly complicated: "SApNP", why not use NP when possible.

Reviewer #2:

Remarks to the Author:

This review is about the germinal Center analysis portion of this manuscript.

1. Specifying how many LN were collected and the exact side of immunization and collection (afferent lymph nodes) would be important.
2. In Figure S 11, the Numbers of Axis for all the plots are missing, and the plots are out of focus, so it isn't easy to see the cut-off.
3. Is SSC in Figure A in logarithmic scale?
4. Fluorescence Minus One (FMO) control, or isotype controls should be included for PD-1 and CXCR5 in Figure S11.

Reviewer #3:

Remarks to the Author:

The authors characterized the properties of single-component, self-assembling protein nanoparticles that present the BG505 UFO trimer with wildtype and modified glycans, showing trimming the glycan shield improved Env recognition in multiple layers. Importantly, the mechanism of vaccine-induced immunity was examined, which further benefit the next step of HIV-1 vaccine development. The work is of great importance and well-deigned. The data and results adequately support the conclusion. There is no additional evidence needed from my view. The methodology employed in this research has been well-described, which meet the expected standards. I believe this manuscript is high quality and of broad interest to readership.

Reviewer #4:

Remarks to the Author:

The manuscript by Zhang et al described some extensive studies of a couple of vaccine constructs based the concept of uncleaved prefusion-optimized (UFO) originated from the PI's lab, including rational design, structural characterization, glycan analyses, antigenicity, immunogenicity in mice, rabbits as well as NHPs of these constructs for the WT and glycan modified forms. In addition, they authors had also checked the antigen retention of these constructs in mice. These constructs were showed to be able to induce mostly autologous neutralizing antibody responses in the animal models, and the glycan trimmed forms increased the frequency of vaccine responders and steered the Ab responses away from the glycan hole of HIV-1 Env trimer. Unfortunately, immunogenicity data did not provide any breakthrough for the neutralizing Ab responses that one might hope for, nevertheless, these results will help us to further understand the immunogenicity of the stabilized trimer and its limitation as an HIV-1 vaccine candidate.

Some specific comments and suggestions:

- The manuscript contains two types of studies (immunogenicity and antigen retention) that could be separated into two manuscripts; they authors may want to further strengthen the common theme that might help the readers.
- The glycan trimming would steer the Ab responses away from the glycan hole is very interesting, and any more detailed epitope mapping will help to understand the phenomena.
- Text, figures, and tables: "446-52D" is a typo, it should be "447-52D".
- It is a bit puzzling that trimming glycan for the SApNPs increases binding of both trimer-specific bNAbs (PGDM1400/PGT145) and V3 tip-directed nNAbs (19b/447-52D) (Fig.4A). Although it is possible the UFO trimer is compatible with both bindings, could there be a mixture of species?
- Fig. 1D: LD7 should be LD7; LD4 should be LD4.
- All the section subtitles are statements, except one "How HIV-1 Env ...".

Response to Reviewer Comments

Reviewer #1

General comments: Zhang et al. describe the design of HIV-1 Env-based nanoparticles and the effect of glycan trimming on their folding and immunogenicity. The authors used their previously reported UFO design for Env and used three different nanoparticle platforms. Subsequently, the authors trimmed glycans on these immunogens and used them to immunize mice, rabbits and macaques. The number of (advanced) techniques that were used to study these immunogens is truly impressive (cryoEM to study NPs, site-specific analysis, immunohistochemistry, TEM) as is the number of experiments (Octet runs, neutralization assays) that have been performed. Unfortunately, however, I failed to see what could actually be learned from this enormous study. The lack of proper controls for the immunogenicity studies also did not help (see below).

In summary, in my opinion, while most of it seemed technologically sound, this manuscript lacks focus and probably needs thorough reevaluation by its authors

Response: We thank the reviewer for carefully reading our manuscript and recognizing the technical strength of this study. We understand that our manuscript presents an unusual volume of data covering several different research areas. We consider the integration of multiple scientific disciplines to be one of the strengths of this comprehensive study. We have revised our manuscript to emphasize our key findings and provide context in order to clarify its significance to the HIV-1 vaccine research community.

Although it could be argued that neither native-like Env nor nanoparticle vaccines are entirely novel (since we and several other groups have reported these types of studies in the past), we have systematically optimized our UFO trimer-presenting nanoparticle constructs and combined them with a novel element – glycan trimming – to create a rational HIV-1 vaccine strategy that has been shown to induce autologous tier-2 neutralizing antibody (NAb) responses in mice, rabbits, and nonhuman primates (NHPs) with higher responder frequency. The consistency of these results across three animal species suggests that this may be a general vaccine strategy. More studies would be needed to explore the full potential of this strategy.

The central finding of this study – that glycan trimming can improve the NAb response to native-like Env trimers in solution or displayed on nanoparticles (NPs) – touches on a major issue in HIV-1 vaccine research. Ambiguity surrounding the role of glycans in shaping the host immune response to the native HIV-1 Env has led to the development of several conceptually distinct vaccine strategies, e.g., mimicking the glycan shield on native virions, adding or removing glycans, or sequentially deleting and reintroducing glycans to guide B cell maturation towards broadly neutralizing antibodies (bNAbs). Notably, questions about the role of the glycan shield concern not only the NAb recognition of HIV-1 Env but also the transport and retention of HIV-1 Env vaccines. For example, it was recently reported that glycans may be essential for localization and retention of NP immunogens in follicular dendritic cell (FDC) networks within germinal centers (GCs) [Tokatlian et al., 2019, *Science* 363:649-654]. A follow-up study showed that NP immunogens treated with endoglycosidase H (the enzyme used to trim off glycans in our study) significantly reduced localization to the GC follicles [Read et al., 2022, *Cell. Rep.* 38:110217]. The theory that having a dense glycan shield may improve vaccine trafficking, lymph node targeting, and ultimately NAb response has been the basis of several prominent HIV-1 vaccine designs. However, our comparative analyses of wildtype (WT) and glycan trimmed Env immunogens *in vitro* and *in vivo* produced rather different results, which indicate that glycan trimming can improve NAb elicitation with no adverse effect on NP trafficking and retention. Removal of the immunosuppressive mannose groups also resulted in stronger GC reactions to the NP immunogens.

With regard to the reviewer's second point, we agree that some of the immunogenicity studies would have benefited from additional control groups. This paper presents research that was conducted over the course of four years, over which time our focus and methods evolved in response to new data from our own studies and from the literature. Unfortunately, limited time and resources prevented us from repeating every early study, so we prioritized those which we deemed most critical to the central finding of this paper: that glycan

trimming can improve the NAb response to HIV-1 Env with greater responder frequency. At the same time, much of this early data, while not perfect, is still relevant and valuable in that it adds supporting evidence, sheds light on particular points of interest, or suggests avenues for future research. For this reason, we have retained much of this data in the paper. Based on the reviewer's feedback, we have revised our manuscript to more clearly distinguish data that is intended to be observational rather than evidentiary in nature.

Specific comments and suggestions:

1. *In many cases, claims could not be substantiated by the data, mostly because the comparison could not be made. For example, the authors claim that glycan trimming (GT) improved NAb responses in macaques (lines 490-492, 501). But the macaques receiving GT versions received 4 shots (instead of 3) at different intervals, using IM immunization instead of SQ. This is not a fair comparison.*

Response: The nonhuman primate data shown in **Fig. 7** were generated in two separate studies that were performed at different stages of our research. We agree that these two studies cannot be considered a head-to-head comparison of WT vs. GT UFO-Env SApNPs, and we did not intend to present them as such. Based on the reviewer's comment, we have revised the summary statements on page 23, lines 512-514, to describe the limitations of the current study and provide future plans: "However, the current studies, while promising, were mostly observational due to the differences in the immunization protocol (e.g., group size, vaccine dose, and injection route). More extensive NHP studies with matching regimens would be required to verify these findings".

Our initial nonhuman primate study was conducted to compare the NAb responses generated by the E2p/AV vs. I3-01v9/AP formulations. The vaccine regimen (3 subcutaneous injections at w0, w8, and w24) was modeled after a large-scale nonhuman primate study of HIV-1 Env trimer vaccines reported in the literature [Pauthner et al., 2017, *Immunity* 46: 1073-1088]. The second study was designed to verify the results we had obtained in mice using glycan-trimmed (GT) immunogens with or without wildtype (WT) boost. We adopted an extended-boost vaccine regimen (4 injections at weeks 0, 4, 12, and 24) based on our findings in mice that SApNPs can be well retained in draining lymph nodes (dLNs) for 5 weeks (**Fig. 8c**). We used intramuscular (IM) rather than subcutaneous (SC) route to make the results more compatible with human vaccination. As discussed in our response to the general comments, the small number of macaques available made it impossible to both repeat the regimen used in the first study and include additional GT groups, so we chose to prioritize the latter. One critical factor in this decision was the fact that, in the first study, the WT SApNP formulations only induced neutralizing activity in one macaque with an ID₅₀ barely above the threshold for positive neutralization (**Fig. 7b**). Repeating these groups would not be the most productive use of the precious limited resources. In addition, while SC outperformed IM in terms of NAb response to a soluble Env trimer vaccine, the difference was not statistically significant [Pauthner et al., 2017, *Immunity* 46: 1073-1088], making it unlikely that the SC-to-IM switch was responsible for the higher immunogenicity and vaccine responder frequency observed in the second (GT) study compared to the original (WT) study. Similarly, because both vaccine regimens consist of multiple injections at prolonged intervals over a 24-week period, it seems unlikely that a single 4th injection in the second study would induce a shift from near-complete absence of neutralizing activity to about 50% responder frequency. The rationales for choosing different regimens in the NHP studies are explained in the main text, e.g., on page 22, lines 477-478, "We first tested wildtype SApNPs in NHPs using a regimen modeled after a recent NHP study of BG505 trimers", and on page 22, lines 489-491, "We next tested glycan-trimmed BG505 SApNPs in NHPs using two regimens adopted from the mouse study with extended boost intervals". We have added a new statement on page 22, lines 494-496, to further explain our rationale and highlight our focus in the second NHP study: "Due to the limited availability of animals, the experiments were designed to prioritize the evaluation of the glycan trimming strategy with a focus on the comparison of homologous vs. heterologous boost".

Similarly, rabbits were immunized with trimers nanoparticles with Addavax, except I3-01v9 (receiving AP). The authors claim I3-01v9 outperformed the other groups: but is this not an adjuvant effect?

We thank the reviewer for raising this important point. In this study, we formulated our E2p and I3-01v9 SApNPs with adjuvants that are commonly used in human and animal vaccines: aluminum phosphate (AP), aluminum hydroxide (AH), and AddaVax (AV). Our previous studies on HIV-1, HCV, Ebola virus (EBOV), and SARS-CoV-2 SApNPs (He et al., 2018, *Sci. Adv.* 4: eaau6769; He et al., 2020, *Sci. Adv.* 6: eaaz6225; He et al., 2021, *Nat. Commun.* 12: 2633; He et al., 2021, *Sci. Adv.* 7: eabf1591) demonstrated that E2p could induce robust NAb responses when paired with AddaVax, while I3-01v9 was most effective when paired with AP, so these were the SApNP/adjuvant combinations used in this comprehensive study where time and/or animals needed to be conserved. We have reviewed the manuscript to confirm that in all such cases we state clearly that results are attributed to the SApNP/adjuvant combination rather than the SApNP alone (e.g., lines 452, “to compare the E2p/AV and I3-01v9/AP formulations”). Based on the reviewer’s comments, we have also removed the statement that I3-01v9 outperformed other groups in the rabbit study.

2. *Figure 6A: why did the authors not include a regimen 3: 4x WT? This would be a direct comparison to show if glycan trimming is indeed affecting immunogenicity. The lack of such a comparator arm makes it difficult to make comparisons in terms of immunogenicity.*

Response: Thanks for highlighting this point. The all-WT regimens were included in our comparison but were not shown in the schematic representation of the immunization regimen in **Fig. 6a**, because this figure was intended to emphasize homologous *vs.* heterologous (WT *vs.* GT) vaccine boosting strategies. For soluble trimers, the data from an extensive SOSIP study [Hu et al., 2015, *J. Virol.* 89: 10383–10398] and our previous study [He et al., 2018, *Sci. Adv.* 4: eaau6769] were included for comparison and shown in the first column of panel #1 in **Fig. 6b** (labeled as “SOSIP/HR1_N-redesign”). Briefly, in the former study, the BG505 SOSIP trimer was tested in mice with diverse adjuvants and immunization setups but failed to induce any tier-2 NAb response, while in the latter study, we immunized mice with 13 HR1-redesigned BG505 trimers, which also failed to elicit tier-2 NAb. For trimer-presenting E2p and I3-01v9 SApNPs, the 4×WT regimen was used for the immunizations and the results are shown in the first column of each panel of **Fig. 6b**, labeled “E2p (AV/AP) WT” and “I3-01v9 (AP) WT”, respectively. We have relabeled the control groups for all three panels of **Fig. 6b** with the symbols • to represent SOSIP and HR1_N-redesigned trimers (published data) and * to represent wildtype E2p or I3-01v9 SApNPs (from **Fig. 5** of this study). We have modified the figure legends, on page 68, lines 1725-1729, to clearly indicate the inclusion of all-WT regimens as control in the comparison and provide the source of data, either derived from the previous publications (references provided) or generated in the current study.

3. *The authors often do not present statistical evidence for their claims: e.g. line 405-407: I3-01v9 is claimed to outperform the rest (see comment 1), but eyeballing suggests that UFO trimer elicited similar titers (at least at wk26). Additionally, it is unclear how (immunogenicity) groups are compared: Figure 6B: where is the SOSIP/HR1N-redesign group? Was it tested (not clear from Fig 6A)? Or is this compared to what is known from literature?*

Response: We thank the reviewer for noting a similar ID₅₀ titer from the trimer group at w26. Based on the reviewer’s comment, we have removed this statement in the revised manuscript. The purpose of **Fig. 5E** is to illustrate the longitudinal patterns of NAb responses in rabbit plasma elicited by trimer and various types of SApNP formulations. We did not perform statistical analysis for individual timepoints. Since tier-2 NAb responses can be readily elicited by Env immunogens in rabbits, in this study we focused on whether glycan trimming can overcome the “glycan hole” problem in rabbit immunization (**Fig. 5f vs. Fig. 6f**).

Regarding the labeling “SOSIP/HR1_N-redesign” in **Fig. 6b**, the reviewer was correct that these data were from studies reported in the literature, including an early SOSIP trimer study [Hu et al., 2015, *J. Virol.* 89: 10383–10398], where the BG505 SOSIP trimer was tested with diverse adjuvants and immunization setups but failed to induce any tier-2 NAb response, and our previous study [He et al., 2018, *Sci. Adv.* 4: eaau6769], where 13 HR1-redesigned BG505 trimers were tested in mice and found to be ineffective in eliciting tier-2 NAb responses. As noted above, we have relabeled these groups in **Fig. 6b** and modified the legends to clearly indicate the inclusion of all WT regimens as control and provide the source of the all-WT data.

4. *The whole paper and study lack focus. E.g. in the design phase locking domains are used, but it is unclear if these ended up in animal testing. Why was suddenly a cocktail immunization group included? The authors did a lot of work, but I often failed to see why this was done and what the value for it was.*

Response: We understand that our manuscript presents an unusual volume of data covering construct design, biochemical/biophysical characterization, structural characterization, glycan profiling, antigenic profiling, immunogenicity (in three animal models), and vaccine mechanism. Our intention was to provide the field with a clear roadmap of how we arrived at our vaccine strategy and final vaccine constructs in a coherent narrative. We think that it is imperative to fully explain our rationale and report all relevant data in one comprehensive article rather than publishing several papers. Given the complexity involved in the process of rational vaccine design, we are also concerned that splitting one paper into several will create gaps and hinder the readers' understanding of our work. Nonetheless, based on the reviewer's comment, we have further revised the manuscript to improve the clarity of the writing.

The locking domains (LDs) are an integral part of our multilayered 1c-SAPNP design. In our previous study (He et al., 2021, *Nat. Commun.* 12: 2633), we reported the LD concept, selected nine LDs from the protein database for their ability to stabilize the NP shell, and displayed the redesigned EBOV GP trimers on SAPNPs containing various LDs. In the EBOV vaccine study, E2p + LD4 and I3-01v9 + LD7 appeared to be the best combinations and were used in the final constructs. However, it remained unclear whether these two combinations are specific to EBOV GP or can be extended to other viral antigens. Therefore, in this HIV-1 vaccine study, we tested all NP + LD combinations in the design phase. Remarkably, the same two combinations showed the best yield and stability for HIV-1 UFO-Env SAPNPs. For example, LD7 and LD7 + PADRE increasing the I3-01v9 yield by 5-fold and 8-fold relative to the WT construct, respectively. Once the optimal constructs were determined (**Fig. 1**), all subsequent experiments (**Figs. 2-10**), including structural characterization, glycan profiling, antigenic profiling, and animal studies, were based on E2p + LD4 and I3-01v9 + LD7. Therefore, the full names of our final vaccine constructs are BG505 UFO.664-E2p-LD4-PADRE and UFO.664-10GS-I3-01v9-LD7-PADRE. To avoid repeating the long names, we used the NP backbone names (i.e., E2p and I3-01v9) to refer to the full vaccine constructs. We have revised the summary statement of the "design" section, on page 9, lines 185-189, to clarify that only the finalized UFO trimer-presenting SAPNP constructs based on E2p + LD4 + PADRE and I3-01v9 + LD7 + PADRE were further characterized *in vitro* and *in vivo* (**Figs. 2-10**).

5. *The use of all kinds of different ways to measure serum neutralization: ID₅₀ or IC₅₀ titers are commonly used. However, reporting IC₃₀ is highly unusual and only seemed to have been done to "boost" the data. E.g. Fig. S5G suggests that the authors achieved relatively broad neutralization in 1/3 of the BG505 UFO trimer immunized rabbits! This seems highly improbable and this has not been achieved by any other group using similar immunogens and animal models.*

Response: We thank the reviewer for noting the use of different sample types and metrics in neutralization assays, which were based on the considerations to minimize the background noise in pseudovirus assays (ID₅₀ vs. IC₅₀) and detect low-level neutralization signals (30% vs 50% inhibition of the activity).

ID₅₀ vs. IC₅₀: As reported in an extensive study of BG505 SOSIP.664 trimer (Hu et al., 2015, *J. Virol.* 89: 10383–10398) and in our previous study (He et al., 2018, *Sci. Adv.* 4: eaau6769), mouse serum contains non-specific antiviral activities that may lead to false positives in TZM-bl assays. For this reason, purified IgGs, instead of sera or plasma, were used to obtain reliable readouts for mouse samples, and IC₅₀, instead of ID₅₀, was calculated from the neutralization data. Judging by the MLV-pp assays, samples from the first rabbit study didn't show any non-specific neutralizing activity; however, a few rabbit samples from the second study did exhibit anti-MLV signals. Although we performed TZM-bl assays using both serum and purified IgG samples for the second rabbit study, we only reported the IgG data. For the first rabbit study and both macaque studies, we performed serum neutralization assays and reported the ID₅₀ values.

We have revised the manuscript to clarify the use of different sample types in neutralization assays. For example, we modified the manuscript, on pages 17, lines 382-384, to add “Immunoglobulin G (IgG) was purified from mouse samples to reduce non-specific antiviral activity in neutralization assays”.

30% vs. 50%: We chose to report the ID₃₀ and IC₃₀ titers based on a recent study from David Montefiori’s lab showing that ID₃₀ can be used as a relevant cut-off for detecting and reporting low-level neutralization [Seaman et al., *J Immunol Methods* 2020; 479: 112736]. We did not intend to use these data to claim or imply that our results are significantly better than or different from those previously reported in the literature. The ID₃₀ and IC₃₀ titers were shown only for assays using the 12-virus panel and we confined ourselves to the ID₅₀ data when drawing important conclusions. For example, on page 19, lines 418-420, we state that “all SApNP formulations failed to generate a strong cross-clade NAb response to other tier 2 isolates in a 12-virus global panel, with no advantage in breadth seen for the Env mixture vaccines (**Fig. S5g**).”

Nonetheless, we agree with the reviewer that the use of ID₃₀ and IC₃₀ titers may cause unnecessary confusion and therefore have removed all of the ID₃₀ and IC₃₀ data from the manuscript.

6. *Usually, serum neutralization assay results differ between experiments, this is normal. However, the differences between the repeats of the individual mouse sera neutralization (Table on top in Fig. S6D) is extreme: from non-neutralizing to relatively potent neutralization and back. To be blunt: how can these data be trusted? Or did I misinterpret these data?*

Response: We thank the reviewer for bringing up this point. We realize that the use of the word “repeat” in the table is misleading. We would like to clarify that the “1st, 2nd, 3rd, 4th” rows in this table were not repeated neutralization assays on the same IgG samples, but instead results from independent immunization experiments. To confirm our finding that glycan trimming can significantly improve the vaccine responder frequency and ensure that this was not an experimental anomaly caused by a particular batch of proteins, adjuvants, or animals, we repeated the mouse immunization experiment using the same protocol two to four times. In the revised manuscript, we have changed the word from “repeat” to “Expt. repeat” and included a legend in the table to state that “Each Expt. repeat indicates a different immunization experiment. Results were obtained from multiple immunizations using the same protocol”. We also included a sentence in the figure legend, on page 68, lines 1728-1729, to explain the experimental repeats used in **Fig. 6b**.

7. *The authors tested lots of different bNAbs in Octet. I would be curious if the authors also checked binding of germline versions of the PGT12x family of bNAbs (or other germline bNAbs). I can imagine that glycan trimming might have a beneficial effect on this.*

Response: We thank the reviewer for raising this interesting possibility. Based on the reviewer’s comment, we produced two PGT121 antibodies, the putative germline GL (Sok et al., 2013, *PLoS Pathog* 9: e1003754) and an early heavy-chain immediate 3H109L (Garces et al., 2015, *Immunity* 43: 1053), and assessed their binding to WT and glycan-trimmed Env immunogens using BLI. The results indicate that glycan trimming has no effect on interactions between BG505 Env immunogens and these two PGT121 lineage variants. We have revised the manuscript, on page 16, lines 343-346, to report this new finding.

8. *Site-specific analysis was performed on non-processed, swainsonine and kifunensis treated Env-NP proteins. But many other assays and the immunizations were performed EndoH-treated Env(-NP). Why did the authors not check Endo H-treated Env-NP using site-specific analysis?? This should definitely be done and it would be highly informative for interpreting the antigenicity and immunogenicity data.*

Response: We thank the reviewer for this comment. In this study, we generated both global (**Fig. 3g**) and site-specific glycan profiles for endo H-treated UFO-Env SApNPs, but only included the global glycan profile in the submitted manuscript. During this project, we came to realize that while both types of glycan analysis provide important information, the global profile would be more useful in revealing what type of oligomannose-type glycans is more sensitive to or protected from the endo H treatment. As shown in **Fig. 3g**, Man5-7 glycans were completely trimmed by endo H, whereas a small fraction of Man8 and 40% of the Man9 glycans remained intact after treatment. In addition, given the methods used in this study, the

global analysis of release glycans was also more accurate than the site-specific analysis for the trimmed materials. The reason is that the molecular weight of the peptide would account for most of the molecular weight of a glycopeptide that contains a N-acetylglucosamine (GlcNAc) residue resulted from endo H treatment, thus affecting the accuracy of database matching. Based on this consideration, we plan to evaluate other methods for the site-specific analysis and report the results in future follow-up studies.

9. *Line: 85-87. This seems a strange statement, since manipulating individual glycans is important to engage germline bNAbs (i.e. VRC01-class, PGT12x-class). So, in contrast, manipulating individual glycans might actually be part of a general solution for a broadly protective vaccine. Numerous studies have been published on this: Jardine Science 2013, Escolano/Gristick Nature 2019, etc. The 64 reference showed that strain-specific glycan holes (i.e. 241/289 glycan hole in BG505) are not easily broadened, which is something else entirely. Other examples (e.g. Ringe et al. J Virol. 2019 and del Moral-Sanchez et al. npj Vaccines 2022 come to mind) showed that you can redirect responses when closing (or opening) glycan holes. This statement should be removed or clarified.*

Response: We thank the reviewer for spotting this confusing statement. Manipulating individual glycans around a bNAb epitope could certainly be part of a solution for a broadly protective vaccine. The problem, however, is that such a vaccine solution would be limited to one bNAb epitope while antibody responses to other epitopes remain unoptimized. As for glycan holes, as pointed out by the reviewer, one can certainly redirect responses (though not necessarily to bNAb epitopes) by closing or opening individual glycan holes. The problem is similar to manipulating glycans around a bNAb epitope in that such manipulation will be limited to a particular glycan hole. Since glycan holes are strain-specific, their manipulations will also be strain-specific and Env-dependent. Nonetheless, based on the reviewer's comment, we have modified this statement in the revised manuscript to clarify our points, one page 4, lines 85-87, "The manipulation of individual glycans may improve targeting of a bNAb epitope or avoiding a glycan hole but may not provide a general vaccine solution as it could be Env- or strain-specific."

10. *Some language should be used more carefully: line 137: I would not call those responses "potent"*

Response: In our previous study (He et al., 2018, *Sci. Adv.* 4: eaau6769), an I3-01 SApNP presenting HR1-redesigned BG505 trimers induced a strong tier-2 NAb response in wildtype mice. In the follow-up study (Kumar et al., 2021, *mBio*, 12: e0042921), we isolated the first tier-2 murine NAb (M4H2K1), which targets the C3/V4 epitope with an IC_{50} of 0.067 $\mu\text{g/ml}$ approaching two of the most potent human bNAbs PGT121 ($IC_{50}=0.029 \mu\text{g/ml}$) and PGT128 ($IC_{50}=0.013 \mu\text{g/ml}$). Regardless, based on the reviewer's comment, we have revised the manuscript to remove or change the word "potent" throughout the main text.

11. *Unclear: line 142: redesigned? Newly designed bent? Or from the earlier design?*

Response: Our UFO trimer design consists of a modified HR1 N-terminal region, which was truncated to an 8-aa segment and computationally redesigned, and a 10-GS linker at the cleavage site [Kong et al., 2016, *Nat. Commun.*, 7: 12040]. The HR1 modification is crucial for the improved Env properties, whereas the cleavage site linker appears to be less important and is only included in the UFO construct to eliminate the need for furin co-expression during trimer production. Therefore, the cleaved, HR1-redesigned trimers were tested in several of our early studies [Kong et al., 2016, *Nat. Commun.*, 7:12040; He et al., 2018, *Sci. Adv.* 4: eaau6769; Kumar et al., 2021, *mBio*, 12: e0042921]. To avoid any confusion, we have modified the text, on page 7, lines 142-143, to change "redesigned" to "computationally optimized" for clarity.

12. *% polydispersity in DLS is missing?*

Response: We thank the reviewer for noting this oversight. We have included a new table of polydispersity indices (PI) in **Fig. S1b**. In summary, E2p SApNPs appear to have slightly higher PI values than FR and I3-01v9 SApNPs with the highest PI value reaching 0.18. We have also modified the manuscript, on page 9, lines 183-184, to add "...wide distribution (51.9-69.0 nm), consistent with higher polydispersity index (PI) noted for some E2p samples (**Fig. S1b**)".

13. *Line 278-281: slightly unclear what was done. Reader might think both samples were put at 100C and then processed at 25C for 5h or 37C for 1h?*

Response: We thank the reviewer for pointing out this confusion. We have clarified this section in the revised manuscript, on page 13, lines 278-282, as “Next, reducing SDS-PAGE was used to analyze E2p and I3-01v9 SApNPs treated with endo Hf followed by subsequent enzyme removal using the MBP-specific resin (see Methods). Notably, the endo H treatment was performed either at 25°C for 5 hours or, alternatively, at 37°C for 1 hour after fully denaturing the SApNPs at 100°C (**Fig. 3f**)”.

Line 441-443: it is not escape, but simply polyclonal sera not targeting this epitope.

Response: We have revised this sentence on page 20, lines 444-445, to “In contrast to those from the trimer groups, most samples from the SApNP groups appeared to lack NAbs to this immunodominant glycan epitope.”

14. *What was the resolution of the attached Envs in the reconstructions? The text seems to only refer to the resolution of the core of the NPs?*

Response: The local resolution of the Env-corresponding portion prior to localized reconstruction was very low (<40 Å) due to the flexible linkage that exists between the nanoparticle core and Env. By using a localized reconstruction method, we were able to isolate the signal corresponding to Env subparticles and reconstruct them to the resolutions listed in **Fig. 2**. To address the reviewer’s comment, we have revised the manuscript, on page 10, lines 208-209, to provide the resolution of the BG505 Env trimer before and after localized reconstruction for both E2p and I3-01v9.

15. *The different names of the nanoparticles in the manuscript seem overly complicated: “SApNP”, why not use NP when possible.*

Response: We thank the reviewer for noting the subtle but critical differences in the terminology. Context is a determinant factor here. Nanoparticle (NP) is a broad term that can be used to describe lipid-, liposome-, organic-, inorganic-, and protein-based nanoscale (1-100nm) particles. We used the most stringent term “1c-SApNP” to distinguish the specific type of single-component self-assembling protein nanoparticles which we developed and used in this study from other nanoparticle designs in the field, e.g., two-component protein nanoparticles. In many cases, a slightly loose term “SApNP” was used to simplify the writing if the context was clear. In some cases, we also used the term “NP” for our vaccine constructs when describing a general scenario (e.g., particulate vs. trimer vaccines). Based on the reviewer’s comment, we have checked these terms throughout the manuscript to ensure consistency and clarity.

Reviewer #2

1. *Specifying how many LN were collected and the exact side of immunization and collection (afferent lymph nodes) would be important.*

Response: We appreciate the reviewer’s comments and suggestions. Mice were euthanized after a single-dose injection or after the boost injection (4 footpads, 10 µg/footpad). Brachial and popliteal sentinel lymph nodes (6 lymph nodes in total) were isolated from both sides of the mouse body. In the revised manuscript, we have clarified these details in the main text, on page 24, lines 523-524, and in the Methods section, on page 49, lines 1111-1114.

2. *In Figure S 11, the Numbers of Axis for all the plots are missing, and the plots are out of focus, so it isn't easy to see the cut-off.*

Response: In the revised manuscript, we have included all the numbers of Axis for all the plots and used high-resolution images for the flow plots shown in **Fig. S11**.

3. *Is SSC in Figure A in logarithmic scale?*

Response: Yes, the logarithmic scale is for SSC in **Fig. S11a**.

4. *Fluorescence Minus One (FMO) control, or isotype controls should be included for PD-1 and CXCR5 in Figure S11.*

Response: In the revised manuscript, we have included FMO controls for PD-1 and CXCR5 in **Fig. S11**.

Reviewer #3

General comments: The authors characterized the properties of single-component, self-assembling protein nanoparticles that present the BG505 UFO trimer with wildtype and modified glycans, showing trimming the glycan shield improved Env recognition in multiple layers. Importantly, the mechanism of vaccine-induced immunity was examined, which further benefit the next step of HIV-1 vaccine development. The work is of great importance and well-deigned. The data and results adequately support the conclusion. There is no additional evidence needed from my view. The methodology employed in this research has been well-described, which meet the expected standards. I believe this manuscript is high quality and of broad interest to readership.

Response: We thank the reviewer for the positive evaluation of our work, especially on the impact of glycan trimming to HIV-1 vaccine development, the vaccine mechanism, and the overall high quality.

Reviewer #4

General comments: The manuscript by Zhang et al described some extensive studies of a couple of vaccine constructs based the concept of uncleaved prefusion-optimized (UFO) originated from the PI's lab, including rational design, structural characterization, glycan analyses, antigenicity, immunogenicity in mice, rabbits as well as NHPs of these constructs for the WT and glycan modified forms. In addition, they authors had also checked the antigen retention of these constructs in mice. These constructs were showed to be able to induce mostly autologous neutralizing antibody responses in the animal models, and the glycan trimmed forms increased the frequency of vaccine responders and steered the Ab responses away from the glycan hole of HIV-1 Env trimer. Unfortunately, immunogenicity data did not provide any breakthrough for the neutralizing Ab responses that one might hope for, nevertheless, these results will help us to further understand the immunogenicity of the stabilized trimer and its limitation as an HIV-1 vaccine candidate.

Response: We thank the reviewer for the concise summary of our main findings and recognizing the two major advantages of our glycan trimming strategy: increasing the vaccine responder frequency and steering antibody responses away from glycan holes on HIV-1 Env. Although the immunogenicity data did not reach the level of a “breakthrough” that one might ideally hope for, we nevertheless would assert that these results still represent a significant step forward in HIV-1 vaccine research, considering that these immunogenicity data were obtained by using the most common (and arguably least effective) adjuvants – aluminum phosphate (AP) and AddaVax. These immunogenicity data provide a baseline or benchmark for the future development of more effective HIV-1 Env vaccine formulations. In a recent study, we examined the adjuvant effect on the NAb responses induced by our SARS-CoV-2 spike SApNP vaccines (Zhang et al., 2021, *Sci. Adv.* 7: eabj3107). Remarkably, while both SApNP/AP and SApNP/AddaVax formulations produced reasonable ID₅₀ titers at 2650 and 3954 against the Wuhan-Hu-1 strain, respectively, the use of CpG (a TLR9 agonist) substantially enhanced the serum neutralization to an ID₅₀ titer of 6784, highlighting the impact of a potent adjuvant to the vaccine-induced NAb response. Given the encouraging results for the SApNP/AP formulation in this study, we are hopeful that a potent adjuvant will improve the tier-2 NAb response and responder frequency for our glycan-trimmed HIV-1 UFO-Env SApNP vaccines. The adjuvant effect, as well as other factors in the vaccine formulation, will be the focus of our future investigations.

Specific comments and suggestions:

1. *The manuscript contains two types of studies (immunogenicity and antigen retention) that could be separated into two manuscripts; they authors may want to further strengthen the common theme that might help the readers.*

Response: We thank the reviewer for this comment. In our response to the first reviewer's comment #4, we explained our rationale for presenting all the data related to our vaccines in a single article.

We are fully aware that our manuscript presents an unusual volume of data. Our intention was to provide the field with a clear roadmap of how we arrived at our final vaccine constructs and vaccine strategy in a coherent narrative. We agree with the reviewer that it could be split into two papers, one focusing on vaccine design and immunogenicity and the other focusing on vaccine trafficking, retention, and GC reactions. However, given the recent studies on the role of the glycan shield in lymph node targeting and retention of Env-NP immunogens [Tokatlian et al., 2019, *Science* 363:649-654; Read et al., 2022, *Cell. Rep.* 38:110217], it is very likely that we would be asked to provide the SApNP retention data. For this reason and others, we feel that it is necessary to explain our rationale and report all the experimental data in one article rather than separating the results into two papers. We are also concerned that splitting the paper will create gaps and hinder the readers' understanding of our work. Nonetheless, based on both reviewers' feedback, we have thoroughly revised the manuscript to improve the clarity of the writing.

2. *The glycan trimming would steer the Ab responses away from the glycan hole is very interesting, and any more detailed epitope mapping will help to understand the phenomena.*

Response: We thank the reviewer for the suggestion to further investigate the epitope specificity of tier-2 NAb responses induced by GT vs. WT SApNPs. We will work closely with our collaborators to determine crystal structures and obtain EM models for the single-cell-sorted NAbs in complex with the wildtype and glycan-trimmed trimers. The detailed structural mapping of NAb epitopes will require an intensive amount of experimental work that will be included in our follow-up studies.

3. *Text, figures, and tables: "446-52D" is a typo, it should be "447-52D".*

Response: We have corrected this typo in the revised manuscript.

4. *It is a bit puzzling that trimming glycan for the SApNPs increases binding of both trimer-specific bNAbs (PGDM1400/PGT145) and V3 tip-directed nNAbs (19b/447-52D) (Fig.4A). Although it is possible the UFO trimer is compatible with both bindings, could there be a mixture of species?*

Response: We thank the reviewer for noting this pattern. The BG505 UFO trimer tested in this study does not contain any Env mutations that disrupt the binding of nNAbs 19b and 447-52D to the V3 tip. Therefore, the trimer-specific V2 apex and the immunodominant V3 tip are equally accessible on the surface of a UFO trimer, regardless of being in solution or displayed on a protein NP. For the soluble trimer, glycan trimming exerted a similar effect on antibody recognition of the V2 apex and V3 tip, reducing their binding to bNAbs PGDM1400/PGT145 and nNAbs 19b/447-52D, respectively. Interestingly, enhanced antibody binding was observed for both wildtype and glycan-trimmed UFO-Env SApNPs; this might be attributed to the avidity effect caused by the multivalent display of trimers. On the SApNP surface, antibodies – bNAbs or nNAbs alike – can bind to the exposed V2 apex and V3 tip of the neighboring trimers, resulting in higher binding signals in the BLI assays. It is also worth noting that "quantitation" biosensors were used for SApNPs in the BLI assays, because this type of biosensor is more effective in detecting "local antigen density", a major difference between SApNPs presenting a dense array of trimers and soluble trimers.

We speculate that a phenomenon called glycan promiscuity [Sok et al., 2014, *Sci. Transl. Med.*, 6: 236ra63] may provide a plausible explanation as to why glycan trimmed SApNPs bind well to those glycan-reactive bNAbs that recognize specific mannose groups. As exemplified by the glycan-V3 supersite, various glycan sites and moieties can substitute for one another when interacting with glycan-reactive bNAbs. The avidity effect on the NP surface then further compensates for the weakened glycan-bNAb interactions on a trimmed trimer by enabling interactions with surrounding trimers or to surrounding glycans that are not completely

trimmed. As a result, the glycan trimmed SApNPs showed rather balanced binding to bNAbs that target various protein and glycan epitopes on the gp120 head.

Fig. 1D: LD7 should be LD7; LD4 should be LD4.

Response: We have revised the manuscript to change LD₇ to LD7, and LD₄ to LD4.

5. *All the section subtitles are statements, except one “How HIV-1 Env ...”.*

Response: We thank the reviewer for noting this inconsistency. In the revised manuscript, we have changed the subtitle to a statement “Interactions of HIV-1 Env SApNPs with FDCs and phagocytic cells in lymph nodes”. To be consistent, we have also changed the subtitle of the next section to “Induction of robust GC reactions by HIV-1 Env trimers and SApNPs”.

Reviewers' Comments:

Reviewer #1:

Remarks to the Author:

The authors have improved the manuscript and it is now much easier to follow due to changes at key parts of the manuscript. I also like to thanks for the elaborate and clear explanations per comment. Below some remaining comments and suggestions.

1. I am still puzzled by the discrepancy of their findings regarding trafficking of their findings versus earlier work (Tokatlian Science/Read Cell rep). Could the authors comment on this more in the discussion?
2. Line 411: one might argue that tier-1 V3-focused responses is not advantageous. And I do not see how the particles are performing better than the soluble trimers.

In response to comment #8: Considering the central role EndoH trimmed immunogens play in the study it would be highly valuable to include site-specific data if this can be done reliably: estimates of occupancy could already be highly informative. Otherwise, the authors should at least highlight this point/weakness of the study in the discussion section.

Reviewer #2:

None

Reviewer #4:

None

Reviewer #5:

None

Response to Reviewer Comments (Round #2)

Reviewer #1

General comments: *The authors have improved the manuscript and it is now much easier to follow due to changes at key parts of the manuscript. I also like to thank for the elaborate and clear explanations per comment.*

Response: The reviewers' previous comments and suggestions have helped us to significantly improve the quality and clarity of the manuscript. We thank reviewer #1 for the positive feedback. We have further revised our manuscript based on reviewer #1's new comments and suggestions in the Editorial Checklist. To differentiate from the previous revision, the new modifications are highlighted in green shading.

Specific comments and suggestions:

1. *I am still puzzled by the discrepancy of their findings regarding trafficking of their findings versus earlier work (Tokatlian Science/Read Cell rep). Could the authors comment on this more in the discussion?*

Response: We thank the reviewer for highlighting this point. We have further clarified it in the Discussion section, on page 31, line 686: "Notably, the methods used to track NP immunogens in lymph node tissues may result in different experimental outcomes. In our current study, the trafficking and retention of I3-01v9 SApNPs were quantified through immunostaining of HIV-1 Env trimers displayed on SApNPs using a human bNAb (VRC01) at individual time points, whereas in the previous studies, fluorescent dyes conjugated to protein NPs may have affected the readout due to their signal decay over time (94, 95)."

2. *Line 411: one might argue that tier-1 V3-focused responses is not advantageous. And I do not see how the particles are performing better than the soluble trimers.*

Response: Thanks for noting this point. We have updated our statement regarding the tier 1 NAb responses against SF162 in the manuscript, on Page 19, line 412: "Of interest, when testing tier 1 clade B SF162, a detectable NAb response was observed as early as week 4 for the two I3-01v9 groups."

3. *In response to comment #8: Considering the central role EndoH trimmed immunogens play in the study it would be highly valuable to include site-specific data if this can be done reliably: estimates of occupancy could already be highly informative. Otherwise, the authors should at least highlight this point/weakness of the study in the discussion section.*

Response: We thank the reviewer for this comment. As explained in our previous response letter, glycan trimming poses a technical challenge to our current method for site-specific glycan analysis, and we propose to address this issue in our follow-up studies. The information about site occupancy is critical for developing glycan-trimmed Env immunogens for diverse HIV-1 subtypes. Based on the reviewer's suggestion, we have modified the Discussion section, on page 31, line 697, to highlight this point: "Second, in addition to the global glycan profiles, site-specific analysis may provide critical information to assist in development of glycan-trimmed Env immunogens for diverse HIV-1 subtypes. Site occupancy combined with structural analysis will help explain how antibodies recognize Env epitopes after glycan trimming."